# Existing function in primary visual cortex is not perturbed by new skill acquisition of a non-matched sensory task

Brian B. Jeon[1,2,3], Thomas Fuchs[2,3,4], Steven M. Chase[1,2,3] & Sandra J. Kuhlman [1,2,3,4✉]

Acquisition of new skills has the potential to disturb existing network function. To directly assess whether previously acquired cortical function is altered during learning, mice were trained in an abstract task in which selected activity patterns were rewarded using an optical brain-computer interface device coupled to primary visual cortex (V1) neurons. Excitatory neurons were longitudinally recorded using 2-photon calcium imaging. Despite significant changes in local neural activity during task performance, tuning properties and stimulus encoding assessed outside of the trained context were not perturbed. Similarly, stimulus tuning was stable in neurons that remained responsive following a different, visual discrimination training task. However, visual discrimination training increased the rate of representational drift. Our results indicate that while some forms of perceptual learning may modify the contribution of individual neurons to stimulus encoding, new skill learning is not inherently disruptive to the quality of stimulus representation in adult V1.

[1] Department of Biomedical Engineering, Carnegie Mellon University, Pittsburgh, PA 15213, USA. [2] Center for the Neural Basis of Cognition, Carnegie Mellon University and University of Pittsburgh, Pittsburgh, PA 15213, USA. [3] Neuroscience Institute, Carnegie Mellon University, Pittsburgh, PA 15213, USA. [4] Department of Biological Sciences, Carnegie Mellon University, Pittsburgh, PA 15213, USA. ✉email: skuhlman@cmu.edu

Successful integration of new skills into an existing network requires balancing the maintenance of perception and action with the acquisition of new function. New skill learning, whether it is perceptual, motor, or abstract, involves changes in cellular physiology that are distributed across the brain[1–4]. Locally, up to 50% of the neurons within a circuit are recruited during task-specific behaviors, and it is observed that synaptic plasticity is pervasive among the activated neurons, particularly in the early phases of training[3,5]. The extent to which these widespread adaptive changes disrupt or otherwise impact existing circuit function is an area of active investigation[6,7]. Although addressing this issue is fundamental to understanding the biological constraints on learning, it is challenging to investigate because it requires identifying the neurons responsible for improved performance and monitoring these same neurons longitudinally throughout learning, as well as monitoring existing network function before and after new skill acquisition. We met these challenges by combining two-photon calcium imaging with a brain computer interface (BCI) behavioral paradigm. We chose to train animals on an abstract task using a BCI so that we could reinforce a pattern of neural activity that was distinct from neural patterns associated with salience in the animals' home-cage conditions. As such, training was a unique experience, different from previously learned relationships, and therefore allowed us to unambiguously test whether perturbation of existing function is inherent to sensory learning. The use of a genetically encoded calcium indicator allowed neural activity of the same neurons to be tracked before and after training. Mice were trained to perform a de novo task wherein they earned a reward by modulating the activity of a selected set of primary visual cortex (V1) neurons in a user-defined pattern. Stimulus tuning and discriminability were quantified before and after skill acquisition to assess maintenance of visual function.

It is well-established that similar to monkeys[8], adult mice can learn to control BCIs using neurons in primary sensory cortex as well as motor cortex, in a goal-oriented manner that relies on the same plasticity mechanisms that are engaged during perceptual and motor learning[2,4,9–13]. To implement BCIs, neural activity is generally recorded electrically (e.g., using intracortical electrodes, surface EEGs, or EcoG), or more recently from optical signals using genetically encoded calcium sensors[14]. In all cases, neural activity is directly coupled to the movement of a device such as a cursor or robotic arm and is typically coupled to an effector that provides feedback regarding the neural trajectory generated by the subject. Here we used a one-dimensional auditory cursor to provide sensory feedback. Our implementation of BCI can be considered a multimodal, non-matched sensory task, given that an auditory feedback cue was associated with reward, and the reward was earned by modulating the activity of neurons in the visual cortex.

We focused our study on V1 because the functional output of this circuit is well-characterized, and therefore training-induced alterations can be readily detected. For example in adult animals, enhancing eye-specific input by transiently closing one eye destabilizes orientation preference in single neurons as well as signal correlation between pairs of neurons[15]. Modification is not limited to deprivation paradigms. Sensory representations are persistently altered by reinforced behavioral training under conditions in which specific visual cues are associated with salience[16–20], and learning impacts the representation of non-trained stimuli as well[21]. These studies raise the possibility that every time new associations are made, existing function is altered. On the other hand, recruitment of new neurons selective for task-relevant stimuli has been shown to be context-dependent in training paradigms that cue reward contingencies[16] or carry context-specific information[22]. These observations are an indication that new associations can be learned without perturbing existing function, in situations where the task goals are recognized by the subject to be distinct from previously learned behaviors and associations.

We found that de novo skill training induced plasticity in the majority of V1 neurons in our imaging field of view. In addition, pairwise noise correlation was altered in a manner that persisted outside of the trained context. Despite these changes, stability of neural tuning to visual stimuli presented outside of the task context was not perturbed. Furthermore, response amplitude of individual neurons during vision remained stable, and decoding analysis revealed that the estimated amount of information carried in V1 was unchanged. Thus, stimulus representation was robust to new skill learning. Our results directly demonstrate that a new, non-matched sensory skill can be integrated into existing sensory networks without disrupting previously developed function.

Given that the BCI task was designed to be multimodal in nature, we sought to determine whether our results apply to a sensory-matched perceptual task. In a second cohort of mice the stability of stimulus representation, outside of the trained context, was assessed before and after learning a visual discrimination task. Similar to BCI training, neural tuning to visual stimuli and the estimated amount of information in V1 was largely unperturbed. However, in contrast to BCI training, we found that stimulus representation at the population level was altered. These results indicate new skill learning is not inherently destabilizing, and that some forms of perceptual learning may alter the contribution of individual neurons to stimulus encoding.

## Results

**V1 neurons are capable of driving an optical BCI.** Our goal was to assess the stability of stimulus representation before and after learning. Therefore prior to BCI training, two baseline visual stimulation (VS) sessions were recorded, referred to here as VS baseline 1 and VS baseline 2, respectively. The acquisition of two VS baseline sessions allowed the stability of tuning to grating stimuli of varying orientation and spatial frequency to be assessed before BCI training was initiated. A third VS session was recorded after successful BCI learning, and is referred to as VS post-learning. The third session allowed the impact of training on the stability of tuning to be assessed by comparing stimulus responses in the VS baseline 2 and VS post-learning sessions. The reward delivery device was not visible during the passive viewing VS imaging sessions, nor was auditory feedback engaged. As such, multiple cues were present that distinguished the VS sessions from the BCI training sessions.

To facilitate BCI learning, mice were first conditioned to associate a tone with a reward, prior to being introduced to the BCI task. Mice were subjected to a Go/No-go pitch association task in which a 15 kHz pitch was associated with a water reward, and a 5 kHz pitch was associated with the absence of reward. Animals were required to actively lick before a reward was released. We considered a mouse to have successfully learned the task when mice had 80% performance while performing at least 75 trials in a session. Six out of seven mice learned the task (Supplementary Fig. 1). Four of the mice that learned the task completed BCI training, one mouse was trained on a BCI control experiment where the auditory pitches were not driven by neural activity, and one mouse was removed due to a procedural issue which prevented cross-session alignment of direct neurons. Both the BCI training and the BCI control experiment started within 8 days of learning the pitch association.

In the BCI paradigm, the activity of six V1 glutamatergic excitatory neurons was directly coupled to a one-dimensional

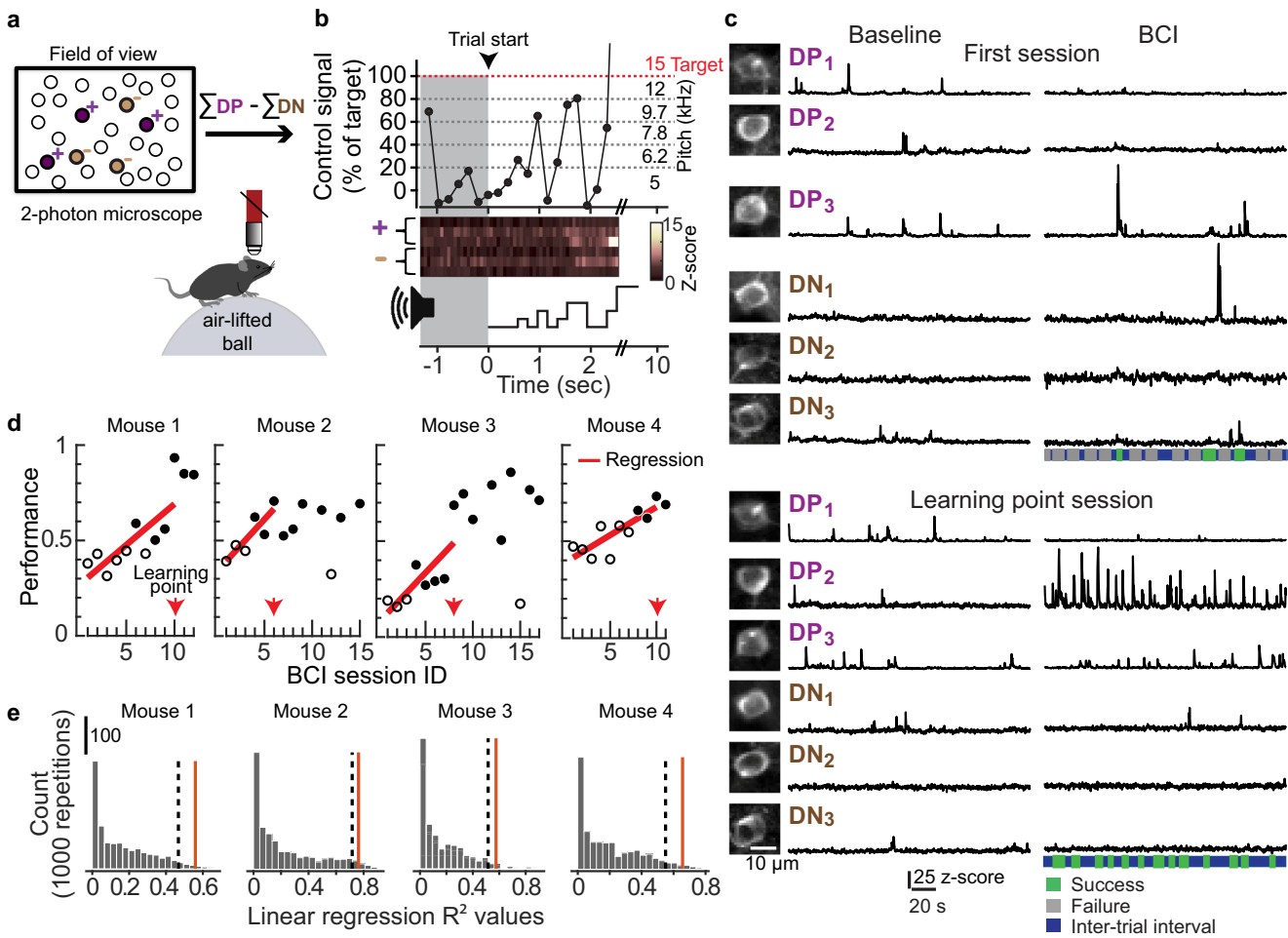

**Fig. 1 Mice gained volitional control of the BCI within 10 days. a** Schematic of BCI set-up in awake mice. Six neurons were selected to drive a one-dimensional auditory cursor. In all cases, three direct positive (DP, purple), and three direct negative (DN, brown) neurons were selected. **b** The control signal was mapped to 6 auditory feedback pitch frequencies; the target pitch frequency was 15 kHz. Example trial in a trained mouse. The calcium signal of individual direct neurons was acquired at 15.5 Hz and accumulated for three frames prior to updating the pitch frequency. Pre-trial start epoch (1 s minimum, gray shading) ended when no running was detected and the control signal was less than the target threshold. The time of success was defined as the time at which the target threshold was reached, 2.6 s in this case. **c** Example activity of 6 direct neurons (200 s for each condition). The same direct neurons were selected across sessions (insets). The total number of success (green) and failure (gray) trials in the session is indicated below. **d** Fraction of trials in which the target was reached. The first BCI session ID in which the linear regression was significant is referred to as the 'learning point' (arrow; mouse #1, $p = 0.013$, $n = 10$ sessions; mouse #2, $p = 0.023$, $n = 6$ sessions; mouse #3, $p = 0.005$, $n = 8$ sessions; mouse #4, $p = 0.005$, $n = 10$ sessions). Sessions in which performance was significantly higher than session 1 are indicated in black (2-sample proportions Z-test, corrected for 12,13,16 or 11 multiple comparisons for mouse #1–4 respectively). **e** Null distributions used to calculate chance performance, for each mouse ($n = 1000$). The target threshold and session numbers for each distribution was matched to the threshold used in the corresponding experimental conditions. In all four mice, the actual $R^2$ value (red solid line) was greater than the 95th percentile (one-sided) of the null distribution (dashed line). Source data are provided as a Source Data file.

auditory feedback cursor (Fig. 1a–c, Supplementary Figs. 2–5). Expression of the calcium sensor was directed by the Vglut1 or EMX1 promoter (Supplementary Table 1). The one-dimensional control signal used to drive the device was computed in the following manner: Three of the six neurons were assigned a positive value such that increases in their summed Z-scored activity drove the cursor closer to a target threshold (direct positive neurons, DP 1–3), and three neurons were assigned a negative value such that increases in their summed Z-scored activity moved the cursor farther away from a target threshold (direct negative neurons, DN 1–3). The target threshold was determined separately for each mouse at the start of BCI training, so that pre-learning success rates would be in the range of 20–30% (Supplementary Fig. 6a, b). The same six neurons were used to drive the BCI throughout the experiment and their

corresponding threshold was held constant (Supplementary Fig. 6c, Supplementary Movies 1–4). See Methods for details on the selection of direct neurons and target threshold. When the control signal reached the target threshold, the auditory cursor pitch was set to 15 kHz and a water reward was released, if the animal licked. The animal had 10 s to reach the target. If the target was not reached within that time period, the trial was scored as a failure. Mice were required to be still at the beginning of each trial (see Methods for trial structure details). The BCI task was performed in the dark, and locomotion was monitored (Supplementary Movie 5).

All mice exhibited significant improvement in performance, defined as the frequency of trials in which the target was reached, within 4–8 days of training. Performance of a given session was considered significantly improved from the first session using a

two-sample proportions Z-test, corrected for multiple comparisons (see Methods for analysis details). The rate of learning varied across mice. Therefore, to examine the changes in direct neurons that were associated with improved performance, we defined a learning state that could be compared across mice. We identified the earliest session in which linear regression of performance against session number was significant ($\alpha < 0.05$). This session was referred to as the learning point (LP) session and was used to assess learning-associated changes in the direct neurons. All mice maintained the skill after the LP session; task performance was significantly higher in the last BCI session relative to the first BCI session in all mice (Fig. 1d). We also found that the latency to reach target threshold was reduced on the LP session compared to the first session (average median latency across mice, session 1: $4.03 \pm 0.15$ s, LP session: $2.88 \pm 0.27$ s, paired $t$-test $p = 0.011$).

In BCI task design used here, although there was learning pressure to keep the activity of the three DN neurons low, there were no additional constraints placed on the indirect neurons. Thus, maximum performance could be achieved without fully solving the credit assignment problem[23,24] at the time of the LP session, i.e., subjects were not required to identify and *selectively* alter the activity of the direct neurons to improve performance. Nonetheless, the subjects can be considered to have gained volitional control of the device, given the significant improvement in the frequency of threshold crossings (Fig. 1d).

To exclude the possibility that instrument noise or random fluctuations in the activity of the direct neurons contributed to improved performance, we verified that the coefficient of determination, $R^2$ of the linear regression, was significantly higher than chance in all four mice (Fig. 1e). Chance was defined post-hoc; the activity of indirect neurons was used to generate a null distribution. Random sets of six neurons were selected from the tracked pool of indirect neurons, and the control signal that each of the random sets would have generated was computed. This was repeated 1000 times to generate the null distribution. Performance across sessions was calculated and linear regression fits were made. Chance was defined as the 95th percentile of this distribution. Importantly, this analysis serves as a control to established that selecting combinations of neurons not coupled to the device did not result in the same improvement in performance across training sessions.

An additional control was performed to control for any systematic increases in neural activity that may spontaneously occur with exposure to elements of the BCI task, such as repeated head fixation and reward delivery. We played back the auditory sequences generated by BCI-trained mouse #1 to a different control mouse. Three 'fictive' direct positive and three fictive direct negative neurons were selected following the procedure described above, and tracked across 14 sessions; the six fictive neurons were not directly coupled to reward delivery or auditory feedback. The calculated performance of the six fictive neurons remained below 0.5 for all sessions, and the $R^2$ of the linear regression was below chance (Supplementary Fig. 7a, b). The same control animal was used to generate additional null distributions matched to the experimental conditions of each of the four BCI-trained mice. Consistent with the results in Fig. 1e, the $R^2$ of the linear regressions generated from the experimental mice were above chance (Supplementary Fig. 7c).

We characterized the activity of the six direct neurons at the time of success and noted that most of the drive to cross the target threshold in BCI session 1 came from the DP neurons, and this was also true in the LP session. Often one DP neuron dominated in a given trial, although there were cases of coordination among DP neurons (Fig. 2, Supplementary Figs. 8a–11a). Because DP neurons were the primary drivers of threshold crossing, next we

examined the extent to which activity among the DP neurons varied between session 1 and the LP session. In each of the four mice, the number of trials that a given DP neuron contributed to threshold crossing shifted between session 1 and the LP session (two-sample proportions Z-test, Supplementary Figs. 8a–11a), thus there was cross-session flexibility in how the target threshold was reached. In addition, the activity level at the time of threshold crossing changed. For example, in mouse #1, DP neuron #2 had increased activity levels at the time of success. In each mouse, at least one DP neuron had a significant change in activity level at the time of success between session 1 and the LP session (Fig. 2b, Supplementary Table 2). Given that in many trials a single DP neuron was responsible for driving the control signal to target threshold, we confirmed that even in the lowest amplitude control-signal crossings, neural activity, and not noise, was responsible for threshold crossing by visually inspecting the minimum control signal trial for the first session and the LP session in all 4 mice (Supplementary Figs. 8b–11b).

It is well established that locomotion during VS increases response gain in V1 excitatory neurons[25–33]. Therefore, a potential concern is that locomotion or general arousal could drive the DP neurons to target threshold. In our conditions, this is unlikely for the following three reasons. First, in baseline conditions of darkness such as experienced during the BCI task, in contrast to VS, only a small fraction of excitatory neurons is modulated by locomotion. The mechanism of this modulation is distinct from the mechanism that mediates the widespread locomotion-induced response gain, and a similar number of neurons decrease activity levels as increase[34]. Consistent with this previous report, the activity of direct neurons was weakly modulated by locomotion, and the amount of modulation was similar between DP and DN neurons (Supplementary Fig. 12a, b). Second, including the DN neurons ensured that there must be differential modulation of activity among the DP and DN neurons at the time the neural trajectory successfully crossed the target threshold, i.e., to reach the target threshold, the DP neurons were required to be more active than the DN neurons. Thus, the animal could not solve the task by uniformly applying an increase in activity level across all neurons in the imaging region, as might happen during states of high arousal, including but not limited to locomotion. Finally, we found no evidence that locomotion was correlated with performance ($r = 0.139$, $p = 0.432$; Supplementary Fig. 12c, d).

To determine whether increased sensitivity to auditory cues contributed to improved performance, in three of the four mice we examined whether V1 neurons gained sensitivity to the auditory pitches used in the BCI paradigm. V1 neurons can be modulated by auditory stimuli that co-occur with visual stimuli; however, in the absence of training, it is well established that auditory stimuli do not drive V1 neurons to spike threshold[35–40]. We found that after BCI learning, V1 neurons were not significantly modulated by any of the six auditory pitches used in the training paradigm (Supplementary Fig. 13). Thus, it is unlikely that BCI performance was improved due to an increased sensitivity to the auditory pitches used here.

**BCI training induced changes in the majority of V1 indirect neurons.** At the time of any given threshold crossing, it is likely that some indirect neurons were also active. To determine the extent to which individual indirect neurons were consistently co-active with one of the DP neurons at the time of success, we measured the correlation of indirect neurons with direct neurons across all trials within a given session. Out of the 653 indirect neurons imaged in BCI session 1 and the last BCI session, 76% were significantly correlated with at least one DP neuron on BCI

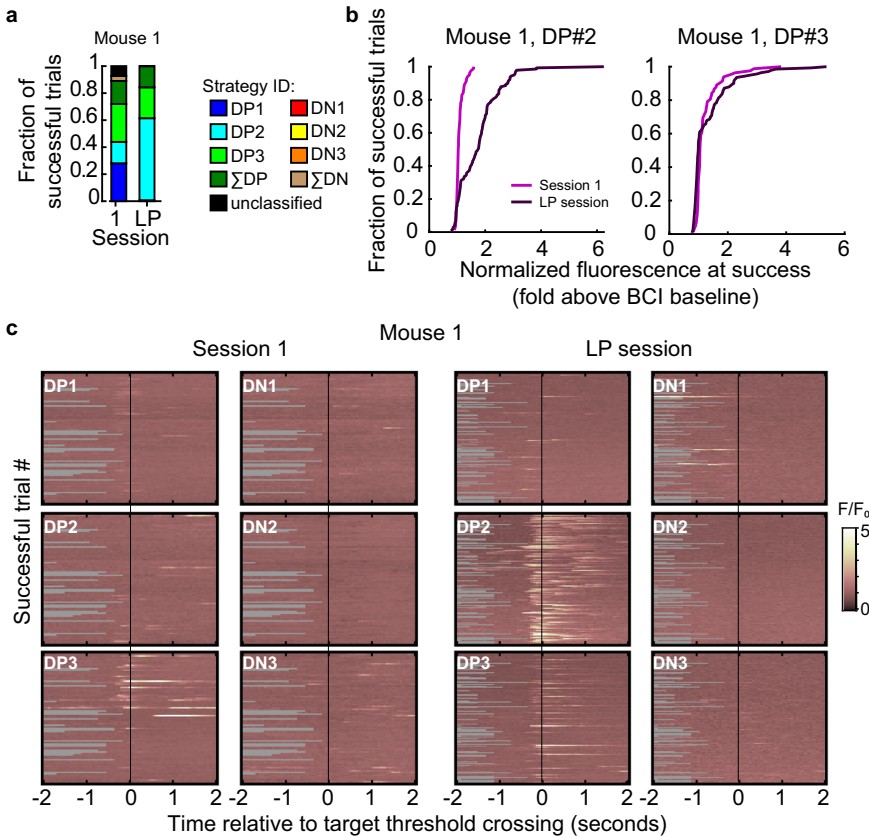

**Fig. 2 Contribution of direct neurons to threshold crossing varied between session 1 and the LP session. a** Neural activity of the direct neurons at the time of success was categorized into eight strategies. The eight strategies accounted for 94.7% of all successful trials across the four mice (Figs. S8–11). Trial strategies were defined as follows: a single DP or DN neuron contributed to 70% or more of the control signal generated at the time of success, or either the summed activity of the 3 DP or the 3 DN neurons coordinated to contribute to 70% or more of the control signal at the time of success. In a minority of cases the strategy did not fall into one of these categories, such trials are referred to as unclassified. **b** The normalized real-time fluorescence at the time of success was significantly higher (K–S test) in the LP session ($n = 150$ trials) compared to session 1 ($n = 216$ trials) in DP neurons #2 and #3 (DP neuron #2, $p = 2.26\mathrm{E}{-}16$; DP neuron #3, $p = 1.92\mathrm{E}{-}05$) and lower in DP neuron #1 ($p = 5.31\mathrm{E}$, Supplementary Table 2) in mouse #1. See Supplementary Table 2 for the change in mean values in all four mice. **c** Activity of the 6 direct neurons during successful trials, aligned to the time of success, in session 1 and the LP session for mouse #1. Some trials were shorter than 2 s (gray background). Note, DP neuron #2 was the dominant contributor to threshold crossings in the LP session. Source data are provided as a Source Data file.

session 1. Similarly, out of the 675 neurons imaged on the last session, 69% were significantly correlated with at least one DP neuron on the last BCI session (Fig. 3a). We noted that there was no correlation between the distance and strength of correlation among indirect-direct neurons pairs (Supplementary Fig. 14).

Given that a substantial fraction of indirect neurons were correlated with at least one DP neuron, and this relationship was maintained throughout training, we next asked whether indirect neurons might exhibit training-induced plasticity. We found that on the last BCI session, the majority of tracked V1 neurons exhibited a change in activity at the time of target threshold crossing. Seventy-nine percent of the indirect neurons tracked in both sessions, pooled across all 4 animals, had a significant change in their level of activity at the time of success (Wilcoxon rank-sum test, $p < 0.05$). The majority of these neurons (77%) increased their activity, and the remaining 23% neurons decreased their activity (Fig. 3b). Taken together, the direct and indirect neuron analysis demonstrates that BCI training induced modifications within V1 such that the pattern of network activity produced at the time of success was distinct between the naive and trained state, similar to observations made in motor cortex[41].

These within-task changes were associated with a change in noise correlation outside of the BCI task context. Pairwise noise correlation among indirect V1 neurons significantly increased

after BCI learning (Wilcoxon rank-sum test, $p = 0.0015$; Fig. 3c, see Supplementary Table 3 for direct neuron values). This is an indication that BCI training drove an increase in functional connectivity among V1 neurons that persisted outside of the BCI task context. In contrast, there was no difference in noise correlation detected between the two baseline VS sessions, VS baseline 1 and VS baseline 2 (Wilcoxon rank-sum test, $p = 0.573$, $n = 885$ and 753 neuron pairs, respectively).

**BCI training did not enhance stimulus responses.** BCI-induced plasticity within V1 could influence visual function in three ways: vision could be enhanced, disrupted, or maintained. As an example of how BCI might enhance vision, first we considered whether visual responses in the subset of indirect neurons that were similarly tuned to the direct neurons prior to BCI training were increased. Pairs of neurons that share orientation preference are preferentially connected at the synaptic level and exhibit correlated activity[42–44], therefore it is possible that BCI training could induce Hebbian plasticity in indirect-direct neuron pairs. Indeed, other paradigms in which user-defined de novo correlations are introduced, Hebbian plasticity is evoked[45]. In our case, Hebbian plasticity would be revealed as a strengthening of responsiveness to the preferred visual stimulus of the direct neuron.

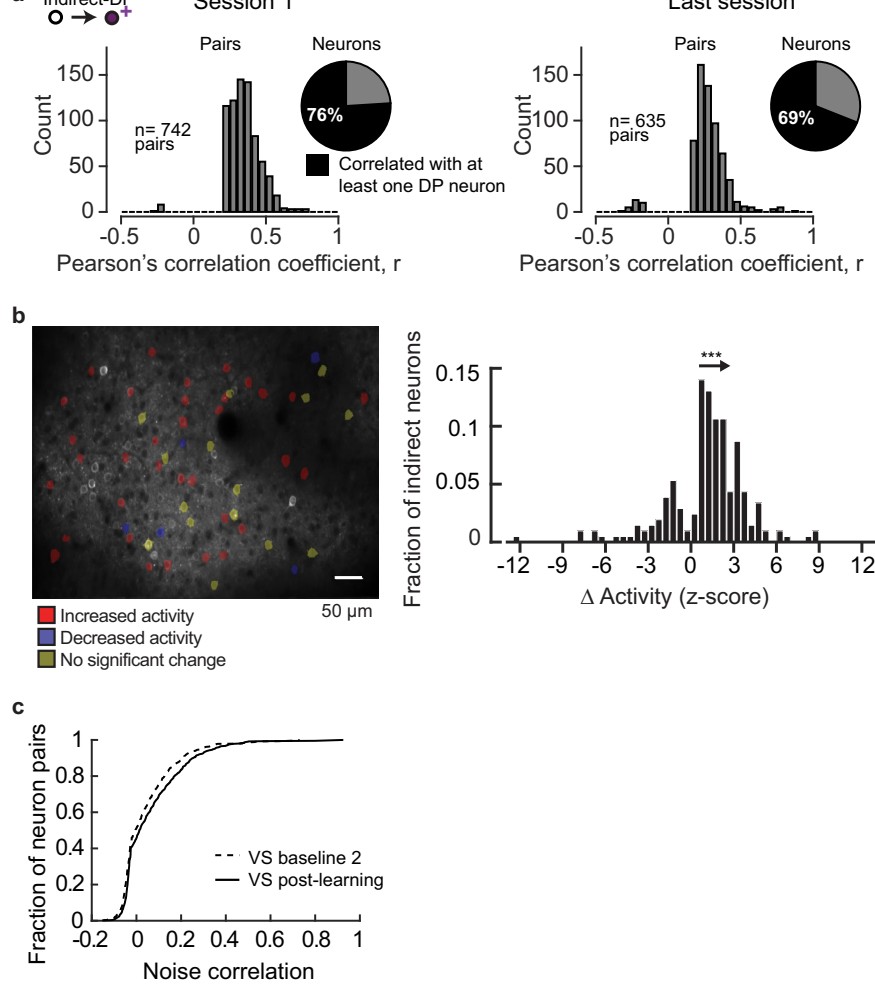

**Fig. 3 The majority of V1 indirect neurons exhibited plasticity during BCI training. a** A substantial number of indirect neurons were correlated with at least one DP neuron. Pairwise correlations were computed across all indirect-DP pairs across trials for a given session, at the time of success; pairs with a significant correlation (Pearson's correlation, $\alpha = 0.05$) are shown. The number of indirect neurons significantly correlated with at least one DP neuron is indicated (pie chart, session 1: 653 and last session: 675 neurons). **b** Left, imaging field of view of an example animal. Indirect neurons that were tracked in Session 1 and the last session are indicated by overlays (54 neurons). The direction of change in activity level at the time of success is indicated. Right, the distribution of the median change in activity amplitude for the neurons that exhibited a significant difference across trials (Wilcoxon rank-sum test, $\alpha = 0.05$, $n = 207$ neurons) between Session 1 and the last session. The distribution was skewed in the positive direction (one-sample K–S test, $p = 7.9E{-}38$(***), $n = 207$ neurons). **c** Pairwise noise correlation between V1 neurons, including both direct and indirect neurons, significantly increased after BCI training (Wilcoxon rank-sum test, $p = 0.0015$). The visual stimulation imaging session immediately preceding BCI training onset (VS baseline 2, $n = 599$ neuron pairs) and after BCI learning (VS post-learning, $n = 609$ neuron pairs) were compared. Trials with locomotion were removed prior to this analysis. Source data are provided as a Source Data file.

To assess whether this was the case, we examined response reliability in the subset of indirect neurons that initially were matched in stimulus preference to a direct neuron. Tuning to randomly presented static grating stimuli was assessed by fitting the deconvolved responses of neurons that were determined to be responsive (see Methods) to a two-dimensional Gaussian function[46] (Fig. 4a, b, Supplementary Fig. 15). The stimulus set consisted of 12 orientations and 15 spatial frequencies spanning a range of 0.02–0.30 cycles/°, resulting in a total of 180 stimuli. Out of the 292 neurons that were tracked in both VS episodes across the 4 BCI mice, 95 neurons, excluding direct neurons, were tuned to grating stimuli. Out of these 95 tuned neurons, 39 had similar orientation tuning preference to a DP neuron in the VS baseline 2 imaging session (Fig. 4c). BCI learning did not increase the trial-by-trial reliability of these 39 indirect neurons (Fig. 4d), nor was the response amplitude increased after removing failure trials for these same neurons (Fig. 4e). In addition, the fraction of tuned

neurons that were similarly tuned to a direct neuron was not significantly increased after BCI learning (paired $t$-test $p = 0.699$); prior to BCI training; $34 \pm 13\%$ ($\pm$S.E.M. across animals) neurons were similarly tuned to a direct neuron, and after BCI learning $35 \pm 12\%$ neurons were similarly tuned.

**Stimulus representation was not disrupted after learning the BCI task**. To determine whether BCI training perturbed stimulus representation, four key aspects of visual processing were assessed in both indirect and direct neurons: (1) stability of tuning, (2) pairwise correlation of neural activity during vision, (3) response amplitude, and (4) discriminability of visual features using decoding methods. Previously, we and others have characterized the stability of orientation and spatial frequency tuning. Despite high trial-to-trial variability in response amplitude in awake conditions, orientation and spatial frequency are stable across days[15,46,47]. The majority of tuned V1 neurons shift their

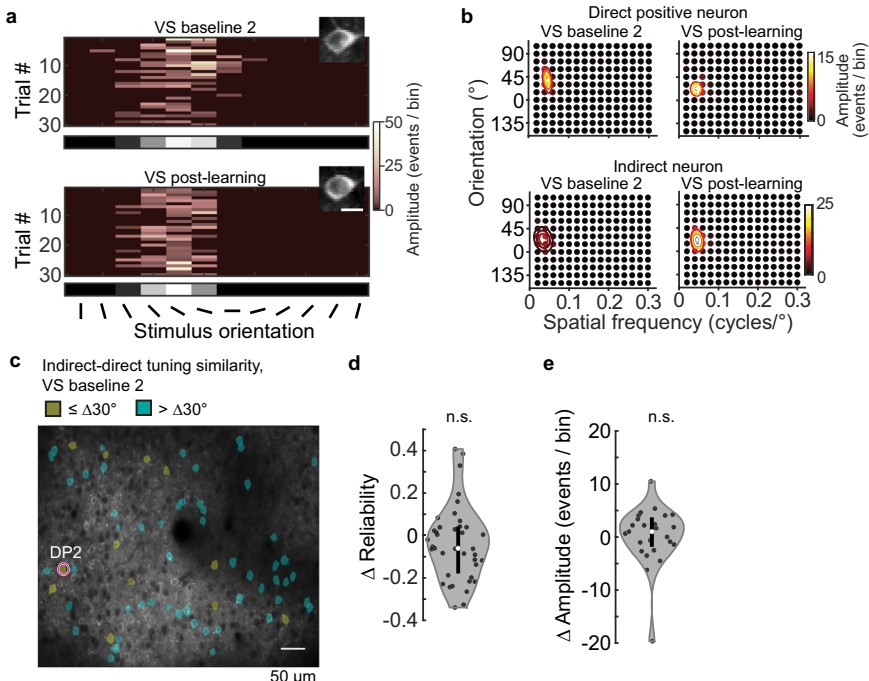

**Fig. 4 BCI training did not enhance stimulus responses in the subset of indirect neurons that matched their stimulus preference with one of the direct neurons. a** Example indirect neuron responses to 12 orientations at the preferred spatial frequency (0.04 cycles/°), before (VS baseline 2) and after BCI learning (VS post-learning). The mean across trials is shown in gray scale at the bottom (scale limits: 0–9 events). Scale bar: 10 μm. **b** Example 2-dimensional tuning profile of one DP neuron, and one indirect before and after BCI learning. Labeled contours of the Gaussian fit are 75%, 50%, and 25% of the peak response. **c** Spatial position of the subset of indirect neurons that were similarly tuned to DP#2 (red circle) in Mouse #2 before BCI training was initiated. Neurons were considered to be similarly tuned if their preferred orientation was within 30° and their spatial frequency within 0.03 cycles/° of a direct neuron, prior to BCI training. **d** The trial-by-trial response reliability to the preferred stimulus was the same before (VS baseline 2) and after BCI training (VS post-learning), the median Δ reliability = −0.06; Wilcoxon signed-rank test, p = 0.093, n = 39 neurons. The kernel density estimate (gray), median (white circle), and interquartile range of the distribution (black bar) are indicated. **e** Change in response amplitude at the preferred stimulus between VS baseline 2 and VS post-learning, same pool of indirect neurons as 'd'. The response amplitude (failure trials removed) was the same before and after BCI training, the median Δ amplitude = 1.02 events/bin; Wilcoxon signed-rank test, p = 0.382, n = 25 neurons. Source data are provided as a Source Data file.

preferred orientation five degrees or less over the course of 2 weeks, and similarly, preferred spatial frequency changes less than 0.006 cycles/° over the same period[46].

First we assessed the stability of tuning in response to grating stimuli before and after BCI learning. Stability in control conditions was assessed by characterizing the changes in tuning between VS baseline 1 and VS baseline 2. To assess stability after learning, referred to here as the BCI condition, changes in tuning were characterized between the VS baseline 2 and VS post-learning imaging sessions. 353 neurons were tracked on VS baseline 1 and VS baseline 2, 292 neurons were tracked on VS baseline 2 and VS post-learning (Fig. 5a, b). The two pools of tracked neurons include both indirect and direct neurons. The stability of orientation and spatial frequency preference, as well as bandwidth for both of these features was compared between baseline and BCI conditions for tuned neurons (i.e., well-fit by the two-dimensional Gaussian function; Fig. 5c–d, Supplementary Fig. 16). VS baseline 2 was collected within 4 days of the completion of the auditory Go/No-Go task prior to the initiation of BCI training, and VS post-learning was collected after the LP in all four mice. This design ensured that the number of days spanning the baseline and BCI condition were matched within subjects and accounted for individual differences in the learning rate of both the auditory pitch association task as well as the BCI task. The median change in orientation preference during the baseline condition was 4.6° (interquartile range = 2.0–12.5°). Similarly, the median change was 4.6° (interquartile range = 1.8–10.2° in the BCI condition (Fig. 5c).

Thus, the distribution of changes in orientation preference in the BCI condition was indistinguishable from that of the baseline condition. Analysis of the distributions on an animal-by-animal basis confirmed that orientation preference was not destabilized by BCI learning (Fig. 5d). This was the case for the other parameters as well (Supplementary Fig. 16, see Supplementary Table 4 for direct neuron values). We noted that the median change in orientation preference during the initial baseline condition was slightly higher than previously reported. Considering that the time span was slightly longer than the previously published work[46], this was expected.

The stability of individual tuning features within a given neuron is independently regulated[46]. For example, a neuron with stable orientation preference could exhibit drift in spatial frequency preference. Therefore, although when assessed separately individual tuning parameters appeared stable across the population of V1 neurons during BCI acquisition, it is possible that if all parameters, including orientation preference, orientation bandwidth, spatial frequent preference, and spatial frequency bandwidth, were considered simultaneously, neurons would appear more unstable. To address this possibility, we examined pairwise signal correlation among the neurons that were responsive and tuned. Signal correlation is the correlation of the average responses to the presented stimuli between a pair of neurons; as such, it is a measurement that is independent of the Gaussian model of tuning and is non-parametric. To assess the stability of signal correlation, we computed the change in signal

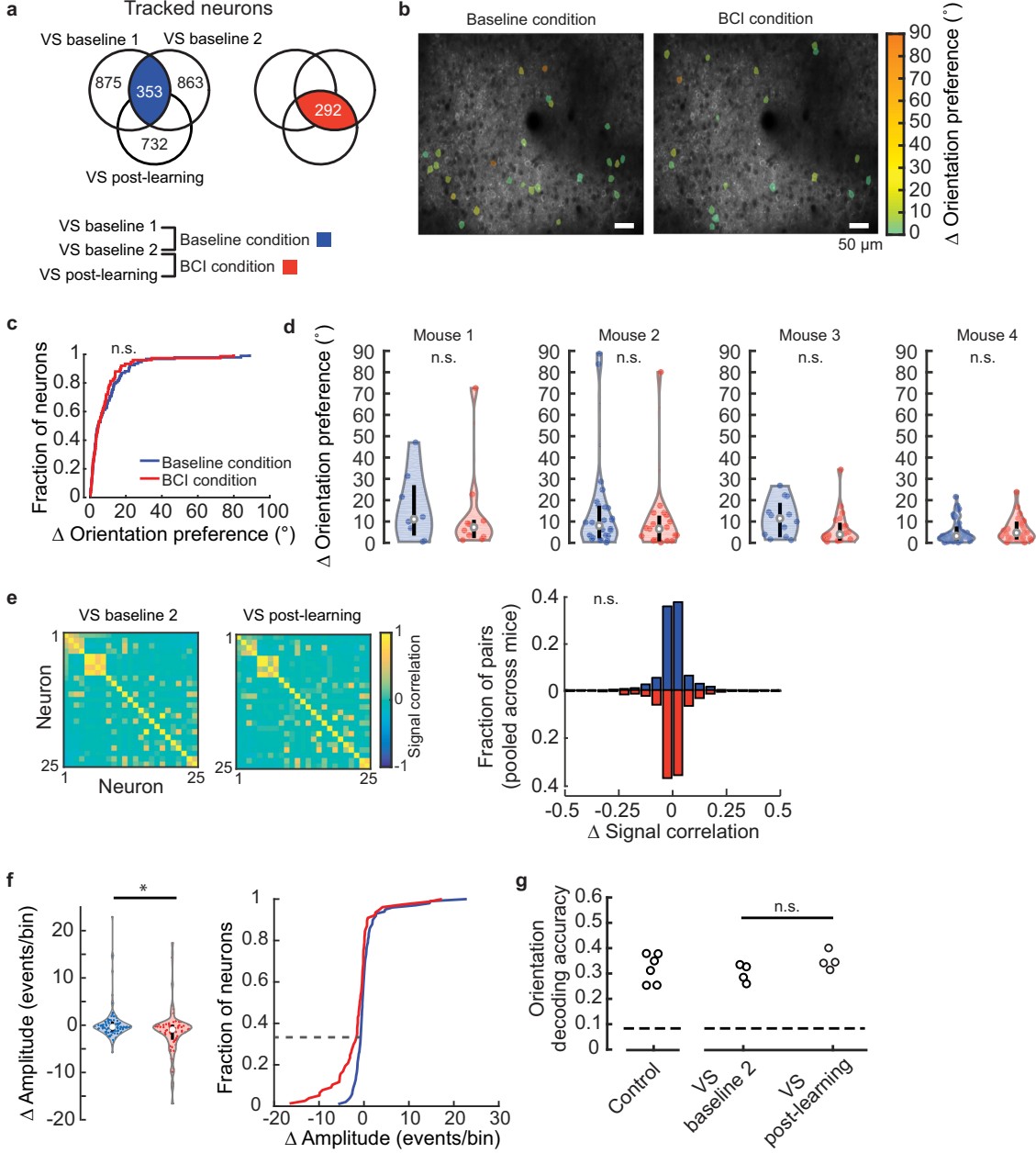

**Fig. 5 Acquisition of BCI control did not disrupt stimulus representation in V1. a** Venn diagrams depicting the total number of imaged neurons in each of the thee imaging sessions. The overlap represents the number of tracked neurons in the baseline (left) and BCI conditions (right). **b** Imaging field of view of an example animal. The change in orientation preference of the tracked and tuned neurons for both conditions indicated by overlays. Note that the tracked pools, although contain some of the same neurons, were different. **c** The distribution of change (Δ) in orientation preference in the baseline condition was similar to that of the BCI condition (K–S test, $p = 0.680$, $n = 93$ neurons baseline, 75 neurons BCI). **d** Stability of orientation preference in the baseline condition and BCI condition. Labels as in Fig. 3d. Distributions were overlapping in all four mice (Wilcoxon rank-sum test, $p$ values for mouse #1–4 respectively: $p = 0.633$, $p = 0.308$, $p = 0.089$, $p = 0.338$; number of neurons in the baseline and BCI conditions for mouse #1–4: $n = 8$, 10; $n = 26$, 19; $n = 12$, 21; $n = 47$, 25). **e** Left, example signal correlation matrices, for the BCI condition. Right, stability of signal correlation was similar between baseline and BCI conditions (Wilcoxon rank-sum test $p = 0.538$, $n = 1500$ pairs baseline, 726 pairs BCI). **f** Left, the change (Δ) in response amplitude for the stimulus that elicited the maximum response in the baseline and BCI conditions was different (Wilcoxon rank-sum test, $p = 0.002(^{**})$, $n = 99$ neurons baseline, 77 neurons BCI. Failure trials were included; data were pooled across the four mice. Right, approximately one-third of the population in the BCI condition had a greater decrease in amplitude compared to the baseline condition (dashed line). **g** Decoding accuracy of orientation at a spatial frequency of 0.04 cycles/° was similar before (VS baseline 2) and after BCI training (VS post-learning), Wilcoxon signed-rank test, $p = 0.375$, $n = 4$ animals. The number of neurons used to classify orientation was matched across condition for each animal (1–4, respectively): 69,33,57,47. A second control group (5 animals) is plotted on the left, 69 neurons were used. Chance probability was 0.083 (dashed line). Source data are provided as a Source Data file.

correlation between baseline and BCI conditions. The distribution of changes in signal correlation was not statistically different between the two conditions (Fig. 5e, see Supplementary Table 5

for direct neuron values). Both distributions were centered on zero; the standard deviation of the baseline condition was 0.103, and the standard deviation of the BCI condition was 0.111.

Next we considered whether BCI training may have altered the strength of responsiveness. Changes in the response amplitude for all responsive neurons, including neurons that were not well-fit by the Gaussian function, in other words not necessarily tuned by classic metrics, were considered[48]. The average amplitude of the stimulus response that elicited the maximum response was compared between the two conditions. The median change in amplitude was −0.35 events/bin in the baseline condition, and was slightly lower in the BCI condition, −0.92 events/bin (Fig. 5f see Supplementary Table 6 for direct neuron values). The distributions of change in response amplitude in the baseline and BCI conditions were largely overlapping, however, we did detect a significant difference between the two conditions (Wilcoxon rank-sum test $p = 0.002$). Approximately one-third of the neurons in the BCI condition had a greater decrease in amplitude compared to the baseline condition (Fig. 5f, right).

Given that a small but significant change in response amplitude, as well as a persistent change in noise correlation were observed outside of the trained context, it is possible that BCI training could have modified the amount of visual information contained within V1. To address this possibility, we utilized decoding methods to estimate the amount of information encoded by a population of V1 neurons. We used a k-nearest neighbor (KNN) classifier to decode stimulus orientation at a spatial frequency of 0.04 cycles/°, the preferred spatial frequency of the majority of V1 neurons[49]. We found that decoding accuracy was not different after BCI acquisition compared to before BCI training in the same mice (Fig. 5g). In addition, decoding accuracy after BCI acquisition was indistinguishable from that of six control mice that never experienced BCI training. These results indicate that despite a BCI-induced change in pairwise noise correlation, the amount of information encoded by the V1 population was not altered by BCI learning.

**Stimulus information was maintained after learning a visual discrimination task, but population coding was disrupted.** The BCI task used in this study was multimodal in nature. To determine whether our results generalize to a single-modality task, we trained mice in a visual discrimination task in which it was previously established that V1 activity is required for improved behavioral performance[16,50,51]. Converging evidence indicates visual discrimination training enhances the neural representation of rewarded stimuli by increasing selectivity for the stimuli experienced during training[16,21] and in some cases improving response reliability[16], and at the same time suppresses responses to non-relevant stimuli[16]. Enhanced responses to rewarded stimuli are known to generalize across task variations experienced in the training environment. However, selectivity for features such as orientation dissipates when reward contingencies are recognizably altered. As such, reward-induced changes in selectivity are considered to be context-specific. Furthermore, in many instances long-lasting changes observed in the training environment are restricted to stimulus-specific assemblies, and enhancement to more than one rewarded stimulus is possible due to assembly-specific plasticity[20,21]. Based on these previous observations, we hypothesized that similar to our non-matched sensory task, reward training-induced changes are context specific, and tuning to visual stimuli remains largely undisturbed outside of the training environment. Conceptually, this would be consistent with the idea that reward-induced enhancement of responses is transient and must be actively recruited during cued episodes, such as would occur in the training environment when the reward delivery device is present.

Six mice were trained to discriminate two visual patterns while running past a virtual wall, where one pattern was rewarded.

Locomotion was directly coupled to changes in visual stimuli presented on a screen such that the animals' running controlled their position along a wall (Fig. 6a, Supplementary Movie 6). After running past a virtual approach wall composed of black and white circles overlaid on a gray background, mice were abruptly presented with either a vertical (0°) or angled (135°) grating. Mice received a water reward for licking in response to the vertical grating. Trials were scored as a hit if a lick was detected when the animal was in the in the reward zone. No punishment was given for licking in response to the angled grating (false alarm trial). Mice learned to lick preferentially on the vertical grating within 10 days. Performance on the visual discrimination task was quantified by calculating the behavioral d-prime for each training session, which is a measure of the difference in the proportion of hit and false alarm trials (Fig. 6b). For the first 14 days, mice were trained every day, thereafter the training sessions occurred every 2–3 days. One day prior to visual discrimination training, a baseline VS imaging session was acquired, a second VS imaging session was acquired 15 days after the first training session (15-days training), and a third VS imaging session was acquired 30 days after the first training session (30-days training). As in the BCI task, the reward delivery device was not visible during the passive viewing VS imaging sessions.

Tuning stability between the baseline and 30-day training VS sessions was computed for orientation preference and bandwidth, as well as spatial frequency preference and bandwidth. All neurons that were tracked and tuned on both sessions were included in the analysis. The median change in orientation preference, pooled across mice, was $5.7 \pm 0.9°$. Mice trained in the visual discrimination task were compared to a separate cohort of control mice that did not receive visual discrimination training (see Supplementary Table 1 for details). No difference in stability in any of the 4 tuning parameters (orientation preference, orientation bandwidth, spatial frequency preference, and spatial frequency bandwidth) was detected (Fig. 6c, Supplementary Fig. 17a, b).

However, when considering all responsive neurons (tuned and untuned) for a given session, there was a significant change in the distribution of orientation preference. Under baseline conditions, cardinal orientations are over-represented[52–54] (Fig. 6d). We found that the preference for 90° orientated grating stimuli decreased after 15 days of visual discrimination training. With extended training the decreased preference for 90° orientated stimuli was maintained, and trended toward being enhanced (Fig. 6d). The shift in preference was due to a loss of responsiveness to the 90° stimulus, rather than an increase in responsiveness to the trained stimuli (Supplementary Fig. 17c). A suppression of responses to the non-trained stimulus outside of the task context may be related to the observation that non-relevant stimuli are suppressed when assayed within the training environment[21], and raises the possibility that changes during training are consolidated.

Next we assessed whether the training-induced shift in orientation preference was associated with a loss of information encoded in V1. A KNN classifier was used to decode stimulus orientation at a spatial frequency of 0.04 cycles/°, for the neural activity imaged on the baseline and 30-day training imaging sessions. No difference in decoding accuracy was detected (Fig. 6e, f). Closer examination of the confusion matrices in which the accuracy of each of the 12 stimulus orientations can be considered separately, revealed that the decoding accuracy of the 90° stimulus was not impacted (paired t-test, $p = 0.54$, $n = 6$ animals), despite the shift in preference shown in Fig. 6d. This is an indication that there is sufficient redundancy in the representation of orientation that small changes in the preference distribution did not degrade the information content encoded in V1. Nor was discriminability of either of the two

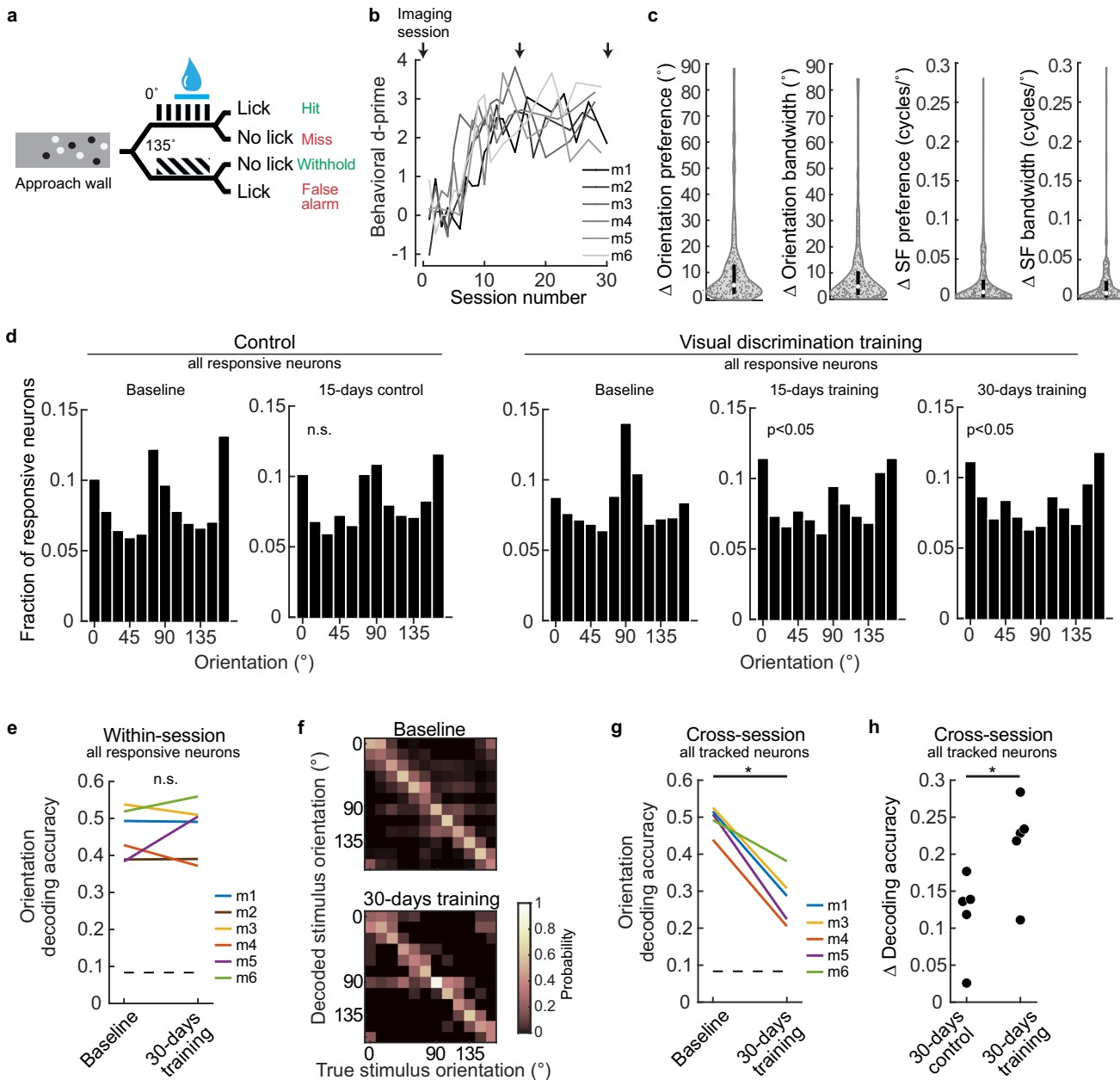

**Fig. 6 Visual discrimination impacted the stability but not the accuracy of stimulus representation. a** Schematic of visual discrimination task. **b** Six of six mice (m) learned the visual discrimination task. All mice achieved a behavioral d-prime above 1.5 within 10 days. **c** Tuning stability of all tracked, visual responsive and tuned neurons ($n = 320$ neurons), pooled across the 5 of 6 mice. Neurons were required to be tuned on both imaging sessions (baseline and 30-days training) to be included. **d** Distribution of orientation preference for all visually responsive neurons in two cohorts of mice. Six naive mice were imaged 15 days apart. The distributions were not different (Chi-square test, $p = 0.863$; $n = 1178$ and 689 neurons). The six mice in '**b**' were imaged before, during, and after training. The distributions after 15 and 30 days of training were significantly different from baseline (Chi-square test, $p < 0.001$ both conditions; baseline, 15-days, 30 days: $n = 1311, 804,$ and 762 neurons). Data were pooled across animals. **e** Decoding accuracy of orientation (spatial frequency of 0.04 cycles/°) was similar before and after training (Wilcoxon signed-rank test, $p = 0.999$, $n = 6$ animals). The number of neurons used to classify orientation was matched across conditions for each animal: 167,144,122,150,101,110. Chance probability was 0.083 (dashed line). **f** Example confusion matrices before and after visual discrimination training, mouse #1. **g** Cross-session classification of orientation at a spatial frequency of 0.04 cycles/° after visual discrimination training was significantly lower compared to baseline (paired $t$-test, $p = 1.6E−3$, $n = 5$ animals). All tracked neurons were included, including those not visually responsive (number of neurons for each mouse, respectively: 272, 292, 329, 264, 248). Mouse IDs as in '**e**'. Mouse #2 was not included due to a neuron tracking error. Chance probability was 0.083 (dashed line). **\*\***$p < 0.001$. **h** The difference in cross-session decoding accuracy of orientation at a spatial frequency of 0.04 cycles/° was significantly lower ($t$-test, $p = 0.035$, $n = 5$ animals in each condition) after visual discrimination training (data from '**g**') compared to control animals (number of neurons for control mice: 133, 64, 49, 191, 232). *$p < 0.05$. Source data are provided as a Source Data file.

orientations experienced during training enhanced outside of the training environment (paired *t*-test, 135°: $p = 0.69$, 0° = 0.72, $n = 6$ animals). Thus, consistent with the hypotheses stated above, and similar to the non-sensory matched BCI task, stimulus discriminability at the level of V1 was maintained after learning the visual task; we found no evidence of either enhancement or degradation outside of the task context.

However, given that a shift in orientation preference was observed across the population, the manner in which stimuli were encoded may have been altered. To address this possibility, the rate of representational drift was assessed in control conditions, and after discrimination training. KNN classifiers were trained on neural activity recorded on one imaging session, and tested using the neural activity from the same tracked neurons acquired on a second imaging session, 30 days later (cross-session, fixed classifier[55]). When using the fixed classifier, cross-session decoding accuracy significantly decreased after 30 days of visual discrimination training, and the magnitude of decrease was significantly greater than control conditions (Fig. 6g, h). Thus, although the preferential loss of responsiveness to 90°-oriented stimuli did not impact the accuracy of decoding per se, when all responsive neurons were available to train the classifier, the contribution of individual neurons appears to have been altered by visual discrimination training.

To address whether the rate representational drift was similarly increased by BCI training, KNN fixed classifiers were used to assess decoding accuracy after BCI training. In contrast to visual discrimination training, the rate of drift did not appear to be impacted. The drift values of the four mice trained in the BCI task were within range of the control conditions shown in Fig. 6h, and qualitatively lower than the drift assessed after visual discrimination training (BCI mice 1–4, Δ accuracy, respectively: 0.14, 0.02, 0.1, and 0.05; mean: 0.076). Thus, in this regard, the non-matched sensory task was less disruptive to stimulus encoding compared to the visual discrimination task.

## Discussion

Theoretically, learning new skills could result in changes to neural activity that minimally impact previously acquired behaviors and function[56,57]. For example, synaptic plasticity could be restricted to updating activity patterns that occupy the null space of skilled action plans[58]. Experimental evidence establishing that new skills can be acquired without altering the neural activity that underlies previously developed computations and function is an area of active investigation[6,7]. Furthermore, learning in artificial network simulations often leads to catastrophic forgetting[59,60], raising the possibility that networks cannot integrate new information without disrupting previously learned functions[61]. Here we directly demonstrated that integration of a new, non-matched sensory skill does not perturb previously developed function. We were able to address this issue by designing a learning task that associated a user-defined pattern of activity with a rewarding outcome, in combination with tracking the activity of the same neurons across skill learning. The original functional properties of the network were assessed before and after the new skill was acquired.

In principle there are thee ways in which vision would not be disrupted by BCI training. One, a small number of neurons in V1 change their activity during task performance. In this scenario, it would be highly unlikely that training would in any way influence the cortex's ability to process visual stimuli. We can rule out this possibility because we found that ~80% of the neurons in V1 exhibited plasticity during BCI. Two, a substantial number of V1 neurons change their activity during BCI, and these changes persist outside of the BCI context and impact visual stimulus

tuning. However, at the population level the changes are in the null space of the downstream readout. For example, the neurons that changed do not project to V2, and there is sufficient redundancy such that it is the non-relevant neurons that changed. This second possibility can be ruled out because we did not observe an impact of training on visual stimulus tuning. Three, a large number of neurons show plasticity during BCI, but tuning to visual stimuli in V1 remains stable. In other words, the changes are specific to the BCI context. This is the outcome that we observed. An implication of our findings is that the ability to distinguish contexts may be fundamental to the maintenance of stable perception while acquiring new skills.

Our results have implications for the clinical use of BCIs. Many quadriplegic patients, including those engaged in ongoing intra-cortical BCI clinical trials, have residual sensory and/or motor function[62–64]. For clinical use, microelectrode arrays are often implanted into both the primary motor (M1) and primary somatosensory (S1) cortices of those patients, to enable bi-directional communication of motor and sensory function. Intriguingly, motor actions can be decoded out of S1[65–67]. A reasonable question is whether leveraging the information in S1 for motor decoding with a BCI would degrade residual sensory function. Our work suggests that it would not.

Similar to BCI training, a substantial fraction of V1 neurons retained their tuning preference after visual discrimination training, when probed outside of the training environment. It was previously observed that during a similar implementation of the visual discrimination task used here, activity during task execution was slightly depressed[50], selectivity for the stimuli used during training increased in a context-dependent manner[16], and selective suppression dominated when the animals were engaged in the task[51]. Our results are consistent with these previous observations, considering our measurements were made outside of the training environment and therefore can also be considered a different context. In addition, we show that preferential suppression of responses to non-trained stimuli are observed outside of the task context shortly after learning, and that this feature-specific suppression is maintained throughout extended training. Despite the persistent impact on vision outside of the task context, similar to non-matched sensory BCI learning, reliable decoding of visual stimulus features was intact. This is an indication that there is sufficient redundancy such that information is not lost in the case a fraction of neurons is repurposed or their contribution to stimulus representation is altered. However, cross-session fixed classifiers degraded after visual discrimination learning. Whether this degradation has an impact on the ability of downstream regions in the visual hierarchy to read out stimulus information contained in V1 will depend on the extent to which these changes are orthogonal to the latent space through which information is transferred. For example, low-dimensional latent dynamics transferred through the space could remain stable[55] during learning. Alternatively, adaptation could be coordinated across the hierarchy. In future studies it will be of interest to examine whether recurrent connectivity between V1 and higher visual areas, such as LM[68–70], is a substrate for low-dimensional latent dynamics to rapidly adjust during learning and thereby facilitate stable perception.

## Methods

**Animal preparation**. All experimental procedures were compliant with the guidelines established by the Institutional Animal Care and Use Committee of Carnegie Mellon University and the National Institutes of Health, and all experimental protocols were approved by the Institutional Animal Care and Use Committee of Carnegie Mellon University (protocol # PROTO201600014). To express

the calcium indicator GCaMP6f selectively in excitatory neurons, either homozygous Emx1cre mice (Jackson Laboratories, stock number 005628) or homozygous SLC17a7cre mice (Jackson Laboratories, stock number 023527) were crossed with homozygous Ai93/heterozygous Camk2a-tTA mice (Jackson Laboratories, stock number 024108). Experimental mice were heterozygous for all three alleles. Mice were housed in groups of 2–3 per cage, in a 12 h light/12 h dark cycle; all imaging sessions started at Zeitgeber time (ZT) 14.5 ± 1, where ZT0 is lights on, and ZT12 is lights off. The same enrichment materials were provided in all cages including a Plexiglas hut and nesting material. See Supplementary Table 1 for information on animal sex and genotype. None of the mice used in this study exhibited aberrant, interictal events[71,72] in V1 or adjacent regions. The ambient temperature range was, 68–75 °F, and the humidity range was 18–65%.

Mice (29–39 days old) were anesthetized with isoflurane (3% induction, 1–2% maintenance). A 3 mm diameter craniotomy was made over the primary visual cortex in the left hemisphere, see also[73]. A stainless-steel bar, used to immobilize the head for recordings, was glued to the right side of the skull and secured with dental cement. The craniotomy was then covered with a double glass assembly in which the diameter of the inner glass was fitted to the craniotomy and sealed with dental cement. Mice were allowed to recover for a minimum of 3 days with ad libitum access to food and water.

**Data acquisition, neuron segmentation, and neuron tracking**. Two-photon calcium imaging was performed in awake head-fixed mice mounted atop a floating spherical treadmill or a single-axel foam wheel using a resonant scanning microscope (Neurolabware) outfitted with a 16× Nikon objective (0.80 NA) and 8 kHz resonant scanning mirror. Treadmill motion was recorded using a camera (Dalsa Genie M640-1/3) for off-line analysis of locomotion, wheel motion was recorded with an optical quadrature rotary encoder (US Digital, Vancouver, WA) coupled to an Arduino Mega, and eye blinks were captured using a second camera (Dalsa Genie M1280)[72]. During BCI sessions, a separate tracking laser sensor (Keyence LV-N11MN) was used to sample the motion of the treadmill at a rapid sampling rate of 194 ms[74], in order to restrict BCI trials from starting if the mouse was moving. The sensor was calibrated to detect locomotion >7 cm/s. We noted that locomotion speed exceeded this value in most bouts of activity (e.g., Supplementary Fig. 12d and Supplementary Movie 5).

A laser excitation wavelength of 920 nm was used (Coherent, Inc.); green emissions were filtered (Semrock 510/84-50), amplified (Edmund Optics 59–179), and detected with a PMT (Hamamatsu H1 0770B-40). The imaged field of view was 620 × 504 microns, pixel dimensions were 0.85 × 0.98 μm, and the acquisition rate was 15.5 Hz. The acquired image time series were motion-corrected by computing the horizontal and vertical translation of each frame using phase correlation[72], and individual neurons segmented using the Matlab version of Suite2p toolbox[75], see also ref. [72].

To identify neurons that were tracked across imaging sessions, we registered repeat imaging sessions using the mean intensity image of each session. The mean intensity image for a session was computed by averaging the intensity of each pixel in the aligned calcium image series across time for the entire imaging session (roughly 50,000 frames). Then, the mean intensity images of the two sessions were registered using an affine transform with one-plus-one evolutionary optimizer. Once the sessions were registered, the percentage of pixel overlap between the neurons from two sessions was computed. Neurons were accepted to be the same neuron across sessions, if the percentage of overlapping pixels across the two sessions was larger than 75%. On average, there were 160 pixels in a given neuron. The median proportion of neurons that were tracked from the available neurons was 40%.

**Auditory Go/No-go pitch association task**. Mice learned to discriminate between a 15 kHz 'Go' stimulus and a 5 kHz 'No-go' stimulus prior to BCI training. After recovery from surgery, mice were placed on water restriction. During water restriction, ad libitum access to water was removed. Each day, mice were provided with 750 μL of water placed in a dish, and the dish was available for up to 30 min. The weights of the mice were closely monitored, and weight loss was observed during the early days of water restriction. Training started when the weight of the mice stabilized. We considered the weight of mice to be stabilized when the weight of the mice reached 80% of the weight at the start of water restriction, and the day-to-day change in weight was <0.1 grams for a minimum of 3 days.

Mice were head-fixed atop a spherical treadmill, with a custom-built lick port positioned near their mouth. The auditory stimuli were generated in Matlab (Mathworks, Inc.) by generating a 200 ms sinusoidal waveform of either 5 kHz or 15 kHz frequency with a Gaussian mask, thus creating a 200 ms pulse. The sampling rate of a pulse was 44.1 kHz. The generated pulses were repeated during the duration of a trial at 5 Hz. The stimulus was presented through a speaker positioned 50 degrees to the left of the mouse with respect to the midline at a distance of 30 cm. Each trial started when the stimulus sounded through the speaker. Lick port was armed with reward after a 200 ms sensory delay period only on Go trials. Trials lasted between 2 and 4 s, which remained fixed within a given session. The stimulus lasted for the duration of the trial, and the reward was available for retrieval while the Go stimulus was on. A single duration was used per experimental session. Licking was detected by a

photodiode sensor positioned at the end of the lick port tube. One reward was delivered per trial, and each reward was about 8 μL in size. Trials in which animals correctly licked during Go trials and withheld licks during No-go trials were counted as successful trials. When a mouse licked on a No-go trial (false alarm trials), a timeout was enforced. During the timeout, there was no auditory stimulus, and the animal had to withhold the lick for a minimum duration selected by the experimenter to exit the timeout period. Each time the animal licked during a timeout, the timeout timer was restarted. Upon exiting the timeout period, the mice immediately entered the next trial. The range of timeout used was between 3 and 8 s. Performance in each session was quantified as the maximum percentage of successful trials in 100 consecutive trials. Mice were trained 1 session per day and each session was 30–60 min long, during which the mice completed 200–500 total trials. After reaching 80% performance, mice were advanced to BCI training.

**Brain computer interface task**. Mice were trained on a brain-computer interface task in which they could earn rewards by modulating the activity of six neurons to control an auditory pitch. We refer to these neurons as the direct neurons. The six neurons were randomly selected from a pool of neurons that were tracked and well isolated from other neurons, as such not all neurons were visually responsive to grating stimuli. The neural activity of the direct neurons was transformed into an auditory pitch using the following method: At the beginning of each BCI session, we measured the spontaneous activity of the direct neurons for 3 min in the dark (3000 frames). The mean and standard deviation of this BCI baseline activity was then used to normalize the real-time raw activity into Z-scores. During BCI control, the real-time Z-scored activity was binned by three frames (194 ms). Then, the activities of the six neurons were transformed into a one-dimensional control signal. The control signal at every time bin t was calculated according to the equation below [Eq. 1].

$$\text{Control signal} = \sum_{i=1}^{3} Z_i - \sum_{j=4}^{6} Z_j \tag{1}$$

In the equation above, $Z$ is the Z-scored activity of a given neuron and i and j is the index of the neuron. We define neurons that positively contribute to the control signal as direct positive neurons (DPs) and neurons that negatively contribute to the control signal as direct negative neurons (DNs). Control signal values were mapped to discrete pitches as follows: control values < 20% of the target threshold (defined separately for each mouse below) were mapped to 5 kHz, values between 20 and 40% were mapped to 6.23 kHz; values between 40 and 60% were mapped to 7.76 kHz; values between 60 and 80% were mapped to 9.67 kHz; values between 80 and 100% were mapped to 12.04 kHz; and values greater than or equal to the target threshold were mapped to 15.00 kHz. Once BCI training was initiated, no more than 2–3 consecutive days occurred between BCI training sessions. For three sessions after the LP session, failure trials (trials that lasted 10 s) were not properly saved due to a real-time synchronization issues, performance was not reported for those sessions (Fig. 1d, mouse #2 session 10 and 14, mouse #3 session 11).

*Selection of direct neurons and target threshold.* Prior to BCI training, we recorded a 10 min baseline activity session, in the dark, without auditory stimuli or reward. Selection of direct neurons and threshold was then determined from this baseline recording. We first selected a pool of 3 DP and 3 DN neurons randomly from the segmented population. We then ran an algorithm to determine the threshold for those neurons. That algorithm proceeded as follows. First, we only considered thresholds in the range of 6–15 (Z-score normalized), in an attempt to ensure we were neither too close to the noise floor nor potentially including large, outlier transients. Second, we searched for thresholds that would result in somewhere between 12 and 18 threshold crossings within 10 min. This corresponds to our target non-learning success rate of 20–30%. We initialized our search at a threshold of 10, and adjusted the threshold in step size of 1 to explore the space to determine the threshold value that resulted in the desired success rate. If 10 resulted in the right number of threshold crossings, we stopped, but if it was too many we increased the threshold by 1 and if too few we decreased the threshold by 1, continuing until we first found a threshold that resulted in the 12–18 threshold crossings or until we were out of range. If we were out of range, we selected another group of 6 neurons at random, and repeated. Once a threshold was found that resulted in a success rate of 20–30%, the search was stopped. On each iteration, all neurons were sampled form the entire available pool, including neurons that were selected from previous iterations.

*Trial structure.* Time of success was defined as the bin in which the auditory pitch reached 15 kHz. Mice had up to 10 s to reach the target once the trial started. Mice then had up to 4 s to retrieve the water reward by licking the lick port. Water was released only if the animal licked. Only a single drop (~8 μL) was delivered per success. Prior to trial start, we enforced an inter-trial-interval where the mice had to satisfy two conditions in order to start a trial. First, mice could not move. Second, the control signal had to be lower than the target threshold. When both of these conditions had been met for five consecutive time bins (1 s), the trial was started. The trial start was cued to the animal by the onset of auditory feedback. The target for each mouse was determined so that the mouse would have succeeded in about 30% of the trials during a 10 min recording of spontaneous activity in the

dark, recorded before mice were ever exposed to BCI training. In all BCI sessions within a mouse, the same six neurons were used, and the same target was used across the sessions.

*Assessment of baseline drift.* Neural activity was normalized by Z-scoring to allow for the assessment of performance across sessions. However, a decrease across sessions in the standard deviation of DP neuron raw fluorescence signal during the 3 min spontaneous baseline recording would lead to an inflation of DP neuron activity that could increase the frequency of threshold crossings. Likewise, an increase in the standard deviation of DN neuron raw fluorescence signal across sessions during the 3 min baseline could result in an increase in the frequency of threshold crossings. To examine whether this was an issue for any one of the four mice, we assessed the correlation between the standard deviation of the 3 min bassline and session number. None of the DP neurons had a significant negative correlation ($R < 0$ and $p < 0.05$) between standard deviation and session number, nor did any of the DN neurons have a significant positive correlation ($R > 0$ and $p < 0.05$) between standard deviation and session number (Supplementary Fig. 18a, Supplementary Table 7). Thus, drift across sessions in the standard deviation of baseline fluorescence did not contribute to increased threshold crossings. Within-session change in the standard deviation of the residual signal after calcium events were removed was similar for all sessions in each of the four mice (Supplementary Fig. 18b, Supplementary Table 7).

**Visual discrimination task.** Water restricted mice were trained on a self-paced, closed loop orientation discrimination task[16,50]. Wheel speed was coupled to a virtual wall presented on a single screen, positioned at a 50° angle with respect to the midline of the mouse. Prior to the discrimination task, mice were trained on a shaping task in which they were rewarded a drop of water for every 30 cm traversed while viewing a gray screen. When mice drank at least 80 drops within 45 min (on average 53.3 cm/min), they were transitioned into the discrimination task (typically 3–4 sessions).

Within the visual discrimination task, the motion of the visual stimulus was calibrated to the locomotion of the mice so that 1 cm locomotion on the wheel caused a 1 cm displacement of the visual stimulus consistent with the direction of locomotion. A trial started when the mouse was positioned at the beginning of an approach wall. The approach wall had black and white circles between 10 and 30 degrees in size placed randomly on the screen. The length of the approach corridor was between 20 cm and 100 cm for different sessions but was kept constant within a given session. Once the mouse completed running through the approach wall, a grating was encountered. Either a vertical grating (0°) or a diagonal grating (135°) with a spatial frequency of 0.04 cycles/° was presented. The vertical grating was the Go stimulus and the diagonal grating was the No-go stimulus. The length of the grating wall was 50 cm, and the reward zone included the last 41.7 cm. Licks in the reward zone during the presentation of a Go grating were rewarded; no reward was given for licks during a No-go grating. No punishment was given for licks during a No-go grating. Mice could earn one drop of water per presentation of the Go stimulus. The probability of a Go stimulus for any given session was set between 40 and 50%. When the end of the grating wall was reached the trial was considered complete. A 2 s inter-trial interval was included to reset synchronization of the system. During the inter-trial interval, a gray screen was presented and the locomotion of the wheel was not coupled to the motion of visual presentation. Mice completed between 38 and 220 trials. Prior to discrimination training, one baseline VS session was recorded, and two more imaging sessions were performed 15 and 30 days after training was initiated.

Performance on the visual discrimination task was quantified as d-prime (*d'*). *d'* for a given session was calculated as [Eq. 2]

$$d' = z(\text{Hit rate}) - z(\text{False alarm}) \quad (2)$$

where $z$ was the Z-transform. Hit rate was defined as the proportion of go trials in which the mice licked at least once, and false alarm was defined as the proportion of no-go trials in which the mice licked at least once.

**Locomotion modulation index.** Locomotion modulation index was computed for direct neurons using the same method as in Dipoppa et al. 2018[34]. We divided the spontaneous dark activity of a given direct neuron recorded during VS sessions before the first BCI training session into ~1 s bins (16 consecutive frames). To extract locomotion information, we computed speed of locomotion from ball-tracking camera as in our previous study[46]. The computed speed was smoothed with a moving window of 5 frames (323 ms) and was down-sampled to match the binning of the spontaneous dark activity (1 Hz). Locomotion threshold was set to 1 cm/s. The modulation index, M, for a given direct neuron was computed as below [Eq. 3].

$$M = \frac{\bar{F}_{\text{locomotion}} - \bar{F}_{\text{stationary}}}{\sqrt{\sigma^2 [F_{\text{locomotion}}] + \sigma^2 [F_{\text{stationary}}]}} \quad (3)$$

**Visual stimulation.** Static sinusoidal grating stimuli were generated using psychophysics toolbox (http://psychtoolbox.org/) in Matlab (Mathworks, Boston,

MA). The stimulus was presented on a screen positioned 25 cm away from the right eye angled at 50° with respect to the midline of the animal. The size of the screen was 64 cm by 40 cm, thereby subtending 142° × 96° of visual angle. The spatial frequency range of the stimulus set was 0.02 cycles/° to 0.3 cycles/° at 0.02 cycles/° interval. The orientations ranged from 0° to 180° at 15° spacing interval, yielding a total of 180 different sinusoidal gratings with 12 different orientations and 15 different spatial frequencies. Each grating was presented for 250 ms consecutively in a random order without interleaved gray screen. Each stimulus was shown at least 20 times.

**Quantification of visual responses.** Reverse correlation was used to determine the response window of a given stimulus[76]. The peak in the stimulus-averaged events was observed 194–320 ms after the stimulus was presented on the screen. Therefore, for each stimulus, the corresponding event activity was computed by averaging the number of events between 194 ms and 320 ms window. We defined this period as the response window for a given stimulus.

A neuron was defined as responsive to visual stimuli when the number of events following a presentation of a visual stimulus was modulated by the stimuli presented. To test for modulation, we performed a one-way analysis of variance (ANOVA, α = 0.01) on the observed events during the response window using stimuli as the factor for each neuron.

GCaMP6f expressed in neurons had longer decay than the presentation rate of our stimuli[77]. Therefore, we used deconvolution to remove the effects of decay in calcium fluorescence in quantifying responses of each neuron to our visual stimuli as in ref. [46]. Briefly, amplitude of calcium transients was expressed in units of inferred events. For each segment $n$, inferred events $s_n$ were estimated from fluorescence using the following model [Eq. 4]:

$$f_n = s_n * k + \beta_n p_n + b_n \quad (4)$$

where k is the temporal kernel and $b_n$ is the baseline fluorescence. Neuropil fluorescence, which is a contamination of the fluorescence signal $f_n$ from out of focus cell bodies and nearby axons and dendrites, is modeled by $p_n$, the time course of the neuropil contamination, and, $\beta_n$ the scaling coefficients. * denotes convolution. Using this model, $s_n$, k, $\beta_n$, and $b_n$ were estimated by a matching pursuit algorithm with L0 constraint, in which spikes were iteratively added and refined until the threshold determined by the variance of the signal was met.

Trials containing locomotion or eye blinks were removed. Pupil location was estimated from eye-tracking videos using a circular Hough Transform algorithm; the algorithm failed to find the pupil on frames during which the mice were blinking. These frames were marked as eye blink frames and removed from further analysis. Trials with locomotion were identified as in ref. [46]. Briefly, after applying a threshold on the luminance intensity of the treadmill motion images, phase correlation was computed between consecutive frames to estimate the translation between the frames. To define a motion threshold, the data were smoothed using a 1 s sliding window. Any continuous non-zero movement periods during which the animal's instantaneous running speed exceeded 10 cm/s threshold for at least one frame were marked as running epochs.

*Estimation of preferred stimulus and tuning bandwidth (Figs. 4, 5, 6, Supplementary Figs. 15–17).* Orientation and spatial frequency preference were determined using a two-dimensional Gaussian model, fit to single trial responses. For neurons that were responsive to grating stimuli, a two-dimensional Gaussian model was fit using nonlinear least-squared regression such that the number of events $R$ as a function of the orientation θ and the spatial frequency φ of the stimulus was [Eq. 5]

$$R(\theta, \varphi) = \frac{A}{2\pi\sigma_\theta\sigma_\varphi\sqrt{1-\rho^2}} e^{\left(-\frac{1}{2(1-\rho^2)}\left[\frac{(\theta-\mu_\theta)^2}{\sigma_\theta^2} + \frac{(\varphi-\mu_\varphi)^2}{\sigma_\varphi^2} - \frac{2\rho(\theta-\mu_\theta)(\varphi-\mu_\varphi)}{\sigma_\theta\sigma_\varphi}\right]\right)} + B \quad (5)$$

where $\mu_\theta$ was the preferred orientation and $\mu_\varphi$ was the preferred spatial frequency of the stimulus, and the $\sigma_\theta$ and $\sigma_\varphi$ described the widths of respective tuning. The covariance of responses for orientation and spatial frequency was captured by the correlation term ρ. A was a parameter accounting for the amplitude of the responses in number of events, while B was the baseline event activity of the cell. For fitting, the lower and the upper bound of allowed values for $\mu_\varphi$ was set by the range of the presented stimuli, which was 0.02–0.30 cycles/°. The lower bound for $\sigma_\theta$ and $\sigma_\varphi$ was set at 1 degree and 0.001 cycles/° respectively to prevent fits with zero or negative widths. Prior to fitting, the preferred orientation was initialized by estimating the preferred orientation by averaging the response, $R$ across all spatial frequencies for a given stimulus orientation, θ and calculating half the complex phase of the value [49,78] [Eq. 6].

$$S = \frac{\sum R(\theta)e^{2i\theta}}{\sum R(\theta)} \quad (6)$$

The preferred spatial frequency was initialized by selecting the spatial frequency that generated the maximal significant response at the estimated preferred orientation.

For the model above, $R^2$ of the fit was used to find neurons with significant tuning. The chance distribution of $R^2$ was calculated from fitting the above model with permuted stimulus labels on individual trials 1000 times for each neuron. Neurons whose $R^2$ exceeded the 95th percentile of the chance $R^2$ distribution were accepted as tuned to grating stimuli.

The bandwidths of the Gaussian tuning were described using half-width at half-maximum (HWHM). The HWHM bandwidths for both orientation and spatial frequency were calculated as [Eq. 7]

$$BW = \sqrt{2 * \ln(2)} * \sigma \qquad (7)$$

where $\sigma$ was the width parameter of the Gaussian fit.

*Trial reliability and failure quantification (Fig. 4).* Trial-by-trial reliability[79] of a given stimulus was computed by generating a null distribution of spontaneous activity (deconvolved events) for each neuron with the screen turned off. Trials in which the number of events was larger than the 95th percentile of the null distribution were labeled as significant responses, those less than the 95th percentile were considered failures. Spontaneous event activity was binned into same size bins as the stimulus response window (two calcium imaging frames). The number of samples for the null distributions was matched across all imaging sessions as the minimum number of bins available across the sessions. The minimum number of samples was 776. For sessions with more than the minimum samples, 776 were randomly selected from the available samples.

*Decoding (Figs. 5 and 6).* K-nearest-neighbor (KNN) classifiers were used to decode orientations from vectors of single trial population responses to grating stimuli presentation[72,79]. In our case, the k-nearest-neighbor classifier estimated the stimulus identity for a given response vector by identifying the most frequent stimulus identity of its k closest response vectors. To identify the nearest neighbors for a given response vector, we computed the Euclidean distance to other response vectors. For each session, data was divided so that a single set of response vectors consisted of one trial of each stimulus. This resulted in the number of sets being equal to the number of trials that each stimulus was shown. During decoding, the possible neighbors for a test response vector consisted of all response vectors not belonging to the test set. This ensures an unbiased representation of possible nearest neighbors across stimuli. This process was repeated across each response vector and each set. We reported the performance of this decoding process as accuracy across all response vector tested. Only the neurons that were responsive to grating stimuli were included in decoding. To compare accuracies across sessions, we matched the number of neurons used in classification to the minimum number of neurons across the session pair. To ensure that the computed accuracies were not biased by subsampling of neurons, each session was decoded ten times using a randomly selected set of neurons, and we computed the average accuracy for each session. To determine the optimal value of k, the number of neighbors, we performed decoding on the first session of each of the 4 BCI mice (4 out of 15 total mice used for decoding), sweeping k from 3 to 15. Three of four mice had accuracy ranges >5% across different values of k. For the three mice that had accuracy ranges >5% across different values of k, we ranked the values of k that yielded the highest accuracy for each mouse. We found $k = 4$ had the best average rank across mice. Therefore, we used $k = 4$ to decode orientations for sessions that were compared in the main results. The chance performance of the decoder was 8.33%.

To quantify the change in orientation discriminability of the same neurons (the tracked population) across sessions, we modified the KNN classifier described above. We trained a KNN classifier on the neuronal responses from the initial session and decoded held-out responses from the initial session and responses from the 30-day session. Only the responses from neurons that were tracked in the initial session and the 30-day session was used train and test the classification algorithm. The number of trials was matched to the minimum number of available trials across the two sessions. When the number of trials available was larger than the minimum number of trials, trials were randomly subsampled from the available trials. The value of k was fixed at 4. Each session was decoded ten times and the average accuracy across averaged across the repetitions. The difference in the accuracy between the held-out responses from the initial session and the 30-day session was considered to be the representational drift.

*Computation of signal correlations (Fig. 5).* Signal correlation $\rho^{sig}$ between a pair of neurons is defined as Pearson's correlation between the average responses to stimuli[80]. Therefore, we computed pairwise signal correlation between neuron $i$ and neuron $j$ as [Eq. 8]

$$\rho_{i,j}^{sig} = corr(\bar{R}_i, \bar{R}_j) \qquad (8)$$

where $\bar{R}$ is a vector of average response in number of spikes to 180 sinusoidal gratings for the respective neuron.

## BCI data analysis
*Two-sample proportions Z-test (Fig. 1).* We computed the Z-statistic for the hypothesis that two performances came from two separate binomial distributions against the null hypothesis that they came from the same distribution. The Z-statistic for the difference in performance between the first session and the $i$th session was computed by the following equation [Eq. 9].

$$Z = \frac{p_1 - p_i}{SE} \qquad (9)$$

SE was the standard error of the sampling distribution difference between the two performances and $p_1$ and $p_i$ were the performance of the first session and the $i$th session respectively. SE of the first session and the $i$th session was computed by the equation below [Eq. 10],

$$SE = \sqrt{p * (1-p) * \left(\frac{1}{n_1} + \frac{1}{n_i}\right)} \qquad (10)$$

where $p$ is the pooled performance between session 1 and session $i$, weighted by the number of trials, $n$, of the respective session [Eq. 11].

$$p = \frac{p_1 * n_1 + p_i * n_i}{n_1 + n_i} \qquad (11)$$

$P$ values were computed from the Z-statistic and corrected for multiple comparisons (the number of sessions) by controlling for the false discovery rate using the Benjamini–Hochberg procedure independently in each mouse.

*Quantification of DP activity at the time of success (Fig. 2b, Supplementary Table 2).* The normalized real-time fluorescence signal of DP neurons was corrected for slow drifts in fluorescence for each direct neuron, to ensure that any learning induced changes were not over-estimated. We removed events larger than 3 scaled median absolute deviations using the Matlab function 'rmoutlier'. The signal trend was estimated by computing the median with a 3000-frame moving window and subtracted from the raw fluorescence, and the resulting signal was re-centered using the median of the BCI baseline. The average fluorescence over a 387 ms window (6 frames) centered at the time of target threshold crossing was compared between session 1 and the LP session.

**Statistics and reproducibility.** We conducted experiments independently across the animals listed in Supplementary Table 1; in the the case neurons were tracked across sessions, this is noted in the legend. Error is reported as standard error of the mean (S.E.M.), unless noted. In the case data were not normally distributed, non-parametric tests were used. Alpha was set to 0.05 unless noted. Two-sided tests were used unless noted. $P$ values were not adjusted for multiple comparisons, unless noted.

**Reporting summary.** Further information on research design is available in the Nature Research Reporting Summary linked to this article.

**Disclaimers.** The views and conclusions contained herein are those of the authors and should not be interpreted as necessarily representing the official policies or endorsements, either expressed or implied, of IARPA, DoI/IBC, or the U.S. Government (S.J.K.). The U.S. Government is authorized to reproduce and distribute reprints for Governmental purposes notwithstanding any copyright annotation thereon.

## Data availability
The neural response data (Figs. 1–6) generated in this study have been deposited in the GIN database, available at at https://gin.g-node.org/bjeon/V1_BCI.git. Source data are provided with this paper.

## Code availability
The code used for analysis is available on GitHub (https://github.com/bjjeon5111/BCI_V1).

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

## Acknowledgements

We thank Alison Barth, Aaron Batista, and Emily Oby for useful discussion, Jeffrey Good for performing surgeries, and Scott Szuhay for artwork. Funded by: NIH R01EY024678 (S.J.K.), The Curci Foundation (S.J.K. and S.M.C.), PA Health Grant SAP 4100072542, NIH R21NS115036 (S.J.K. and S.M.C.) and Intelligence Advanced Research Projects Activity (IARPA) via Department of Interior/Interior Business Center (DoI/IBCContract D16PC00007 (S.J.K.).

## Author contributions

B.B.J., S.M.C., and S.J.K. designed the experiments; B.B.J. and T.F. collected the data; B.B.J., T.F., S.M.C., and S.J.K. analyzed the data, and B.B.J., S.M.C., and S.J.K. wrote the paper.

## Competing interests

The authors declare no competing interests.
