## [Peer Review File · Nature Communications]

Existing function in primary visual cortex is not perturbed by new skill acquisition of a non-matched sensory taskREVIEWER COMMENTS

Reviewer #1 (Remarks to the Author):

This work addresses an important question in the general field of neuroscience. When an animal learns a new skill, this modifies neuronal networks underlying skill execution. What is the impact of these modifications on the original function of the involved networks? In other words, can the networks accommodate one more function, or do they have to downgrade the encoding of previous functions? Focusing on the visual cortex, this study chooses to train animals to directly modulate the activity of neurons in order to receive a reward, taking advantage of direct interfacing with V1 neurons. This is an original approach that brings us closer to causality, because the skill developed is by construction due to the activity of the "direct neurons". The results argue that the network retains its original properties while developing a new function. This is important for our understanding of memory formation and maintenance, but also in the context of how brain networks can be used and manipulated for brain-machine interfaces: it suggests it is feasible to use a preserved cortical area for analyzing new sensory information without losing the normal function of that area.

Generally, the results support the main conclusion about no change in function of V1. The methodology is sound and the methods including data analysis are very detailed.

However I have several major concerns, and I think that the paper could be improved a lot by taking them into account, giving it a better chance at having a large impact.

Main concerns

1. The organization of the paper, fine at first, becomes more and more confusing as the Results part unfolds. In fact it looks like a paper that would have been carefully written once, then reorganized (order and/or content changes, vocabulary changed...) but then never fully read and corrected. This general remark includes:

- * I had to read carefully the Methods before the Results just to understand what is shown. There were some basic information missing in the main text or in the legends. Some are indicated in Minor concerns below, non-exhaustively.

- * Most analysis is done on the three visual stimulation sessions, which are referred to with names that are not homogeneous. "episode 2" is both a session in "Baseline condition" and "BCI condition" (?) and is sometimes called "Before BCI", including all 4 terms in the same Figure. This was very confusing and made the graphs difficult to understand. One terminology has to be consistently used.

- * Figure 5 was confusing. It does not address the main point of the paper, that is, the maintenance of V1 function. It is an attempt at describing the BCI trajectories - an attempt that is not convincing at least in the present form, see my comments on Fig5A and B, strategies are not clear - and it is closely related to Figure 1. So either it should be eliminated (my recommendation is to switch it to a different paper) or greatly simplified and integrated in Figure 1, if the message fits with Figure 1.

- * Figures 3 and 4 and the accompanying main text are arbitrarily organized. They both address the possible modifications of functional properties of visual neurons due to BCI training, compared to modifications around a control period (Baseline). But the rationale of the analysis shown is not clear. More systematic analyses could be shown, particularly on the direct neurons, which are largely absent in the figure (but the legends are not being always clear about which neurons are included so it's hard to tell). The difference between DP and DN neurons is not shown either.

2. The main difficulty of the paper in convincing readers is that it is based on negative results, which are always more difficult to interpret than positive results. In fact, the only significant changes shown in the paper are those occurring during the BCI task. Outside of the BCI task, there is no indication that the neurons express any plasticity. This is quite surprising, but it also means that one cannot rule out that the changes have little to do with the neuronal networks in V1. A positive result about a persistent change in the network would help.

- * Is there a change in spontaneous activity levels at rest, of direct neurons vs. Indirect neurons? Or across learning? Ideally at the start of the visual stimulation sessions, but even at the start of BCI sessions during rest would be interesting. If there is, it would mean that despite a change in

excitability at rest of the direct neurons, the network nonetheless can accommodate and reorganize so that their functional properties for visual perception are maintained. It would strengthen the conclusions.

* Another possibility, more ambitious, would be to contrast BCI training to a visual perception task that is known to induce plasticity changes in the V1 network, that could be detected. The protocol of Poort et al, cited by the authors, could be one option. I understand that these experiments are long and difficult, and it might be outside of the scope of this study. The authors could nonetheless discuss their study in relation to studies like Poort et al.

3. A related remark is that the Supplementary Figure about the Total Darkness experiment, which is discussed in the main text as a telling control, is not very helpful because it again leads only to negative results, and it is quite a drastic manipulation that does not seem very relevant to the BCI training here. If I understood correctly, BCI training occurs in the dark phase, and the animals still experience the light phase, so they cannot suffer from complete darkness effects? This point was quite confusing. The Baseline control analysis seems much more appropriate.

4. In fine, the data comes from 4 animals. I know many great papers based on few animals, so this is not necessarily an issue. However, I was sometimes confused about whether the authors think that the animals behave the same or not. The message should somehow be clearer. For example, description of Figure 2C was misleading. DP neurons AND DN neurons increase for the first 3 mice, and both decrease for the last. The authors make the increase in DP activity sound like an explanation for BCI training, and conversely the decrease in DN activity for the last mouse, but this is by construction. In fact DP and DN neurons always behaved similarly. Another confusing point concerns Fig5, which seems to show DP increases for Mouse 4. The similarity or dissimilarity of mice should be much clearer, and kept to what is informative for the question at stake.

Minor concerns

- The abstract is vague about which kind of skill is learnt (neuronal operant conditioning), in which area (motor or sensory – visual cortex should be mentioned). It makes it difficult to grasp the aims of the study.
- work by Fetz first demonstrating operant conditioning could be added to the references (together with the Koralek and Clancy studies, already cited)
- first paragraph of Results is more an Introductory paragraph, except the last sentences.
- line 81, please say that these neurons are recorded by two-photon imaging, and with which gene expressed under which promoter (at the very least).
- line 97-98, the method of identifying the first set of sessions leading to significant regression is not clear without the Methods, but it is important so should be stated at least in the legend of Figure 1. The reader does not even understand without Figure 5 that there were more sessions than shown in Figure 1.
- line 103 and Fig title, The word "excitability" is associated with other phenomena than what is meant here. I would use something much more precise and unambiguous. The authors mean the activity at BCI success times.
- Line 145 and Fig3 legend, what is meant by reliability, ie trial-by-trial, has to be said briefly without having to go to the Methods
- line 198, what are the four parameters?
- lines 207 to 209 quite redundant
- line 275, "...is lacking." Relevant works would be those of Hartmann, ... Nicolelis 2016 who show rats can learn an ultrasound task and still retain S1 normal responses, and most importantly the work by Ganguly and Carmena 2009 showing the coexistence of several motor decoders in the monkey motor cortex
- last paragraph of Discussion, the available data does not allow to say much about the benefit of variability. In fact I understood Suppl Fig 6 as showing the opposite.
- line 371, anesthesia before craniotomy
- line 451, j going from 4 to 6 would be more correct mathematically

- line 463, check frequencies, one appears twice
- line 469, say already briefly what it means to not move
- later in the paragraph, target could mean target pitch or target threshold. When used alone it's a bit confusing.
- line 492, I had the impression that forward correlation is described?
- line 503, 99% or 0.05?
- line 511, Neuropil fluorescence
- Fig 1 and later, "Learning point" used by itself was a bit confusing, the word session should appear
- Fig 2, "Activity" misspelled
- Is the Target for z-score fixed to 15 always? This would be very useful information.
- Fig2 legend is a bit incomplete. Panel C legend should describe the graph, then give the test. Panel B is not clear about whether it includes DP and DN neurons. The number of neurons should be given.
- Fig3 Title is imprecise. Enhance or modify? Which stimulus response?
- Fig3A, describe inset of neuron image in legend. Is this a direct ? DP? DN? In the legend, what is the scale mentioned? What is the spatial frequency?
- Fig3B, there is one map for a Direct neuron, and then two maps for the Indirect neuron. Why not show two maps for the Direct neuron? Why not zoom on the interesting part of the map? Also a raw response map (spots in color for example) would give a better idea than reconstructed contours, which are not very appropriate when built from 6 to 10 pixels really.
- Fig3C, the classification does not refer to before/after but to similarity to direct neuron – this is really not easy to catch
- Fig3E legend, what is meant by failures?
- Fig3F, it was difficult to understand that there are 6 points for the 6 DP neurons, but the points are actually not measured on the DP neurons, but using a population analysis. The legend is really not clear about that, at first I thought decoding was done on one neuron's data.
- Fig4A, I found the fact that there were two diagrams that are somehow the same confusing. Using only one with red and blue outlines would have been easier.
- Fig4, for each panel it should be clear if it includes DP, DN, indirect neurons.
- Fig4F, this is more or less the same graph as 3E, both on indirect neurons (?) but in a different format, why? Why is the scale only positive?
- Fig4G, same question as above
- Fig5A, first example, I don't see how this trial incorporates a decrease in DN, which should be of -20% or more of a T=15 z-score? I don't see any decrease. Same for the first trial of Learning point.
- Fig5B, the description of the colors should largely be in the legend and not in the main text. Mouse 4, I thought that success came from a decrease in DN activity, and that DP neurons had a decrease of activity too, but I see only increases in DP here?
- Suppl Fig2, the same trace appears twice for DN3, Top

Reviewer #2 (Remarks to the Author):

Jeon et al., investigate the interference of new skill learning to previously established functional properties and network activity in the primary visual cortex (V1) in mice. They use an auditory biofeedback paradigm along with real-time monitoring of 6 selected neurons per mouse within V1 to train animals to control the modulation of activity within the selected population of cells to ultimately change the frequency of an auditory tone to reach a given threshold – after which they receive a reward. They further investigate the effects of this task learning, and modulation of V1 activity, on the fundamental response properties of neurons in V1 with traditionally defined parameters such as orientation selectivity. They found that even after mice successfully learnt to modulate neuronal activity to control the auditory tone, this modulation of activity in V1 did not substantially affect visual response properties. They further describe the relationship between the responses of the 6 individually selected neurons per mouse in relation to trial-by-trial 'strategies' for reaching the given activity thresholds, and ultimately reward success. The manuscript is very well written and the flow and analysis are clear and experimentally sound. The paradigm is interesting, although perhaps the

terminology is somewhat exaggerated in relation to 'neuroprosthetic control'. The findings/conclusions are not entirely surprising, but they are demonstrated in a robust way. The paper was very interesting to read and highly captivating – so much so that it left me wanting more from the study. Although the experimental paradigm is complex, involving long training periods and sophisticated analysis/real-time experimental control, so far, the conclusions are relatively limited in their scope. Although the animal numbers are relatively low, the data is presented comprehensively and across all mice, and the analytical/statistical methods are well applied. In general, the study is well done and will be of interest to the field, but the novelty of the paradigm also leaves many unresolved questions in relation to the overall significance of the 'volitional control' of the individual neurons.

I have detailed below general points that should be addressed followed by detailed comments to the text.

1. It is unclear to me why this paradigm is referred to as a 'brain computer interface device' and not a biofeedback paradigm. For instance, in the abstract line 11. In subsequent areas they refer to it as a brain computer interface behavioural paradigm (line 34), which seem to me marginally better, as there is no specific interface device implanted here etc., but it is still unclear what the specific brain-computer 'interface' component is. For instance, there is no artificial stimulation of the selected neurons by an interface device. The change in cell activity is monitored in real-time and fed-back to the animal in the form of an auditory signal, which they can alter based on naturally altered changes in innate cell activity. For me this is a clear example of a biofeedback paradigm. This does not impact the conclusions etc., but it feels more accurate in relation to the experimental design. If the authors disagree, they need to make the definition of the specific BCI more clear in relation to traditional definitions of this term to avoid confusion or misleading the reader based on the abstract.

2. The finding that ~80% of neurons in V1 are modulated with task learning calls into question the significance of the specific modulation of the 6 selected neurons. If these responses are highly correlated, how can one conclude that the 6 specific neurons are uniquely important (as in inferred throughout the rest of the study examining the visual response properties and strategies for successful BCI performance). Without a manipulation to block/stimulate artificially, etc. these specific neurons, or evidence that their activity is uniquely 'controlling' the auditory cursor (i.e. if other combinations of cell populations can not result in the same ultimate performance outcome), then what can be concluded is more along the lines of there is a modulation of V1 network activity that correlates with the bio-feedback task. This element could easily be based on general arousal and therefore, may be affecting all areas in the brain but without a logical assumption that this would change visual response properties. If the modulation of activity is based on general arousal, the import of the 6 neurons is diminished. The inclusion of the DN neurons seems to have been oriented to control for this eventuality, so that not all cells could increase activity and still lead to success. However, over 78% of neurons in V1 still did just that – including the DN neurons in most cases (3 of 4 mice, with the 4th mouse an outlier in this regard in most measures). Without a specific control for general arousal it is difficult to know how to assign importance to the activity of the 6 selected cells.

3. Somewhat related to point 2, the effects of movement/arousal may be more elaborately monitored/analyzed in relation to the current dataset. Line 524: locomotion periods were removed (with the criteria of an instantaneous speed exceeding 10 cm/s on at least one frame), however, there are many levels of movement between completely still and 10 cm/s that may also substantially affect neural activity in V1. A further analysis of particular movements/movement patterns correlating to success and neural drive target would be interesting – particularly in the context of 'strategies' for BCI success. It would be interesting if the mice produced stereotyped motor behaviours (or not) to drive neuronal activity, regardless of locomotion (e.g. whisking, twitching, etc). Similarly, line 314-317: the link to behavioural variability/strategy is not made clear in the context of the current study that does not directly link behaviour to performance but only neural activity in a small subset of neurons. If the authors have video monitoring behaviour, an analysis of this along with the changes in neural activity

and 'strategy' would be beneficial to make a stronger point here.

4. The process of Hebbian mechanisms is tested in relation to altering the visual response properties of the V1 population. However, again, not only the 6 selected neurons are altering their activity during the task. When ~80% of neurons increase their activity the process of Hebbian plasticity will span many different response types – so I am not sure why one would expect a shift in visual response properties. If the authors had, for instance, selected 6 cells that were known to have a specific initial tuning and required them to increase their activity to reach a frequency threshold, perhaps one would expect plasticity. However, I appreciate that the point of the study was to select 'random' populations of direct neurons – but the result that the authors found with such wide-spread modulation of activity in the entire imaged V1 population diminishes the importance of these neurons in relation to a clear Hebbian hypothesis. Unfortunately, I do not have a nice suggestion for solutions here beyond further experiments to test the limits of what does and does not produce functional plasticity in V1. This seems beyond the scope of this study, but perhaps the authors can rework this to have less emphasis on the individual cells and more on the general population dynamics (in relation to cross-modal new learning not interfering with established functional properties).

5. Throughout the manuscript, from the title to the conclusion, it should be made clear that the study is examining the effects of a cross-modal task. If the title is interpreted to mean a new visual skill in visual cortex (or any sensory matched area) then this would not be accurate, and from a plethora of previous literature (as cited by the authors) I suspect the outcome would be different. In general, there is not enough focus on emphasizing that these are effects from a non-matched sensory task in the title/abstract especially.

Specific points:

- Line 59/60: "our results demonstrate that a new skill can be integrated into existing networks...". It is unclear as to how the acquired 'skill' is really 'integrated' into the V1 network here. The majority of neurons altered their activity, but this may have been brain wide. There did not seem to be a clear difference between the activity of the direct neurons and that of the rest of the V1 population, indicating that the task elicited a general V1-wide gain in response. It is unclear how this correlates to the integration of a new skill into the V1 network. If the authors have a justification for this wording it should be made more explicit, otherwise it should be reworded.

- Line 80/441: neurons were randomly chosen. Since changes in activity to drive the auditory cursor were bi-directional, was there no pre-determination of spontaneous activity level in neurons? Can you explain the 'randomness' of randomly choosing in more detail. Were they spatially spread out or just had to be isolated from other neurons but could be in close proximity to each other, had a certain baseline activity, etc. A sample FOV with real example instead of (or in addition to) the schematic in Figure 1A may be helpful. For instance, in Figure S2 the DN neurons all seem to have a very low spontaneous rate even in the first session. It is true that these neurons are still acting as controls so that the mouse can not simply increase the activity of all neurons to be successful in the BCI, but it seems unlikely that these particular neurons would be able to really be 'downregulated' with such low spontaneous rates? Can the authors present the spontaneous rates of the direct neurons across sessions and expand on their dynamics in relation to the general population in this regard.

- Line 88: it would be helpful to indicate explicitly in the results paragraph at this point that the target threshold is 15 kHz on the auditory cursor

- Figure S3 (line 116-117): it would be useful to see the equivalent data for the 15 kHz tone in Figure S3, since this is perhaps the most behaviourally relevant tone, including the comparable numbers that are presented in the Figure caption in S3B for 15 kHz. Additionally, since (B) shows data by trial, it would also be good to see the data averaged across trials for all cells (connecting the individual cells between baseline and stimulus conditions) for the 5 kHz and 15 kHz tones. Since the auditory systems

shows relatively fast adaptation, I wonder if there would be differences if a smaller time window was selected for this analysis (e.g. first 1 second, for instance).

- Lines 116-117 state that V1 neurons were not modulated by the pitches, however, it should be noted in the results that 14/95 neurons were actually modulated by the tones, as stated in the S3 Figure caption. This is a small proportion, but not none. Of course it would be useful to have some data regarding how many neurons were modulated by the tones before training in these same mice, but presumably the study did not include this measure. As the authors state, auditory stimuli are not known to have a net excitatory influence on neurons in V1 – although they can exert inhibitory modulation in V1, even in untrained mice. If one follows the authors previous statement and assumes the number would be 0/95 pre training, this still represents an increase in the number of auditory modulated neurons in V1 after learning. Considering the paradigm is using modulation of a very small number of neurons to elicit behavioural changes, increase in even a small number of neurons here may contribute to an overall effect. The authors may still conclude that BCI performance was not likely to be improved by auditory sensitivity alone – but I am not sure it has to be an all or nothing effect in this regard. Especially since they show that 81% of neurons show a significant change in activity at the time of success, just because the experiment monitored 6 neurons, does not exclude the possibility that one of the other 81% was highly correlated with task success and was one of the 14/95 modulated cells. The authors state that the changes in these 14/95 cells were small on average, but this may also be partly because responses are averaged across 2 second windows. I think it would be more convincing to show more of this data. Perhaps the conclusion will not change – but the current analysis leaves me with many open questions.

- Lines 124-125: this seems like a strong statement considering that such a large proportion of neurons in V1 had significant changes in their activity level. Presumably if Figure 2C is done with all non-direct neurons the result would be the same? Is there any evidence to suggest that BCI performance was improved as a direct result of the changes in direct neurons? One may randomly select any 6 cells post-hoc and come to the same conclusion, that changes in their activity could account for improved performance. If this is not the case, I would present more evidence for a unique role of the direct neurons in performance. Alternatively, I am not sure this is necessary to the point, as changes in the population in general may be reflected in the direct neurons and vice versa – but the specific conclusions should be made more generic to apply to the whole population in that case.

- Relatedly, it might be useful to examine the precise correlations between modulated cells in the rest of the FOV and the direct neurons. Do these cells have stronger pairwise correlations for instance? Can the selected direct neurons be used as a sort of 'hub' for activity change in the population during the task? Or perhaps they are simply reflective of it.

- Lines 121-124: it is a bit difficult to interpret the difference between DP and DN modulation here. Since they both change in the same direction, simply based on which mouse is examined. The way the statements are formulated are a bit misleading to suggest that the DP and DN neurons were more or less relevant to learning in particular mice – but it seems they are not actually being modulated differently with respect to each other (DP vs DN populations). This is not an irrelevant point and the significance of this should be made more clear.

- Section 127-156: the results here presume that the selectivity of the direct neurons was also stable. This is discussed in further sections, but largely in relation to the indirect population. It is only digging in figure captions that one gets more information regarding the 6 selective direct neurons and still their properties are not fully presented across episodes as the indirect population are. In Figure 3B the authors should also show the direct neuron prior to and after BCI. Line 144: 39 had similar orientation tuning to a direct neuron, at which time point? Before or after BCI? Or either? Again, this does not change the conclusion that the BCI task did not enhance stimulus responses, I think this is a logical conclusion, but the stability/selectivity, or lack thereof, for direct neurons should also be made clear throughout this and the following results section.

- Line 472: the target was predetermined at the start of learning and did not change across sessions, but the spontaneous activity was redone each session – to calculate z-score. If the spontaneous activity drifted towards the target across sessions then absolute changes in activity would have become smaller to reach the same target. This could create a 'learning' curve simply with persistent changes in baseline activity across days. Can the authors comment on this possibility or present evidence to the contrary.
- Figure 1A: please define DP and DN in caption.
- Supp Figure 1C: perhaps highlighting the first lick response following the onset would help to visualize the learning effect in the given example.
- Line 389: I assume the system was a resonant-galvo system? Could the authors please indicate the frequency of the resonant scanner (8kHz, 12 kHz?) as this generally provides auditory noise during imaging.
- Line 402: Could the authors explicitly comment on how many neurons reached this criteria, i.e., the proportion of neurons removed during this process.
- Line 420-421: 'Lick port was armed with reward...' Can the authors describe how this was done in more detail. Was the reward droplet pushed out only on Go trials? Would this not act as a cue in and of itself? Was this associated with an auditory cue (pump for reward spout) or perhaps olfactory cue was important here. I am not sure if affects the outcome of the results, but specifics on the parameters here would be useful.
- Line 577: Response amplitude was computed as the average response across trials with a significant response... what if this measure is taken across all trials? As in, did the general modulation of activity in V1 lead to increased noise?
- Line 594: only neurons that were responsive to grating stimuli were included in the decoding. The numbers across animals are listed in Figure 3F caption, but how did these numbers change across timepoints?
- Line 283: extra 'well as'

Reviewer #3 (Remarks to the Author):

The manuscript "New skill acquisition does not perturb existing function in sensory cortex" from Kuhlman and colleagues describes how existing cortical responses in visual cortex react to BCI-based conditioning of neurons in the same area. This is a very pertinent and long standing question in the field and the authors have chosen an original experimental approach to answer it. We are in principle very enthusiastic about the study and at a first glance, the results are very intriguing.

However, while reading more carefully we discovered several important points which need substantial clarification. Please find below our suggestions and comments. We hope they help to improve the next version of this manuscript.

Our main concerns are about the implementation of the BCI learning paradigm. We have doubts that the animals actually learn a BCI task. The characterization of the visual response types seems to be done correctly, but we did not go over this part in detail. All the results hinge on the assumption that

animals actually undergo a BCI task.

Several important controls are currently missing (see details below). Also, it looks like some of the data has been omitted (additional days in Fig. 1C) and other data seems to be incorrect (Ca⁺ traces of DN3 in Supplementary Figure 2A are identical in both conditions, trial structure in the last 20 sec in DP1&2 of Supplementary Figure 2B should be identical to DP3, but it is not). These might be simple copy-paste errors, but they might also indicate that more careful work is necessary to claim robust learning. Finally, the relatively low number of animals also does not help to make a strong case.

Behavior:

It has previously been shown that movement in complete darkness causes strong activation of visual cortex neurons. Since this behavior is carried out on an air-lifted ball, it seems crucial to closely monitor and analyze the behavior of the mice and compare their actions across sessions. Much of the observations (increase in the DP and general increase in a majority of neighboring neurons) might be solely related movement parameters. The movement parameters might simply become more stereotypical and this induces stronger overall activity, which might lead (among others) to an increase in DP neurons. The authors will need to convincingly demonstrate that this is not the case.

Finally, the 1s of "not moving" seems much too short to assure that actual movements are not reinforced.

Learning point:

It is not clear how the "learning point" is defined. If the learning point is defined by the crossing of a high performance level, then the slope of the regression will inherently be positive. If the learning point is defined by some goodness of fit (as indicated by the p-value?) then we don't see what this has to do with learning.

As pointed out in the intro, we have the impression that the performance plot in Fig. 1C only shows a part of the actual sessions (the additional days are visible in Fig. 5B). How did the performance evolve after the day indicated by the "learning point"?

Unfortunately, Fig. 5B also does not allow us to determine this, since only the successful trials are shown. The day where the animal reaches the learning point seems to be accompanied with a clear change in the "strategy" which could reflect a performance decrease. Please clarify and show all the available data (overall performance for all days).

Quality of the Ca⁺ signals, definition of "neuronal drive" and choice of the neurons:

The few available calcium traces show very little activity (Fig. S2) and lots of noise (Fig. S2A: DN2, Fig. S2B Dn2, DN3). The z-scoring most probably adds to this distortion. As outlined above, a part of the data Fig. 2A seems clearly incorrect (Ca⁺ traces of DN3 in Supplementary are identical in both conditions, trial structure in the last 20 sec in DP1&2 of Supplementary Figure 2B should be identical to DP3, but it is not). Also, most of the "activity" in Fig. 5A looks more like noise with no actual events. How do the authors explain this?

A second issue is related to the definition of "neuronal drive" and "success". Although the threshold is fixed in the beginning for each animal, the data are z-scored across sessions, which does not allow to compare the neuronal activity across sessions. This is highly problematic. Furthermore the noise level of the z-scored data seems to change within a session (DN neurons in S2A&B) which is very weird. How can this be explained if the imaging conditions were stable and Z-scoring constant? Finally the initial threshold definition (30% of all trials successful) is also problematic since with extremely low activity levels, noise could trigger threshold crossings. How can the author demonstrate that this is not the case?

Taken together, the method used by the authors seems to allow for the slightest change in activity in a single neuron (see for example: Fig. S2A cell DP3 @ 160 sec) to trigger a "successful" trial. This does not make any sense and contrary to most other BCI studies where either robust single cell activity or strong combined activity of multiple neurons is necessary.

In addition, it is not clear how the authors exactly chose the DP and DN neurons. They describe it as random. This is obviously not ideal for many reasons. In Fig. S2A&B all DN neurons seem very silent. How come? There could be an inherent bias (see suggestions for controls below). Better choice definitions or at least quantifying the activity rates of all neurons might help to clarify this point.

Controls

The following controls might help to show more convincingly that the animals actually "learn" and the chosen cells or that feedback play a role:

The authors need to show that learning is specifically linked to the chosen neurons and not just due to random fluctuations, a biased choice or artifact induced by z-scoring. One easy way to do this might be to choose arbitrary sets of 3 neurons from the existing data set of active neighboring neurons, apply the same analysis methods and assess in how many cases "learning" would occur. Furthermore the timing of the neighboring neurons should be analyzed in relation to the peak of the DP neuron or to reward delivery. This might provide precious cues of conditioning locations.

The authors also need to show that the "learning" can not be observed in animals where the rewards are delivered based in a non-BCI setting. For example during fixed-pattern, random or played back of previously recorded activity. Animals would need to be trained for >10 day with this uncoupled activity to see if any increased activity starts to occur within the selected set of neurons. "Playback" would ideally be chosen based on the neuronal activity from existing "learning" sessions to match the timing and the number of rewards.

Other general issues:

It would be helpful to show the activity rate for all other neighboring neurons (Fig.1 & 5) and determine if they also change in a similar way.

Showing more representative unprocessed /processed data across the entire session or across sessions would help to illustrate the overall quality of the imaging data and evolution of the activity. Showing representative animal behavior videos would be very helpful (see above).

More detailed issues:

Fig.1 :

the neuronal drive is plotted on an arbitrary axis & on a pitch level axis. Why is the second pitch level representation slightly different? It only makes sense to show actual auditory feedback on a pitch axis.

Fig. 2:

Fig. 2 clearly shows that there is an overall increase in the DN neurons across sessions. How is that compatible with the idea that they actually should decrease in activity within a session? This might need to be discussed. What does the dashed vertical line stand for? How can DN activity be negative at the start??

Fig 3 :

The general message seems to be that there is no change. However, we can only see 6 DP because they were the only ones responding to specific visual stimulation. It would be nice if all neurons used for the BCI were characterized before training. What is the function of the remaining 6 DP neurons? What about DN neurons?

The amplitude scale in Fig. 3B might work better from black-blue-yellow

Fig 4:

The left panels of Fig. 4E clearly shows that the signal correlation clearly becomes weaker after BCI. However, the summary plot on the right does not reflect this. Everything is centered around 0. This is most likely because the low correlation in all other neurons dominates the analysis. There might be a

better way to analyze this point?

Fig. 5.

Fig. 5A all calcium traces look bizarre and noisy. How come? Please show longer traces or more trials.

In A, the target dashed threshold line is not visible enough

Legend of B currently reads "Stragagy"

FigS1:

Typo in the legend for B. Probably means 81% instead of 0.81%? It might also be important to show the number of days it takes to get this the initial association.

FigS2: See multiple comments above.

We hope these comments help to strengthen this otherwise interesting study.

Daniel Huber & Antoine Philippides

We thank the reviewers for their thorough and helpful comments. In addition, we were pleased that the overall importance of the work was considered high. We addressed all of the concerns with more experiments and further analysis; the manuscript is much improved.

There were three major concerns raised:

(1) Reviewer #2 and #3 asked for more brain computer interface (BCI) control experiments and analysis to confirm that the neurons directly coupled to the BCI were unique in their function relative to the other neurons in the field of view that were not directly coupled to the device ('indirect neurons'). We now include a shuffle analysis of the indirect neurons (new Figure S8) in which chance performance was estimated by creating a null distribution of indirect neuron activity. The new analysis demonstrates that the improved performance of the neurons directly coupled to the BCI is indeed statistically distinct from the indirect neurons. Furthermore, an additional BCI control animal is now included, to demonstrate that a longer period of training (>10 sessions) does not lead to increased performance of indirect neurons. Many more examples of raw data as well as movies are now included, and new analysis of the raw fluorescence signal confirms that our imaging conditions were sufficiently stable both within and across BCI training sessions to interpret our method of z-score normalization.

(2) Reviewer #1 and #2 suggested that the impact of the work could be improved by discussing the results of our non-matched sensory study in comparison with single-modality learning, e.g. visual discrimination, given that we are imaging neural activity in primary visual cortex. We are excited to now include results from 6 new animals that were trained in a modified version of a closed-loop visual discrimination task first published by Dr. Sonia Hofer's lab. In terms of addressing how new skills are integrated into existing circuits, we were able to confirm that many of our non-matched sensory task results also apply to single modality learning. In addition, we identified novel effects of learning on circuit function that persisted outside of the trained context.

(3) Reviewer #1 asked that improvements be made to the organization of the manuscript in terms of writing, and all reviewers provided thoughtful suggestions to improve readability. We addressed these organizational concerns and incorporated all of the provided suggestions.

Our responses to individual comments are included below. We uploaded a version of the revised manuscript in which all major changes are highlighted in blue, and a second version was uploaded in which changes are not highlighted. Note, to improve readability, we now refer to 'neural drive' (the pooled activity of the 6 direct neurons that drives target threshold crossings), as the 'control signal'.

REVIEWER COMMENTS

Reviewer #1 (Remarks to the Author):

This work addresses an important question in the general field of neuroscience. When an animal learns a new skill, this modifies neuronal networks underlying skill execution. What is the impact of these modifications on the original function of the involved networks? In other words, can the networks accommodate one more function, or do they have to downgrade the encoding of previous functions? Focusing on the visual cortex, this study chooses to train animals to

directly modulate the activity of neurons in order to receive a reward, taking advantage of direct interfacing with V1 neurons. This is an original approach that brings us closer to causality, because the skill developed is by construction due to the activity of the “direct neurons”.

Thank you for identifying our work as important.

The results argue that the network retains its original properties while developing a new function. This is important for our understanding of memory formation and maintenance, but also in the context of how brain networks can be used and manipulated for brain-machine interfaces: it suggests it is feasible to use a preserved cortical area for analyzing new sensory information without losing the normal function of that area. Generally, the results support the main conclusion about no change in function of V1. The methodology is sound and the methods including data analysis are very detailed. However I have several major concerns, and I think that the paper could be improved a lot by taking them into account, giving it a better chance at having a large impact.

Main concerns

1. The organization of the paper, fine at first, becomes more and more confusing as the Results part unfolds. In fact it looks like a paper that would have been carefully written once, then reorganized (order and/or content changes, vocabulary changed...) but then never fully read and corrected. This general remark includes:

* I had to read carefully the Methods before the Results just to understand what is shown.

Substantial updates to the organization were made, including clearly defining which analyses included direct versus indirect neuron data.

There were some basic information missing in the main text or in the legends. Some are indicated in Minor concerns below, non-exhaustively.

* Most analysis is done on the three visual stimulation sessions, which are referred to with names that are not homogeneous. “episode 2” is both a session in “Baseline condition” and “BCI condition” (?) and is sometimes called “Before BCI”, including all 4 terms in the same Figure. This was very confusing and made the graphs difficult to understand. One terminology has to be consistently used.

Agreed, we now use the terms visual stimulation (VS) baseline 1, VS baseline 2, and VS post-learning. The analyses that report the change between two imaging sessions are referred to as the Baseline condition (the difference between ‘VS baseline 1’ and ‘VS baseline 2’) and the BCI condition (the difference between ‘VS baseline 2’ and ‘VS post-learning’).

* Figure 5 was confusing. It does not address the main point of the paper, that is, the maintenance of V1 function. It is an attempt at describing the BCI trajectories - an attempt that is not convincing at least in the present form, see my comments on Fig5A and B, strategies are not clear - and it is closely related to Figure 1. So either it should be eliminated (my recommendation is to switch it to a different paper) or greatly simplified and integrated in Figure 1, if the message fits with Figure 1.

Agreed, this figure was removed. The relevant aspects are now integrated into Figs. 2A, S4-7.

* Figures 3 and 4 and the accompanying main text are arbitrarily organized. They both address the possible modifications of functional properties of visual neurons due to BCI training,

compared to modifications around a control period (Baseline). But the rationale of the analysis shown is not clear.

We identified and updated confusing phrasing in the Fig. 3 legend and the Results section. We hope that the rationale for specifically examining whether the responses of the subset of indirect neurons that were initially similarly tuned to direct neurons (e.g. prior to BCI training) were enhanced is now clear.

More systematic analyses could be shown, particularly on the direct neurons, which are largely absent in the figure (but the legends are not being always clear about which neurons are included so it's hard to tell). The difference between DP and DN neurons is not shown either.

Agreed. We now include a new Fig. S3 and Movies #2-5 to clearly show the contributions of DP and DN neurons. Direct neuron analysis is now reported in table format, Tables 1-4. In addition, examples of direct neuron activity patterns during individual BCI trials (F/F_0) are now included in Figs. 2, and S4-7.

2. The main difficulty of the paper in convincing readers is that it is based on negative results, which are always more difficult to interpret than positive results. In fact, the only significant changes shown in the paper are those occurring during the BCI task. Outside of the BCI task, there is no indication that the neurons express any plasticity. This is quite surprising, but it also means that one cannot rule out that the changes have little to do with the neuronal networks in V1. A positive result about a persistent change in the network would help.

* Is there a change in spontaneous activity levels at rest, of direct neurons vs. Indirect neurons? Or across learning? Ideally at the start of the visual stimulation sessions, but even at the start of BCI sessions during rest would be interesting. If there is, it would mean that despite a change in excitability at rest of the direct neurons, the network nonetheless can accommodate and reorganize so that their functional properties for visual perception are maintained. It would strengthen the conclusions.

We are pleased that the reviewer found the results to be novel. We thank the reviewer for this thoughtful suggestion, and now report that noise correlation among V1 neurons outside of the BCI context is increased after BCI learning (new Fig. 2F). Analysis is reported for all neurons, including indirect neurons. Direct neuron values can be found in new Table 1. Furthermore, a more in-depth analysis of indirect-direct neuron pair coupling is provided. Together, these new analyses give a complete picture of the changes to V1 circuit function both during and outside of the task context.

* Another possibility, more ambitious, would be to contrast BCI training to a visual perception task that is known to induce plasticity changes in the V1 network, that could be detected. The protocol of Poort et al, cited by the authors, could be one option. I understand that these experiments are long and difficult, and it might be outside of the scope of this study. The authors could nonetheless discuss their study in relation to studies like Poort et al.

We thank the reviewer for this suggestion, and we were actually able to train mice in this task as recommended (new Fig. 5).

3. A related remark is that the Supplementary Figure about the Total Darkness experiment, which is discussed in the main text as a telling control, is not very helpful because it again leads

only to negative results, and it is quite a drastic manipulation that does not seem very relevant to the BCI training here. If I understood correctly, BCI training occurs in the dark phase, and the animals still experience the light phase, so they cannot suffer from complete darkness effects? This point was quite confusing. The Baseline control analysis seems much more appropriate.

We agree. The description of these results is now moved to the end of the Results section, and the rationale for the experiment is clarified.

4. In fine, the data comes from 4 animals. I know many great papers based on few animals, so this is not necessarily an issue. However, I was sometimes confused about whether the authors think that the animals behave the same or not. The message should somehow be clearer. For example, description of Figure 2C was misleading. DP neurons AND DN neurons increase for the first 3 mice, and both decrease for the last. The authors make the increase in DP activity sound like an explanation for BCI training, and conversely the decrease in DN activity for the last mouse, but this is by construction. In fact DP and DN neurons always behaved similarly. Another confusing point concerns Fig5, which seems to show DP increases for Mouse 4. The similarity or dissimilarity of mice should be much clearer, and kept to what is informative for the question at stake.

We agree that the description of Fig. 2C was unclear, and removed this analysis. We now provide a slightly different analysis in Fig. 2 emphasizing the similarities of DP neurons in all 4 mice. This, in combination with the new movies and Fig. S3 regarding the role of DNs should fix the problem.

Minor concerns

- The abstract is vague about which kind of skill is learnt (neuronal operant conditioning), in which area (motor or sensory – visual cortex should be mentioned). It makes it difficult to grasp the aims of the study.

The abstract is now updated as suggested.

- work by Fetz first demonstrating operant conditioning could be added to the references (together with the Koralek and Clancy studies, already cited)

Thank you for this suggestion. The Fetz reference is now integrated.

- first paragraph of Results is more an Introductory paragraph, except the last sentences.

Agreed and updated.

- line 81, please say that these neurons are recorded by two-photon imaging, and with which gene expressed under which promoter (at the very least).

Updated as suggested.

- line 97-98, the method of identifying the first set of sessions leading to significant regression is not clear without the Methods, but it is important so should be stated at least in the legend of Figure 1. The reader does not even understand without Figure 5 that there were more sessions than shown in Figure 1.

Updated as suggested. See also new Fig. S10 for analysis of the last BCI session.

- line 103 and Fig title, The word “excitability” is associated with other phenomena than what is meant here. I would use something much more precise and unambiguous. The authors mean the activity at BCI success times.

Updated as suggested.

- Line 145 and Fig3 legend, what is meant by reliability, ie trial-by-trial, has to be said briefly without having to go to the Methods

Updated as suggested.

- line 198, what are the four parameters?

Updated to include this information and clarified. The 4 parameters are: orientation preference, orientation bandwidth, spatial frequency preference, and spatial frequency bandwidth.

- lines 207 to 209 quite redundant

Updated.

- line 275, “...is lacking.” Relevant works would be those of Hartmann, ... Nicolelis 2016 who show rats can learn an ultrasound task and still retain S1 normal responses, and most importantly the work by Ganguly and Carmena 2009 showing the coexistence of several motor decoders in the monkey motor cortex

Thank you, these references are now included in the Introduction, paragraph #1.

- last paragraph of Discussion, the available data does not allow to say much about the benefit of variability. In fact I understood Suppl Fig 6 as showing the opposite.

This paragraph is now removed.

- line 371, anesthesia before craniotomy

Updated as suggested.

- line 451, j going from 4 to 6 would be more correct mathematically

Updated as suggested.

- line 463, check frequencies, one appears twice

Updated.

- line 469, say already briefly what it means to not move

Updated as suggested, under the Methods section heading '*Data Acquisition, Neuron Segmentation, and Neuron Tracking*' (see also Fig. S9 for an example of locomotion detection).

- later in the paragraph, target could mean target pitch or target threshold. When used alone it's a bit confusing.

Updated as suggested, 'target' was removed when referring to pitches.

- line 492, I had the impression that forward correlation is described?

The confusing sentence was deleted

- line 503, 99% or 0.05?

Updated to $\alpha = 0.01$.

- line 511, Neuropil fluorescence

Updated as suggested.

- Fig 1 and later, "Learning point" used by itself was a bit confusing, the word session should appear

Updated as suggested.

- Fig 2, "Activity" misspelled

Thank you, new Fig 2E is updated.

- Is the Target for z-score fixed to 15 always? This would be very useful information.

The target is not always fixed to 15. The range used in this study was 7-15. This information is now stated in the legend of Figure S8.

- Fig2 legend is a bit incomplete. Panel C legend should describe the graph, then give the test. Panel B is not clear about whether it includes DP and DN neurons. The number of neurons should be given.

Fig2D-F panels now all include number of neurons.

- Fig3 Title is imprecise. Enhance or modify? Which stimulus response?

Updated as suggested.

- Fig3A, describe inset of neuron image in legend. Is this a direct ? DP? DN? In the legend, what is the scale mentioned? What is the spatial frequency?

Updated as suggested. (it is an indirect neuron, 0.04 cycles/°)

- Fig3B, there is one map for a Direct neuron, and then two maps for the Indirect neuron. Why not show two maps for the Direct neuron? Why not zoom on the interesting part of the map? Also a raw response map (spots in color for example) would give a better idea than reconstructed contours, which are not very appropriate when built from 6 to 10 pixels really.

We now include a cropped version as new Fig. S13. The Fig. 3B display is designed to emphasize that the neurons are well-tuned and not responsive to other stimuli, as well as indicate the range of spatial frequencies used in the study.

- Fig3C, the classification does not refer to before/after but to similarity to direct neuron – this is really not easy to catch

Updated as suggested.

- Fig3E legend, what is meant by failures?

Failures are now defined in the Methods.

- Fig3F, it was difficult to understand that there are 6 points for the 6 DP neurons, but the points are actually not measured on the DP neurons, but using a population analysis. The legend is really not clear about that, at first I thought decoding was done on one neuron's data.

This analysis was removed.

- Fig4A, I found the fact that there were two diagrams that are somehow the same confusing. Using only one with red and blue outlines would have been easier.

We want to make sure that it is clear that the Baseline condition and the BCI condition include a slightly different pool of neurons. Now that we clarified the description and labeling of the three visual stimulation imaging sessions, we believe that this panel is less confusing and would like to keep as is.

- Fig4, for each panel it should be clear if it includes DP, DN, indirect neurons.

Agreed. Tables of direct are now included (Tables 1-4)

- Fig4F, this is more or less the same graph as 3E, both on indirect neurons (?) but in a different format, why? Why is the scale only positive?

Thank you for this suggestion. We performed additional analysis and detected a small but significant difference in the stability of response amplitude. Fig. 4F is updated to include the new analysis. Interestingly, the fraction of neurons that have a decrease in amplitude is larger in the BCI condition compared to baseline conditions.

- Fig4G, same question as above

We agree, decoding in Fig. 3 did not add much to the narrative and is removed.

- Fig5A, first example, I don't see how this trial incorporates a decrease in DN, which should be of -20% or more of a T=15 z-score? I don't see any decrease. Same for the first trial of Learning point.

- Fig5B, the description of the colors should largely be in the legend and not in the main text. Mouse 4, I thought that success came from a decrease in DN activity, and that DP neurons had a decrease of activity too, but I see only increases in DP here?

Fig. 5A and B were removed.

- Suppl Fig2, the same trace appears twice for DN3, Top
Thank you, the issue is corrected.

Reviewer #2 (Remarks to the Author):

Jeon et al., investigate the interference of new skill learning to previously established functional properties and network activity in the primary visual cortex (V1) in mice. They use an auditory biofeedback paradigm along with real-time monitoring of 6 selected neurons per mouse within V1 to train animals to control the modulation of activity within the selected population of cells to ultimately change the frequency of an auditory tone to reach a given threshold – after which they receive a reward. They further investigate the effects of this task learning, and modulation of V1 activity, on the fundamental response properties of neurons in V1 with traditionally defined parameters such as orientation selectivity. They found that even after mice successfully learnt to modulate neuronal activity to control the auditory tone, this modulation of activity in V1 did not substantially affect visual response properties. They further describe the relationship between the responses of

the 6 individually selected neurons per mouse in relation to trial-by-trial 'strategies' for reaching the given activity thresholds, and ultimately reward success. The manuscript is very well written and the flow and analysis are clear and experimentally sound. The paradigm is interesting, although perhaps the terminology is somewhat exaggerated in relation to 'neuroprosthetic control'. The findings/conclusions are not entirely surprising, but they are demonstrated in a robust way. The paper was very interesting to read and highly captivating

Thank you for your enthusiasm for our study.

– so much so that it left me wanting more from the study. Although the experimental paradigm is complex, involving long training periods and sophisticated analysis/real-time experimental control, so far, the conclusions are relatively limited in their scope. Although the animal numbers are relatively low, the data is presented comprehensively and across all mice, and the analytical/statistical methods are well applied. In general, the study is well done and will be of interest to the field, but the novelty of the paradigm also leaves many unresolved questions in relation to the overall significance of the 'volitional control' of the individual neurons.

We now include further analysis of the coupling between indirect and direct neurons (Fig. S11) and define chance performance using indirect neurons (Fig. S8). We believe the new analysis should fully address this concern. We clarify our definition of 'volitional control' in the Results section, heading "", last paragraph. By volitional control we are referring to the device, not the individual neurons. Control of individual neurons is now described as 'credit assignment'.

Credit assignment is not a focus of this manuscript. 'Volitional control' was removed from the Results heading (see also our response to comment 1 below).

I have detailed below general points that should be addressed followed by detailed comments to the text.

1. It is unclear to me why this paradigm is referred to as a 'brain computer interface device' and not a biofeedback paradigm. For instance, in the abstract line 11. In subsequent areas they refer to it as a brain computer interface behavioural paradigm (line 34), which seem to me marginally better, as there is no specific interface device implanted here etc., but it is still unclear what the specific brain-computer 'interface' component is. For instance, there is no artificial stimulation of the selected neurons by an interface device. The change in cell activity is monitored in real-time and fed-back to the animal in the form of an auditory signal, which they can alter based on naturally altered changes in innate cell activity. For me this is a clear example of a biofeedback paradigm. This does not impact the conclusions etc., but it feels more accurate in relation to the experimental design. If the authors disagree, they need to make the definition of the specific BCI more clear in relation to traditional definitions of this term to avoid confusion or misleading the reader based on the abstract.

Our definition of the specific BCI in relation to traditional definitions is clarified in the Introduction. The updated Abstract should also help fix this issues. We agree that the wording in the original submission conveyed an over emphasis on 'volitional control', given the manuscript does not address the mechanism by which the direct neurons gain control, rather, our focus is on the impact on function outside of the trained context. Headings and figure titles are updated accordingly.

2. The finding that ~80% of neurons in V1 are modulated with task learning calls into question the significance of the specific modulation of the 6 selected neurons. If these responses are highly correlated, how can one conclude that the 6 specific neurons are uniquely important (as in inferred throughout the rest of the study examining the visual response properties and strategies for successful BCI performance). Without a manipulation to block/stimulate artificially, etc. these specific neurons, or evidence that their activity is uniquely 'controlling' the auditory cursor (i.e. if other combinations of cell populations can not result in the same ultimate performance outcome), then what can be concluded is more along the lines of there is a modulation of V1 network activity that correlates with the bio-feedback task.

We agree that an important control analysis was not included in the first submission. We now include analysis for each of the 4 mice in which chance level of performance is derived from the activity of indirect neurons (new Fig. S8). This new analysis demonstrates that the direct neurons are statistically distinct from indirect neurons. We now clarify the role of DN neurons in the Results section: 'including the DN neurons ensured that there must be *differential* modulation of activity among the direct neurons at the time the neural trajectory successfully crossed the target threshold, i.e. to reach the target threshold, the DP neurons were required to be more active than the DN neurons. Thus, the animal could not solve the task by uniformly

applying an increase in activity level across all neurons in the imaging region, as might happen during states of high arousal, including but not limited to locomotion.'

This element could easily be based on general arousal and therefore, may be affecting all areas in the brain but without a logical assumption that this would change visual response properties.

This concern is related to a concern raised by Reviewer #1, that the lack of a positive effect on V1 activity outside of the BCI context weakens the rationale for examining the stability of visual responses. New analysis revealed that pairwise noise correlation among V1 neurons was increased outside of the BCI task context (new Fig 2F). Given this, it is entirely possible that visual response properties and stimulus decoding were impacted by BCI learning.

If the modulation of activity is based on general arousal, the import of the 6 neurons is diminished. The inclusion of the DN neurons seems to have been oriented to control for this eventuality, so that not all cells could increase activity and still lead to success. However, over 78% of neurons in V1 still did just that – including the DN neurons in most cases (3 of 4 mice, with the 4th mouse an outlier in this regard in most measures). Without a specific control for general arousal it is difficult to know how to assign importance to the activity of the 6 selected cells.

This issue was also raised by Reviewer #3. We now include evidence that locomotion is not correlated with BCI performance (new Fig. S9). We hasten to add, although it is true that many papers in the mouse vision field demonstrate that locomotion, which is strongly coupled to pupil diameter, results in enhancement of visual responses¹⁻⁹ if the animal is in front of a gray screen or visual stimulation, when the impact of locomotion on V1 activity is carefully monitored in darkness similar to our BCI conditions¹⁰, locomotion has little impact on V1 activity.

3. Somewhat related to point 2, the effects of movement/arousal may be more elaborately monitored/analyzed in relation to the current dataset. Line 524: locomotion periods were removed (with the criteria of an instantaneous speed exceeding 10 cm/s on at least one frame), however, there are many levels of movement between completely still and 10 cm/s that may also substantially affect neural activity in V1. A further analysis of particular movements/movement patterns correlating to success and neural drive target would be interesting – particularly in the context of 'strategies' for BCI success. It would be interesting if the mice produced stereotyped motor behaviours (or not) to drive neuronal activity, regardless of locomotion (e.g. whisking, twitching, etc). Similarly, line 314-317: the link to behavioural variability/strategy is not made clear in the context of the current study that does not directly link behaviour to performance but only neural activity in a small subset of neurons. If the authors have video monitoring behaviour, an analysis of this along with the changes in neural activity and 'strategy' would be beneficial to make a stronger point here.

The reviewer raises some interesting questions that could be addressed in future studies. Unfortunately, we do not have video during the BCI task. However, we now include an example video of animal behavior on the spherical treadmill, so that readers will have a clear understanding of the amount of locomotion our real-time locomotion tracker is capable of detecting. The section examining the direct neuron activity and 'strategy' was removed as suggested by Reviewer #1, and replaced with the new visual discrimination task. Taken together, we hope that our re-focusing of the manuscript addresses the reviewer's concern.

4. The process of Hebbian mechanisms is tested in relation to altering the visual response properties of the V1 population. However, again, not only the 6 selected neurons are altering their activity during the task. When ~80% of neurons increase their activity the process of Hebbian plasticity will span many different response types – so I am not sure why one would expect a shift in visual response properties. If the authors had, for instance, selected 6 cells that were known to have a specific initial tuning and required them to increase their activity to reach a frequency threshold, perhaps one would expect plasticity. However, I appreciate that the point of the study was to select ‘random’ populations of direct neurons – but the result that the authors found with such wide-spread modulation of activity in the entire imaged V1 population diminishes the importance of these neurons in relation to a clear Hebbian hypothesis.

In the original conception of this project, our expected outcome was that indirect neurons would broaden their orientation selectivity due to increased correlated activity during training, and potentially degrade stimulus representation outside of the BCI task context, assessed using decoding methods. However, we found that indirect neurons were resistant to perturbation, despite significant changes in activity at the time of success during the BCI task, as well as noise correlation that persisted outside of the BCI task.

Unfortunately, I do not have a nice suggestion for solutions here beyond further experiments to test the limits of what does and does not produce functional plasticity in V1. This seems beyond the scope of this study, but perhaps the authors can rework this to have less emphasis on the individual cells and more on the general population dynamics (in relation to cross-modal new learning not interfering with established functional properties).

Thank you for this suggestion, Reviewer #1 also indicated that including a second behavioral task could improve the impact of our work. The additional data and analysis now included revealed that at the population level, we were able to detect changes outside of a visual discrimination task. Interestingly, for the tracked neurons that remained visually responsive after visual discrimination learning, tuning was remarkably stable- similar to the BCI non-matched sensory task.

5. Throughout the manuscript, from the title to the conclusion, it should be made clear that the study is examining the effects of a cross-modal task. If the title is interpreted to mean a new visual skill in visual cortex (or any sensory matched area) then this would not be accurate, and from a plethora of previous literature (as cited by the authors) I suspect the outcome would be different. In general, there is not enough focus on emphasizing that these are effects from a non-matched sensory task in the title/abstract especially.

This concern is related to Reviewer #1’s suggestion to incorporate a visual discrimination task. We are pleased to now directly address this issue by including a sensory matched task (new Fig. 5) as well as the non-matched sensory task, and are sincerely grateful for the suggestion made here, it was the inspiration for including the visual discrimination task. The Title and Abstract were updated.

Specific points:

- Line 59/60: “our results demonstrate that a new skill can be integrated into existing networks...”. It is unclear as to how the acquired ‘skill’ is really ‘integrated’ into the V1 network here. The majority of neurons altered their activity, but this may have been brain wide. There did

not seems to be a clear difference between the activity of the direct neurons and that of the rest of the V1 population, indicating that the task elicited a general V1-wide gain in response. It is unclear how this correlates to the integration of a new skill into the V1 network. If the authors have a justification for this wording it should be made more explicit, otherwise it should be reworded.

Given that the new shuffle analysis of indirect neurons confirms that there was differential modulation among the direct neurons that was statistically distinct from indirect neurons in V1, and that the new noise correlation analysis in Fig2F revealed a persistent change in V1 outside of the BCI task context, we feel that this sentence is now supported by the evidence provided.

- Line 80/441: neurons were randomly chosen. Since changes in activity to drive the auditory cursor were bi-directional, was there no pre-determination of spontaneous activity level in neurons? Can you explain the 'randomness' of randomly choosing in more detail. Were they spatially spread out or just had to be isolated from other neurons but could be in close proximity to each other, had a certain baseline activity, etc. A sample FOV with real example instead of (or in addition to) the schematic in Figure 1A may be helpful. For instance, in Figure S2 the DN neurons all seem to have a very low spontaneous rate even in the first session. It is true that these neurons are still acting as controls so that the mouse can not simply increase the activity of all neurons to be successful in the BCI, but it seems unlikely that these particular neurons would be able to really be 'downregulated' with such low spontaneous rates? Can the authors present the spontaneous rates of the direct neurons across sessions and expand on their dynamics in relation to the general population in this regard.

Agreed, the description of how direct neurons were selected was missing some key information. That information is now included in the Results section, under heading: 'V1 neurons are capable of driving an optical BCI'. We now show a sample field of view (Fig. S3A) and provide movies of neural dynamics (Movies #2-5). We agree that it is unlikely that these particular neurons (DN neurons) could be downregulated. We apologize for the confusing description, and have now clarified.

- Line 88: it would be helpful to indicate explicitly in the results paragraph at this point that the target threshold is 15 kHz on the auditory cursor

Updated as suggested. Target threshold information is now added to the legend of Fig. S8.

- Figure S3 (line 116-117): it would be useful to see the equivalent data for the 15 kHz tone in Figure S3, since this is perhaps the most behaviourally relevant tone, including the comparable numbers that are presented in the Figure caption in S3B for 15 kHz. Additionally, since (B) shows data by trial, it would also be good to see the data averaged across trials for all cells (connecting the individual cells between baseline and stimulus conditions) for the 5 kHz and 15 kHz tones. Since the auditory systems shows relatively fast adaptation, I wonder if there would be differences if a smaller time window was selected for this analysis (e.g. first 1 second, for instance).

The data for the 15kHz tone was added to new Fig. S12, and we simplified our description of the results in the new Fig. S12 legend: 'Across the three mice, less than 2% (9 out of 495) of all neurons had a mean difference between baseline and stimulus greater than $0.1 \Delta F/F_0$. Of

those neurons, none were significantly modulated by any of the 6 auditory pitch stimuli (paired t-test, $\alpha=0.05$).’ The original analysis presented was quite confusing, this update is a better description of our data. Thank you for the suggestion of examining a shorter response window- we would have been excited to see an effect of BCI training on auditory responses, unfortunately we did not detect an effect. The following information is included here in the Rebuttal but not added to the manuscript: To address the whether using a shorter response window might detect an effect, all neurons that had a mean difference between baseline and stimulus greater than $0.1 \Delta F/F_0$ were visually inspected (13 neurons). None of these neurons had a clear onset of signal aligned to stimulus timing, nor were any of these neurons significantly modulated by any of the 6 auditory pitch stimuli (paired t-test, $\alpha=0.05$).

- Lines 116-117 state that V1 neurons were not modulated by the pitches, however, it should be noted in the results that 14/95 neurons were actually modulated by the tones, as stated in the S3 Figure caption. This is a small proportion, but not none. Of course it would be useful to have some data regarding how many neurons were modulated by the tones before training in these same mice, but presumably the study did not include this measure. As the authors state, auditory stimuli are not known to have a net excitatory influence on neurons in V1 – although they can exert inhibitory modulation in V1, even in untrained mice. If one follows the authors previous statement and assumes the number would be 0/95 pre training, this still represents an increase in the number of auditory modulated neurons in V1 after learning. Considering the paradigm is using modulation of a very small number of neurons to elicit behavioural changes, increase in even a small number of neurons here may contribute to an overall effect. The authors may still conclude that BCI performance was not likely to be improved by auditory sensitivity alone – but I am not sure it has to be an all or nothing effect in this regard. Especially since they show that 81% of neurons show a significant change in activity at the time of success, just because the experiment monitored 6 neurons, does not exclude the possibility that one of the other 81% was highly correlated with task success and was one of the 14/95 modulated cells. The authors state that the changes in these 14/95 cells were small on average, but this may also be partly because responses are averaged across 2 second windows. I think it would be more convincing to show more of this data. Perhaps the conclusion will not change – but the current analysis leaves me with many open questions.

The original description of the auditory analysis was confusing, we apologize. There were no neurons that were modulated by sound. Although 14 out of 495 scored as significant using the paired t-test, visual inspection revealed that none had any evidence of a stimulus locked response, an example neuron that scored as significant is shown in Rebuttal Fig. 1. Given that we did not observe any neurons that had auditory responses and the description of the data was clarified, we decided to not show additional data.

- Lines 124-125: this seems like a strong statement considering that such a large proportion of neurons in V1 had significant changes in their activity level. Presumably if Figure 2C is done with all non-direct neurons the result would be the same? Is there any evidence to suggest that BCI performance was improved as a direct result of the changes in direct neurons? One may randomly select any 6 cells post-hoc and come to the same conclusion, that changes in their activity could account for improved performance. If this is not the case, I would present more evidence for a unique role of the direct neurons in performance. Alternatively, I am not sure this is necessary to the point, as changes in the population in general may be reflected in the direct

neurons and vice versa – but the specific conclusions should be made more generic to apply to the whole population in that case.

Lines 124-125 were removed in the revision.

- Relatedly, it might be useful to examine the precise correlations between modulated cells in the rest of the FOV and the direct neurons. Do these cells have stronger pairwise correlations for instance? Can the selected direct neurons be used as a sort of 'hub' for activity change in the population during the task? Or perhaps they are simply reflective of it.

Thank you for this suggestions, we now include analysis of the modulated cells and their spatial position, as well as correlations among indirect-direct neuron pairs (Fig S11). A substantial fraction of indirect-direct neuron pairs was correlated; an example of an indirect neuron that was correlated with a direct neuron is shown in Fig. S11E. We went on to show that the spatial distribution of strongly correlated pairs was uniform across the field of view (Fig. S11).

- Lines 121-124: it is a bit difficult to interpret the difference between DP and DN modulation here. Since they both change in the same direction, simply based on which mouse is examined. The way the statements are formulated are a bit misleading to suggest that the DP and DN neurons were more or less relevant to learning in particular mice – but it seems they are not actually being modulated differently with respect to each other (DP vs DN populations). This is not an irrelevant point and the significance of this should be made more clear.

We agree, and believe that the new Fig. S3 clarifies this issue. Lines 121-124 were removed, as well as the panel that was referred to in these lines.

- Section 127-156: the results here presume that the selectivity of the direct neurons was also stable. This is discussed in further sections, but largely in relation to the indirect population. It is only digging in figure captions that one gets more information regarding the 6 selective direct neurons and still their properties are not fully presented across episodes as the indirect population are. In Figure 3B the authors should also show the direct neuron prior to and after BCI. Line 144: 39 had similar orientation tuning to a direct neuron, at which time point? Before or after BCI? Or either? Again, this does not change the conclusion that the BCI task did not enhance stimulus responses, I think this is a logical conclusion, but the stability/selectivity, or lack there of, for direct neurons should also be made clear throughout this and the following results section.

The update to Fig. 3B was made as suggested. All direct neuron values are included in Tables 1-4, and the text is now explicit in terms of which analysis include indirect neurons, direct neurons, or both.

- Line 472: the target was predetermined at the start of learning and did not change across sessions, but the spontaneous activity was redone each session – to calculate z-score. If the spontaneous activity drifted towards the target across sessions then absolute changes in activity would have become smaller to reach the same target. This could create a 'learning' curve simply with persistent changes in baseline activity across days. Can the authors comment on this possibility or present evidence to the contrary.

Indeed, changes in spontaneous activity would be a confound, because that would lead to a change in standard deviation, a term that we use to normalize activity across sessions. Care was taken to monitor this, and post-hoc we verified that this was not an issue. The information is now included in the Methods, under heading '*Brain Computer Interface Task*' and new Fig. S17. See also our response to Reviewer #3 below.

- Figure 1A: please define DP and DN in caption.

Updated as suggested.

- Supp Figure 1C: perhaps highlighting the first lick response following the onset would help to visualize the learning effect in the given example.

The reason that we included this plot was not clear. We added a note regarding the number of 'Go' trials that did not have a hit response.

- Line 389: I assume the system was a resonant-galvo system? Could the authors please indicate the frequency of the resonant scanner (8kHz, 12 kHz?) as this generally provides auditory noise during imaging.

Yes, the microscope using a resonant-galvo scanning mirror set, the resonant scanner frequency was 8kHz. The information is added to the Methods section.

- Line 402: Could the authors explicitly comment on how many neurons reached this criteria, i.e., the proportion of neurons removed during this process.

The median proportion of neurons that were tracked from the segmented pool was 40%. This information was added to the Methods.

- Line 420-421: 'Lick port was armed with reward...' Can the authors describe how this was done in more detail. Was the reward droplet pushed out only on Go trials? Would this not act as a cue in and of itself? Was this associated with an auditory cue (pump for reward spout) or perhaps olfactory cue was important here. I am not sure if it affects the outcome of the results, but specifics on the parameters here would be useful.

Thank you, we clarified. This is important, water was only released if the animal licked.

- Line 577: Response amplitude was computed as the average response across trials with a significant response... what if this measure is taken across all trials? As in, did the general modulation of activity in V1 lead to increased noise?

We are grateful for this suggestion and a similar comment made by Reviewer #1. We performed additional analysis, and have now updated Figure 4F. Response amplitude analysis now includes failures. And yes, interestingly, the fraction of neurons that have a decrease in amplitude is larger in the BCI condition compared to baseline conditions.

- Line 594: only neurons that were responsive to grating stimuli were included in the decoding.

The numbers across animals are listed in Figure 3F caption, but how did these numbers change across timepoints?

This analysis was removed; it did not add anything substantial to the narrative.

- Line 283: extra 'well as'

Updated.

Reviewer #3 (Remarks to the Author):

The manuscript "New skill acquisition does not perturb existing function in sensory cortex" from Kuhlman and colleagues describes how existing cortical responses in visual cortex react to BCI-based conditioning of neurons in the same area. This is a very pertinent and long standing question in the field and the authors have chosen an original experimental approach to answer it. We are in principle very enthusiastic about the study and at a first glance, the results are very intriguing.

We thank the reviewers for their enthusiasm.

However, while reading more carefully we discovered several important points which need substantial clarification. Please find below our suggestions and comments. We hope they help to improve the next version of this manuscript.

Our main concerns are about the implementation of the BCI learning paradigm. We have doubts that the animals actually learn a BCI task. The characterization of the visual response types seems to be done correctly, but we did not go over this part in detail. All the results hinge on the assumption that animals actually undergo a BCI task.

We agree that an important analysis of the indirect neurons was missing, and now provide the appropriate control analysis to establish that the mice did indeed learn the task. Thank you for your suggestions, the manuscript is much improved based on your comments.

Several important controls are currently missing (see details below). Also, it looks like some of the data has been omitted (additional days in Fig. 1C) and other data seems to be incorrect (Ca⁺ traces of DN3 in Supplementary Figure 2A are identical in both conditions, trial structure in the last 20 sec in DP1&2 of Supplementary Figure 2B should be identical to DP3, but it is not). These might be simple copy-paste errors, but they might also indicate that more careful work is necessary to claim robust learning. Finally, the relatively low number of animals also does not help to make a strong case.

These concerns are now addressed, our specific responses are below.

Behavior:

It has previously been shown that movement in complete darkness causes strong activation of visual cortex neurons. Since this behavior is carried out on an air-lifted ball, it seems crucial to closely monitor and analyze the behavior of the mice and compare their actions across

sessions. Much of the observations (increase in the DP and general increase in a majority of neighboring neurons) might be solely related movement parameters. The movement parameters might simply become more stereotypical and this induces stronger overall activity, which might lead (among others) to an increase in DP neurons. The authors will need to convincingly demonstrate that this is not the case.

Movement in 'baseline conditions' does indeed cause strong activation of visual cortex neurons, when the baseline conditions use a gray screen monitor¹⁻⁹. However, it is now appreciated that in darkness (the gray screen monitor is turned off), like we use here in the BCI task, there is little modulation of visual cortex neurons, and there is not a net increase of activity, just as many neurons are decreased as increased during locomotion¹⁰. Furthermore, we include new analysis showing that locomotion does not correlate with performance (new Fig. S9). The new analysis is presented in Results, heading 'V1 neurons are capable of driving an optical BCI'.

We would like to take the opportunity to add that including the DN neurons ensured that there must be *differential* modulation of activity among the direct neurons at the time the neural trajectory successfully crossed the target threshold, i.e. to reach the target threshold, the DP neurons were required to be more active than the DN neurons. Thus, the animal could not solve the task by uniformly applying an increase in activity level across all neurons in the imaging region.

Finally, the 1s of "not moving" seems much too short to assure that actual movements are not reinforced.

As noted above, there was no correlation of performance with locomotion, thus our evidence indicates that 1 second is sufficient.

Learning point:

It is not clear how the "learning point" is defined. If the learning point is defined by the crossing of a high performance level, then the slope of the regression will inherently be positive. If the learning point is defined by some goodness of fit (as indicated by the p-value?) then we don't see what this has to do with learning.

The learning point is not defined based on a goodness of fit; it is defined using linear regression. The first BCI session ID in which the linear regression was significant is referred to as the 'learning point'. The legend was updated to make this clearer. Reviewer #1 also raised the same issue.

As pointed out in the intro, we have the impression that the performance plot in Fig. 1C only shows a part of the actual sessions (the additional days are visible in Fig. 5B). How did the performance evolve after the day indicated by the "learning point"?

Correct, animals received BCI training up to the last visual stimulation imaging session. Performance on the last BCI session is shown in the new Fig. S10.

Unfortunately, Fig. 5B also does not allow us to determine this, since only the successful trials are shown. The day where the animal reaches the learning point seems to be accompanied with a clear change in the "strategy" which could reflect a performance decrease. Please clarify and show all the available data (overall performance for all days).

This figure was removed, based on the suggestion by Reviewer #1.

Quality of the Ca⁺ signals, definition of “neuronal drive” and choice of the neurons:

The few available calcium traces show very little activity (Fig. S2) and lots of noise (Fig. S2A: DN2, Fig. S2B Dn2, DN3). The z-scoring most probably adds to this distortion.

We agree that including more data examples is useful. In addition to the z-score examples in Fig. S2, we now include many examples of raw data (F/F_0): Figs. 2C, S4-7. This should alleviate these concerns.

As outlined above, a part of the data Fig. 2A seems clearly incorrect (Ca⁺ traces of DN3 in Supplementary are identical in both conditions, trial structure in the last 20 sec in DP1&2 of Supplementary Figure 2B should be identical to DP3, but it is not).

Thank you, indeed, there was a copy-paste error. The two errors were corrected.

Also, most of the “activity” in Fig. 5A looks more like noise with no actual events. How do the authors explain this?

The binned activity (3 imaging frames) used to drive the cursor was depicted in Figure 5. We agree, this was confusing, and removed.

A second issue is related to the definition of “neuronal drive” and “success”. Although the threshold is fixed in the beginning for each animal, the data are z-scored across sessions, which does not allow to compare the neuronal activity across sessions. This is highly problematic.

Our normalization procedure allowed for assessment of performance across sessions, by necessity some form of normalization is required in such an experimental design to control for potential drift in instrument sensitivity. We agree, given z-scored activity was the signal used to drive the device, it is important to confirm that changes in the denominator used to normalize (in our case, standard deviation) did not change across sessions. This was carefully monitored as the experiment progressed and we now describe how this issue was controlled for in the Methods, heading ‘*Brain Computer Interface Task*’:

A decrease across sessions in the standard deviation of DP neuron raw fluorescence signal during the 3-minute spontaneous baseline recording would lead to an inflation of DP neuron activity that could increase the frequency of threshold crossings. Likewise, an increase in the standard deviation of DN neuron raw fluorescence signal across sessions during the 3-minute baseline could result in an increase in the frequency of threshold crossings. To examine whether this was an issue for any one of the four mice, we assessed the correlation between the standard deviation of the 3-minute baseline and session number. None of the DP neurons had a significant negative correlation ($R < 0$ and $p < 0.05$) between standard deviation and session number, nor did any of the DN neurons have a significant positive correlation ($R > 0$ and $p < 0.05$) between standard deviation and session number. Thus, drift across sessions in the standard deviation of baseline fluorescence did not contribute to increased threshold crossings.

We now include an example DP neuron and DN neuron, across sessions, to show that the baseline 3-minute spontaneous baseline standard deviation is consistent across sessions (new Fig. S17).

Furthermore the noise level of the z-scored data seems to change within a session (DN neurons in S2A&B) which is very weird. How can this be explained if the imaging conditions were stable and Z-scoring constant?

We are unsure exactly what the reviewer is referring to, upon closer inspection we realized the line thickness (line stroke) in the Adobe Illustrator file was set to 0.667 for DN2 in the LP session, whereas all other line thickness is set to a value of 1. This is now corrected. To address the general concern that within session instability of noise contributes to apparent increases threshold crossings across sessions, we estimated 'noise' by quantifying the standard deviation of the residual signal (calcium events were removed). Within-session change in the standard deviation of the residual signal was similar for all sessions in each of the 4 mice (new Fig. S17B and Table 6).

Finally the initial threshold definition (30% of all trials successful) is also problematic since with extremely low activity levels, noise could trigger threshold crossings. How can the author demonstrate that this is not the case?

Taken together, the method used by the authors seems to allow for the slightest change in activity in a single neuron (see for example: Fig. S2A cell DP3 @ 160 sec) to trigger a "successful" trial. This does not make any sense and contrary to most other BCI studies where either robust single cell activity or strong combined activity of multiple neurons is necessary.

In addition to the z-score activity examples, we now provide movies (Movie #2-5 of raw data that include target threshold crossings in which the activity of DP neurons clearly are driving target threshold crossings; the corresponding extracted F/F_0 from the movies is shown in Fig. S3.

In addition, it is not clear how the authors exactly chose the DP and DN neurons. They describe it as random. This is obviously not ideal for many reasons. In Fig. S2A&B all DN neurons seem very silent. How come? There could be an inherent bias (see suggestions for controls below). Better choice definitions or at least quantifying the activity rates of all neurons might help to clarify this point.

We now describe the process of selecting direct neurons more clearly in the Results section, under heading 'V1 neurons are capable of driving an optical BCI'.

Controls

The following controls might help to show more convincingly that the animals actually "learn" and the chosen cells or that feedback play a role:

The authors need to show that learning is specifically linked to the chosen neurons and not just due to random fluctuations, a biased choice or artifact induced by z-scoring. One easy way to do this might be to choose arbitrary sets of 3 neurons from the existing data set of active neighboring neurons, apply the same analysis methods and assess in how many cases "learning" would occur.

Thank you for this suggestion. We now include analysis for each of the 4 mice in which chance level of performance is derived from the activity of indirect neurons as recommended (new Fig. S8). This was an important improvement to the manuscript.

Furthermore the timing of the neighboring neurons should be analyzed in relation to the peak of the DP neuron or to reward delivery. This might provide precious cues of conditioning locations.

This issue was raised by Reviewer #2 as well. We now include analysis of the modulated cells and their spatial position, as well as correlations among indirect-direct neuron pairs (Fig. S11). A substantial fraction of indirect-direct neuron pairs was correlated; an example of an indirect neuron that was correlated with a direct neuron is shown in Fig. S11E. We went on to show that the spatial distribution of strongly correlated pairs was uniform across the field of view (Fig. S11).

The authors also need to show that the “learning” can not be observed in animals where the rewards are delivered based in a non-BCI setting. For example during fixed-pattern, random or played back of previously recorded activity. Animals would need to be trained for >10 day with this uncoupled activity to see if any increased activity starts to occur within the selected set of neurons. “Playback” would ideally be chosen based on the neuronal activity from existing “learning” sessions to match the timing and the number of rewards.

We now include one control animal as suggested (New Fig. S8B). Prior to the start of the experiment 6 ‘fictive’ neurons were selected, auditory tones generated by mouse #1 were played to the control mouse; the 6 fictive neurons were not coupled to the auditory cursor. Performance of the 6 fictive neurons did not improve. Given that the results of the indirect neuron shuffle analysis were so clear, and considering the comments of the other reviewers, we felt adding more than one control animal receiving ‘playback’ was not as important as adding more animals in the new visual discrimination task.

Other general issues:

It would be helpful to show the activity rate for all other neighboring neurons (Fig.1 & 5) and determine if they also change in a similar way.

Showing more representative unprocessed /processed data across the entire session or across sessions would help to illustrate the overall quality of the imaging data and evolution of the activity.

We agree that showing more examples of raw data is important, and now include more examples: Fig. 2C (the entire session is shown as suggested), and Figs. S4-7 include more examples from every mouse.

Showing representative animal behavior videos would be very helpful (see above).

An example of animal behavior on the spherical treadmill is now provided.

More detailed issues:

Fig.1 :

the neuronal drive is plotted on an arbitrary axis & on a pitch level axis. Why is the second pitch level representation slightly different? It only makes sense to show actual auditory feedback on a pitch axis.

The actual auditory pitch is now shown.

Fig. 2:

Fig. 2 clearly shows that there is an overall increase in the DN neurons across sessions. How is

that compatible with the idea that they actually should decrease in activity within a session? This might need to be discussed. What does the dashed vertical line stand for? How can DN activity be negative at the start??

These panels were removed, given multiple reviewers found them unclear.

Fig 3 :

The general message seems to be that there is no change. However, we can only see 6 DP because they were the only ones responding to specific visual stimulation. It would be nice if all neurons used for the BCI were characterized before training. What is the function of the remaining 6 DP neurons? What about DN neurons?

The amplitude scale in Fig. 3B might work better from black-blue-yellow

Four tables (Tables 1-4) detailing the changes in all direct neurons are now provided. Based on input from local colleagues, we decided to keep the current color scheme.

Fig 4:

The left panels of Fig. 4E clearly shows that the signal correlation clearly becomes weaker after BCI. However, the summary plot on the right does not reflect this. Everything is centered around 0. This is most likely because the low correlation in all other neurons dominates the analysis. There might be a better way to analyze this point?

We examined this possibility and found that there was no net shift (Rebuttal Fig. 2).

Fig. 5.

Fig. 5A all calcium traces look bizarre and noisy. How come? Please show longer traces or more trials.

The binned activity (3 imaging frames) used to drive the cursor was depicted in Figure 5. We agree, this was confusing, and removed. We now show more trials and longer traces (Fig. 2C and Figs. S4-7). Fig. 2C is in heatmap format (given there are many more trials shown, the heatmap format is a more effective display).

In A, the target dashed threshold line is not visible enough
Legend of B currently reads "Stragagy"

These elements are removed in the new version.

FigS1:

Typo in the legend for B. Probably means 81% instead of 0.81%? It might also be important to show the number of days it takes to get this the initial association.

Thank you, we corrected the % value, and added the requested information to the legend.

FigS2: See multiple comments above.

We hope these comments help to strengthen this otherwise interesting study.

Thank you, indeed addressing the comments provided greatly improved the manuscript.

Daniel Huber & Antoine Philipides

Rebuttal Figure 1. Example neuron that scored as being significantly modulated using the original paired t-test method comparing baseline and response windows. Left, values from individual trials plotted as in new Fig. S12B. Right, traces (gray) of the data plotted on the left, the mean across trials is shown in black. Note that there is no evidence of a stimulus-locked increase in activity. The mean difference in activity ($\Delta F/F$) between the baseline and response window was 0.0061 in this example neuron.

Baseline condition

BCI condition

Signal correlation, VS baseline 1

Signal correlation, VS baseline 2

Rebuttal Figure 2. Signal correlation is not weaker after BCI training. Pairwise signal correlation for each BCI mouse. The mean and 1-standard deviation of pairs with correlation values higher than 0.05 is indicated. Data points in gray (below the dashed line) are less than 0.05. Note, there was no net decrease in the magnitude of the signal correlation in the BCI condition. Wilcoxon rank-sum *p* values are shown.

Rebuttal References

1. Vinck, M., Batista-Brito, R., Knoblich, U. & Cardin, J. A. Arousal and locomotion make distinct contributions to cortical activity patterns and visual encoding. *Neuron* **86**, 740–754 (2015).
2. Pakan, J. M. *et al.* Behavioral-state modulation of inhibition is context-dependent and cell type specific in mouse visual cortex. *Elife* **5**, (2016).
3. Mineault, P. J., Tring, E., Trachtenberg, J. T. & Ringach, D. L. Enhanced Spatial Resolution During Locomotion and Heightened Attention in Mouse Primary Visual Cortex. *J. Neurosci.* **36**, 6382–6392 (2016).
4. Bennett, C., Arroyo, S. & Hestrin, S. Subthreshold mechanisms underlying state-dependent modulation of visual responses. *Neuron* **80**, 350–357 (2013).
5. Niell, C. M. & Stryker, M. P. Modulation of visual responses by behavioral state in mouse visual cortex. *Neuron* **65**, 472–479 (2010).
6. Dadarlat, M. C. & Stryker, M. P. Locomotion Enhances Neural Encoding of Visual Stimuli in Mouse V1. *J. Neurosci.* **37**, 3764–3775 (2017).
7. Reimer, J. *et al.* Pupil fluctuations track fast switching of cortical states during quiet wakefulness. *Neuron* **84**, 355–362 (2014).
8. Polack, P.-O., Friedman, J. & Golshani, P. Cellular mechanisms of brain state-dependent gain modulation in visual cortex. *Nat. Neurosci.* **16**, 1331–1339 (2013).
9. Ayaz, A., Saleem, A. B., Scholvinck, M. L. & Carandini, M. Locomotion controls spatial integration in mouse visual cortex. *Curr. Biol.* **23**, 890–894 (2013).
10. Dipoppa, M. *et al.* Vision and locomotion shape the interactions between neuron types in mouse visual cortex. *Neuron* **98**, 602-615.e8 (2018).

Sincerely

Sandra Kuhlman on behalf of all of the authors

REVIEWER COMMENTS

Reviewer #1 (Remarks to the Author):

The authors have addressed systematically and thoroughly the Reviewers' concerns. The result of this process is a manuscript that is impressively strengthened. In particular, the new group of mice undergoing a visual discrimination task is providing a key comparison. The shuffling analysis to establish the success of the BCI is also a very welcome addition. Moreover, the reorganization of the figures and extensive work on the text have led to a much more understandable story, notably it is much clearer in the figures which are the different groups of neurons analyzed (direct, indirect, responsive, similarly-tuned, etc.).

- Given that the analysis in Fig S8 is really important to ensure there is BCI learning, it could be incorporated in Fig 1. (at least the top row).
- I still didn't really understand what the darkness experiment was adding to the study. It appears in one sentence at the end of the Introduction but it's not obvious why it's relevant, and again at the end of the Results. It is up to the authors to decide if they want to keep it, but I suggest justifying the added value of this data relative to the main claim of the paper in more depth.
- line 191, somehow this argument sounded a bit weak. Because of the particular selection of the groups (the threshold should be reached enough so that the mouse gets rewards), the DN neurons are probably less active and less modulated than the DP or even indirect neurons. So if DN neurons are almost at 0 all the time, a global multiplication of activity across the network could lead to target.
- line 204, paragraph on the correlated neurons. Somehow it was difficult to understand what was the point of this analysis here. The fact that it does not even refer to a main figure does not help. I got lost as to why this data is presented. It should at least be justified better at the beginning of the paragraph.
- line 371, a BCI task does not have to be multimodal, in fact it can even be blind = no sensory feedback at all.

At this time, an exhaustive list of minor corrections is unfortunately outside of the scope of what I can provide. Here are a few.

- I think the Fetz reference is still missing, please check
- Some sentences should be cut in several, for example: Abstract lines 14-18, lines 25-28; Page 3 line 108-110; Page 4 line 133
- line 142, do you really mean with replacement? Can you get the same neuron twice in a group of both in the DN or DP? I suppose not.
- line 286, there is no panel F anymore. Also I think just panel C is concerned by the sentence?
- Fig2, strategy is misspelled (also in some Supplementary Figs), relateve, and neruons
- Fig3, the labels are confusing. "Tuning similarity..." implies that this is what is measured, and then we see a measure in degrees below. And "prior to BCI training" is not clear here, it could imply between baseline 1 and 2.
- line 319, clarify how you can have a median of 353 values that is then pooled over mice and giving an SEM.
- line 396, please say what this control cohort was subjected to.
- line 403, orientated

Reviewer #2 (Remarks to the Author):

While the authors have made significant improvements to the manuscript in clarity and methodological descriptions, I have some remaining concerns with issues that were brought up previously (the lack of tracked changes also made the manuscript revisions hard to follow in some cases) as well as comments to the new experimental data that was added.

Major concerns:

I would encourage the authors to make better use of key Supplementary Figures by moving some to the main Figures (some suggestions detailed below). A larger portion of the text in the results now refers to supplementary Figures only (lines 163-240).

It is somewhat unclear what the specific hypothesis for the new visual discrimination studies was in relation to the earlier BCI results and what the implications/conclusions of this are specifically in relation to the non-matched sensory task. A clearer matched vs non-match sensory task elaboration to set-up this experimental design and flow would be beneficial in this regard. The last sections feel disjointed to the BCI studies and while I see some merit in the general approach, as presented they are somewhat confusing (see also point 9).

General comments:

1. Figure S2: This Figure is very informative and instantly provides an easy to digest conceptualization as well as convincing traces. I would similarly colour code the DP and DV (purple/orange) and include it in a main Figure (either main Fig 1 or, perhaps better, around Fig 2C, and in relation to lines 162). The Figure panel could be made significantly more compact using a single activity vertical scale bar (as is already done with the time scale bar) instead of a y-axis scale, one behavioural scale bar (green/blue/grey) along the bottom that applies to all traces, and cell images put to the left of each trace to save on vertical footprint. Ideally, similar supplementary Figures could also be provided for mouse 2-4 then.
2. Equally, Figure S8 is also an informative addition. I would advise adding the plot from mouse 1 of Figure S8A to the main Figure 2, which already shows many examples from this mouse.
3. Figure 2C: I am not sure there is much need for the plots to extend to -4 s before target. Is there useful information provided in -4 to -2 for instance? The pertinent change appears to happen from -2 to 0 and perhaps -3 to -2 could be included to establish more baseline activity, but I would also be more interested in seeing a 0 centered onset to also get an idea of what the cells do following the pitch change that occurs at 0. I think a 0 centered -2 to +2 may be informative.
4. Figure 5: Why must the pool of neurons tracked is different between baseline and BCI conditions? The example FOV in 5B appears to be congruent and it should be possible to track the same neurons across time. Why are some overlapping and others not. In the least, the ones that can be reliably tracked across all sessions should be reported, even if compared as a separate subpopulation in a supplementary figure. Relatedly: Line 55: 'allowed the same neurons to be tracked during training' – this statement may add to any confusion about the different populations in Fig 5.
5. Figure S1D: legend describes that all 6 mice had d' of at least 1.68 on the last session, but in the text (lines 122-3), 5/6 mice were said to have learnt the task and proceeded to BCI training. What was the criteria here? It would be useful to add this to Figure legend S1D, in the least.
6. Figures S4-7: min, median, max change trials look comparable – were there any qualitative/quantitative differences in the basic descriptive statistics here across session 1 to last session?
7. Figure S11: why are there fewer Indirect-DP pairs than Indirect-DN pairs? If there are 3 DPs and 3DNs and X indirect neurons in the same FOV, why would these be different? Maybe useful to include indirect-indirect pairs as a 'baseline' information of general change across sessions in the population. Also, the significance of the related paragraph in the results (starting line 204) is somewhat lost. I think the new data is useful, but the implications of the findings should be refined.
8. The control signal threshold is now reported (albeit only in the FigS8 legend) as mouse#1-4: 15,8,7 and 12 respectively. How was this determined for each mouse?
9. Figure 5: the difference in Fig 5D (visual discrimination training) and Fig S15C is entirely unclear and seem contradictory. In Fig 5D – in addition to the decrease in 90 degrees - there is clearly also an increase in the fraction of responsive neurons to the orientations around 0 and 180 – presumably the vertically rewarded orientations. What is the data in Fig S15C representing? The results paragraph starting at line 400 is equally confusing in this regard. Previous studies (e.g. Poort et al., 2015; Henschke et al., 2020, and others) have shown that there is an increased representation of selective neurons in V1 to task-relevant gratings during goal-directed behaviours, and increased discrimination

between two task-related gratings (Poort et al., 2015). It is unclear how the authors interpret that their results relate to these previous findings. Figure 5D seems to indicate that their data show something similar (perhaps somewhat more complex with the orthogonal [90 degree] representation decreasing dramatically, although this was also evident in the Henschke et al., 2020 results), but this is not discussed (in results or discussion). Similarly, the decoding results are unclear. Is Fig 5E because the decoder is not sensitive enough to changes that are seen in 5D? What are the decoding properties specifically for task-related orientations?

10. Line 110: a brief description of what the VS baseline sessions consisted of would be useful here. The word 'tuning' is mentioned at some point in the paragraph, but beyond that there is no indication of what conditions these sessions were recorded under.

11. Line 127: it seems very important to mention here that it increases in their "summed" z-score activity that drove the cursor – in this way, one DP cell can dominate as is indicated in Fig 2. This is not clear until much later and if it was the case that all 3 would need to reach a certain z-score threshold individually, this would presumably lead to a very different outcome.

12. The description of how cells were selected is improved, however, I have some remaining questions. Why 12-18? This seems arbitrary (was this based on the step-size of the pitch increase per threshold crossing???) If so, this should be mentioned), but also seems like a wide range. If an arbitrary number is picked why not a single number that can apply for all mice. Also, the thresholds ranging from 6-15 is quite a large range, and equally arbitrary. It seems a bit strange to have both of these values as a range. What was chosen between a group of 6 cells with threshold 7 with 15 crossings and a group of 6 cells with threshold 10 and 14 crossings. Surely there was more than one group of 6 combinations of neurons in a FOV of that fell into these wide ranges? Then was a visual inspection applied to pick the final group?

13. It was previously requested to define the learning point more clearly. The description in the legend helps, but the text (lines 152-157) remains a bit unclear. I also do not think the comment from reviewer 3 was adequately addressed in this regard: 'How did the performance evolve after the day indicated by the "learning point"?' I would think it would be possible to include all sessions from 1 up to the last visual stim imaging session in the plots for Fig 1C and to indicate with the arrow (or dot, etc., and to keep the associated red regression line) which was the learning point session. This way, the whole learning trajectory is shown (as is done with the new task in Fig 5B), and it is also clear that the designated LP (Fig 2 etc.) and the last session (Fig S10) are used as example days along the training continuum. In my experience mice often show variable performance across sessions even after they may have reached a learning criterion, this would be useful information for the reader and for reproducibility.

14. I appreciate the data value added by the heatmaps (Fig 2C, S11, D, E, etc) – but they are somewhat difficult to parse visually. The scalebar may be too large and there seem to be abrupt transitions between zero and about 1-2. For example, the map goes from black (0) to dark pink (~1-2) within a single data point on the x-axis. This makes abrupt transitions between delta of 0-2 visually highlighted - which are irrelevant to the point. The important transitions are the higher ranges (light pink/white) which take some time to visually pick out. Either a change in the colour map (should not be strictly necessary) or an alteration of the scale (perhaps better) seems warranted to fix this. Visually, this works much better in Figure 3A for instance.

15. Why were both Emx1cre and SLC17a7cre mice used? I did not see a description of when these lines were used, i.e., for which experiments. Are there systematic differences between the populations of excitatory neurons labelled with these two lines?

16. Lines 174-176. This justification sentence reads to me as if there was a misinterpretation of the aim of the playback control. The point is not anything specific to sessions >10 (this is just a necessity to have the same amount of training as is present in the most trained experimental mice). Although this control would be relevant to type 1 errors, in my mind, a more straightforward description would be to control for any systematic increases in activity that may spontaneously occur with exposure to the task elements (reward, etc.). Please check this.

17. Lines 186-190: The authors rely on Dippopa et al., 2018 to make the point that locomotion during darkness does not modulate neurons in V1 to a great extent. However, they also have this data in their own dataset presumably. Can they not provide evidence for/replicate this finding in their own

mice/FOVs? For instance, with a simple index of activity modulation (active vs inactive periods as indicated in Fig S9) during the BCI task in the dark and visual stim baseline days in their populations (as in Fig 4). According to their methods this locomotion data was removed for tuning calculations, but there is no reason why they can not compare general stim (without tuning) vs dark using this same data. For Figure S9, locomotion may not be significantly correlated with overall performance, but could still be correlated to the activity of an individual DP (or DN) neuron (or the indirect populations) – since the performance is dictated by the sum of activity. This is still an important point on the population activity and single cell level.

Minor comments:

- Line 53: specify genetically encoded 'calcium' indicator
- The sentence starting line 158 is confusing. The end of the sentence could use rewording.
- Line 361: pdf missing degree symbol
- Line 208: missing 'correlated'
- Line 231: here, I would refer to the appropriate Figure panel for evidence of this improvement.
- Lines 251 to 252: the word excitability is again used here instead of activity (see previous comment from reviewer 1). Please check throughout manuscript.
- Fig 2D & E: where are the 'no change' neurons from D represented in E?
- Line 257: include relevant stats at the end of the sentence (similar to line 264). What are the stats in the Fig 2F in relation to this?
- Line 259: check wording... indirect-direct indirect
- Lines 339 and 407: presumably S15 is meant here? And related: paragraph starting line 430: S16.
- Fig S15A: add mouse as an x-axis label in A
- Line 461: 'assess' grammar

Reviewer #3 (Remarks to the Author):

The manuscript has much improved and we appreciate the detailed answers and new figures which allowed us to better assess this study. The new data of the visual discrimination task in the second part is also an important improvement. The selection process of the conditioned neurons is clearer now and the shuffling analysis is a nice control.

Regarding the robustness of the BCI learning and how it supports the overall conclusions, the clarifications and details provided have unfortunately rather increased our concerns. We currently have the following interpretations:

During the BCI learning process, only a small fraction of the 24 direct neurons actually change their firing pattern (S4-S8, old figure 5). The authors admit that it is mainly single (or maximum two) DP neurons that show actual changes. DN neurons never seem to change with learning (in addition, the selection process clearly seems to favor silent DN neurons). According to table 2 and 3, this means that this entire study relies on a total of ~6 DP neurons (across 4 mice) which modify their activity during learning. Since only a fraction of those (~3-4 neurons) actually show visual tuning, we end up with a highly underpowered data set. We don't think it is correct to consider all 24 direct neurons (half of which are DNs and thus show very little activity during conditioning, and the other half which show small changes) for the analysis, but only the ones that actually change during conditioning. Comparing neurons which have not undergone BMI related changes with the ones that have not been conditioned does not seem an adequate way to ask if conditioning impacts existing functions and interaction. Ideally, the conditioned neurons should have been chosen from a pool of neurons with intermediate baseline activity. This way the DN could have been conditioned to actually decrease and the DP neurons to increase. Using single cell conditioning would have been another alternative. As pointed out in our first assessment this would mean adding many more experimental animals to the study.

In order to convince the reader that the BCI learning point leads to a stable plateau performance

(similar to the one in Fig. 5B for the visual behavior) we asked to show the performance of all sessions following the learning point (old figure 5) for all 4 mice. New figure 10 does not address this point as it only compares the first with the very last session. If the behaviors do not reach a plateau, then it is questionable if the assessment of visual function a few days after the learning point is meaningful. Our concerns thus still remain. Finally, why is a binomial test used for this paired comparison?

According to our reading, the new figure S9A also shows that mouse 1 has a positive correlation of performance and running. We can thus ask if the observed changes are not simply due to an increase in movements in this mouse? Removing this particular mouse would further remove another one of the four remaining neurons from the analysis...

Playback control: the "fictive" neurons should not be chosen not randomly, but according to the selection criteria described for the BCI mice. In addition, we don't think that a single mouse is sufficient for this type of control. The analysis carried out in S8 will also be important for these control mice.

Minor comments:

Fig. 2A: Strategy instead of "Stratagy"?

Fig5E: training instead of "traning"?

Line 286: there is no Fig 3F?

The first sentence of the abstract might need revision, since the computation of abstract properties would not be the first thing that comes to my mind when thinking about learning a new skill.

There seems to be a mix-up in the figure numbering S14, S15, S16 on lines 399-446?

Taken together, we are very sorry to not be more positive despite all the efforts the authors have put into the revision of the current manuscript.

Daniel Huber & Antoine Philippides

We thank the reviewers for their thoughtful and constructive comments. We addressed all of the concerns raised by Reviewer 1 and 2. Reviewer 3 had concerns regarding performance across all BCI sessions and the statistical test used to determine significant changes in BCI performance; these concerns were addressed.

In consultation with the editor, we identified all of the concerns raised by Reviewer 3 that had the potential to impact the main conclusions of the manuscript, and addressed all of those concerns. We are grateful for all of the comments by Reviewer 3. The 3 issues raised that were deemed to not impact the main conclusions were useful because they highlighted a need for us to clarify the rationale and main conclusions of the manuscript, which we did in this revision. The paper is much improved.

Reviewer #1 (Remarks to the Author):

The authors have addressed systematically and thoroughly the Reviewers' concerns. The result of this process is a manuscript that is impressively strengthened. In particular, the new group of mice undergoing a visual discrimination task is providing a key comparison. The shuffling analysis to establish the success of the BCI is also a very welcome addition. Moreover, the reorganization of the figures and extensive work on the text have led to a much more understandable story, notably it is much clearer in the figures which are the different groups of neurons analyzed (direct, indirect, responsive, similarly-tuned, etc.).

- (1) Given that the analysis in Fig S8 is really important to ensure there is BCI learning, it could be incorporated in Fig 1. (at least the top row).

We agree and updated Figure 1E as suggested. See also our response to Reviewer 2, comment #2.

- (2) I still didn't really understand what the darkness experiment was adding to the study. It appears in one sentence at the end of the Introduction but it's not obvious why it's relevant, and again at the end of the Results. It is up to the authors to decide if they want to keep it, but I suggest justifying the added value of this data relative to the main claim of the paper in more depth.

This experiment was removed, as suggested.

- (3) line 191, somehow this argument sounded a bit weak. Because of the particular selection of the groups (the threshold should be reached enough so that the mouse gets rewards), the DN neurons are probably less active and less modulated than the DP or even indirect neurons. So if DN neurons are almost at 0 all the time, a global multiplication of activity across the network could lead to target.

Thank you for raising this issue. We addressed this point by performing additional analysis. We found that DN and DP neurons are equally modulated by locomotion (new Figure S12 A,B). See also our response to Reviewer 2, comment #17.

- (4) line 204, paragraph on the correlated neurons. Somehow it was difficult to understand what was the point of this analysis here. The fact that it does not even refer to a main figure does not help. I got lost as to why this data is presented. It should at least be justified better at the beginning of the paragraph.

Better justification is now provided: The majority of indirect neurons are correlated with at least one direct neuron at the time of threshold crossing, in the first BCI session as well as the last BCI session. This is important because it provides a premise for considering the possibility that tuning of the indirect neurons is altered by BCI training, given the frequency of threshold crossing increases with training. See also our response to Reviewer 2, comment #7.

- (5) line 371, a BCI task does not have to be multimodal, in fact it can even be blind = no sensory feedback at all.

Agreed, the sentence was updated to: 'The BCI task used in this study was multimodal in nature.'

At this time, an exhaustive list of minor corrections is unfortunately outside of the scope of what I can provide. Here are a few.

- I think the Fetz reference is still missing, please check

The reference is now added to the Introduction, thank you.

- Some sentences should be cut in several, for example: Abstract lines 14-18,

We feel that the information in lines 14-18 is necessary, for the following two reasons. One, it was suggested in a previous review round that we use the term 'non-matched sensory task'- thus it needs to be clear that there is an auditory feedback cue in the BCI paradigm. Two, it was suggested in a previous review round that information regarding critical task details be included in the abstract.

lines 25-28;

Updated to: However, visual discrimination training did increase the rate of representation drift.

Page 3 line 108-110;

It was noted in a previous review round that the design was unclear. This sentence was added to clear up that concern, therefore we prefer to keep.

Page 4 line 133

Updated to: The animal had 10 seconds to reach the target, if the target was not reached in that time period, the trial was scored as a failure.

- line 142, do you really mean with replacement? Can you get the same neuron twice in a group of both in the DN or DP? I suppose not.

We clarified this issue. On each iteration in which a set of 6 neurons was tested, all neurons were sampled from the entire available pool, including neurons that were selected from previous iterations. See also our response to Reviewer 2, comment #8.

- line 286, there is no panel F anymore. Also I think just panel C is concerned by the sentence?

Updated as suggested.

- Fig2, strategy is misspelled (also in some Supplementary Figs), relateve, and neruons

Corrected.

- Fig3, the labels are confusing. "Tuning similarity..." implies that this is what is measured, and then we see a measure in degrees below. And "prior to BCI training" is not clear here, it could imply between baseline 1 and 2.

Thank you, updated to ' $\leq\Delta 30^\circ$ ' and ' $>\Delta 30^\circ$ '.

- line 319, clarify how you can have a median of 353 values that is then pooled over mice and giving an SEM.

Updated to: The median change in orientation preference during the bassline condition was 4.6° (interquartile range = $2.0 - 12.5^\circ$). Similarly, the median change was 4.6° (interquartile range = $1.8 - 10.2^\circ$ in the BCI condition (**Fig. 5C**). Thus, the distribution of changes in orientation preference in the BCI condition was indistinguishable from that of the baseline condition. Analysis of the distributions on an animal-by-animal basis confirmed that orientation preference was not destabilized by BCI learning (**Fig. 5D**).

- line 396, please say what this control cohort was subjected to.

Mice trained in the visual discrimination task were compared to a separate cohort of control mice that did not receive visual discrimination training (see Table 1 for details).

- line 403, orientated

Updated.

Reviewer #2 (Remarks to the Author):

While the authors have made significant improvements to the manuscript in clarity and methodological descriptions, I have some remaining concerns with issues that were brought up previously (the lack of tracked changes also made the manuscript revisions hard to follow in some cases) as well as comments to the new experimental data that was added.

Major concerns:

I would encourage the authors to make better use of key Supplementary Figures by moving some to the main Figures (some suggestions detailed below). A larger portion of the text in the results now refers to supplementary Figures only (lines 163-240).

It is somewhat unclear what the specific hypothesis for the new visual discrimination studies was in relation to the earlier BCI results and what the implications/conclusions of this are specifically in relation to the non-matched sensory task. A clearer matched vs non-match sensory task elaboration to set-up this experimental design and flow would be beneficial in this regard. The last sections feel disjointed to the BCI studies and while I see some merit in the general approach, as presented they are somewhat confusing (see also point 9).

To alleviate the 'disjointed' feeling, we performed the same fixed-decoder analysis on the BCI-trained mice and clarified the confusing visual discrimination results. Our detailed responses are below.

General comments:

1. Figure S2: This Figure is very informative and instantly provides an easy to digest conceptualization as well as convincing traces. I would similarly colour code the DP and DV (purple/orange) and include it in a main Figure (either main Fig 1 or, perhaps better, around Fig 2C, and in relation to lines 162). The Figure panel could be made significantly more compact using a single activity vertical scale bar (as is already done with the time scale bar) instead of a y-axis scale, one behavioural scale bar (green/blue/grey) along the bottom that applies to all traces, and cell images put to the left of each trace to save on vertical footprint. Ideally, similar supplementary Figures could also be provided for mouse 2-4 then.

Similar supplementary Figures were added for mouse 2-4. We include the y-axis so that it clear that the baseline z-score is stable at a value of 0. Former supplementary Figure 2 is now incorporated into a main figure as suggested.

2. Equally, Figure S8 is also an informative addition. I would advise adding the plot from mouse 1 of Figure S8A to the main Figure 2, which already shows many examples from this mouse.

Thank you for this suggestion. We added the shuffle analysis shown in previous Fig. S8 to a main Figure, Figure 1E, following Reviewer 1's suggestion.

3. Figure 2C: I am not sure there is much need for the plots to extend to -4 s before target. Is there useful information provided in -4 to -2 for instance? The pertinent change appears to happen from -2 to 0 and perhaps -3 to -2 could be included to establish more baseline activity, but I would also be more interested in seeing a 0 centered onset to also get an idea of what the cells do following the pitch change that occurs at 0. I think a 0 centered -2 to +2 may be informative.

We updated Figure 2C as suggested, and include -2 to +2, centered on 0. Thank you, the data display is improved.

4. Figure 5: Why must the pool of neurons tracked is different between baseline and BCI conditions? The example FOV in 5B appears to be congruent and it should be possible to track the same neurons across time. Why are some overlapping and others not. In the least, the ones that can be reliably tracked across all sessions should be reported, even if compared as a separate subpopulation in a supplementary figure. Relatedly: Line 55: 'allowed the same neurons to be tracked during training' – this statement may add to any confusion about the different populations in Fig 5.

We updated the sentence on line 55 to: 'The use of a genetically encoded calcium indicator allowed neural activity of the same neurons to be tracked before and after training.' Correct, the planes are congruent across imaging sessions. However, due to slight changes in vascular diameter and structure (vascular itself has some dynamics in intact systems), there will always

be some neurons in which the confidence of tracking is lower than others. Although many papers in the field simply use the centroid of the segment as an indication that it is the same neuron- which would yield a higher number of tracked neurons- and then confirm the neuron is tracked when response properties are the same, in our study that would be a circular argument. For that reason, we required a 75% overlap of segments. After careful consideration, we prefer to keep the analysis as it is. As recommended, the sentence on line 55 was updated to more clearly represent the analysis that was performed.

5. Figure S1D: legend describes that all 6 mice had d' of at least 1.68 on the last session, but in the text (lines 122-3), 5/6 mice were said to have learnt the task and proceeded to BCI training. What was the criteria here? It would be useful to add this to Figure legend S1D, in the least.

Thank you for noticing this. The 'playback control' mouse was not correctly accounted for in the is description. The text was updated to: Six out of seven mice learned the task (**Fig. S1**). Five of the mice that learned the task proceeded on to BCI training, and one mouse was trained on a BCI control experiment where the auditory pitches were not driven by neural activity. Both the BCI training and the BCI control experiment started within 8 days of learning the pitch association.

6. Figures S4-7: min, median, max change trials look comparable – were there any qualitative/quantitative differences in the basic descriptive statistics here across session 1 to last session?

Figures S4-7 were generated in response to Reviewer 3's concern that some threshold crossings could be due noise of individual direct neurons. For each mouse in session 1 and the LP session, we selected examples of direct neuron fluorescent activity from the trial that had the lowest control signal to provide evidence that even in the lowest threshold crossings it is likely that neural activity and not noise drove the threshold crossing. The figure legend is updated to reflect this. A statistical summary of changes in the mean fluorescence of direct neurons can be found in new Table 2.

7. Figure S11: why are there fewer Indirect-DP pairs than Indirect-DN pairs? If there are 3 DPs and 3DNs and X indirect neurons in the same FOV, why would these be different?

In the original version there was an unequal number of Indirect-DP and Indirect-DN pairs because only the indirect-direct pairs that were correlated on the first session were shown. This is no longer an issue, given the figure was updated to address Reviewer 1's concern that the implications should be refined.

Maybe useful to include indirect-indirect pairs as a 'baseline' information of general change across sessions in the population. Also, the significance of the related paragraph in the results (starting line 204) is somewhat lost. I think the new data is useful, but the implications of the findings should be refined.

We simplified the figure so that the significance is more clear. In the updated version, correlation analysis is simply used as a description of co-activity between direct and indirect neurons. The majority of indirect neurons are correlated with at least one direct neuron at the time of threshold crossing, in the first BCI session as well as the last BCI session. This is

important because it provides a premise for considering the possibility that tuning of the indirect neurons is altered by BCI training, given the frequency of threshold crossing increases with training. We no longer draw any conclusions regarding whether the correlation among indirect and direct neurons change. For that reason, we did not add a baseline characterization to the manuscript.

8. The control signal threshold is now reported (albeit only in the FigS8 legend) as mouse#1-4: 15,8,7 and 12 respectively. How was this determined for each mouse?

These details are now provided in the Methods, under section heading *Brain Computer Interface Task*, Selection of direct neurons and target threshold, and included here:

Prior to BCI training, we recorded a 10-minute baseline activity session, in the dark, without auditory stimuli or reward. Selection of direct neurons and threshold was then determined from this baseline recording. We first selected a pool of 3 DP and 3 DN neurons randomly from the segmented population. We then ran an algorithm to determine the threshold for those neurons. That algorithm proceeded as follows. First, we only considered thresholds within the range 6-15, in an attempt to ensure we were neither too close to the noise floor nor including potential large, outlier transients. Second, we searched for thresholds that would result in between 12 and 18 threshold crossings within 10 minutes. This corresponds to our target non-learning success rate of 20-30%. We initialized our search at a threshold of 10, and adjusted the threshold in a step size of 1 to explore the space to determine the threshold value that resulted in the desired success rate. If 10 resulted in the right number of threshold crossings, we stopped, but if it was too many we increased the threshold by 1 and if too few we decreased the threshold by 1, continuing until we first found a threshold that resulted in the 12-18 threshold crossings or until we were out of range. If we were out of range, we selected another group of 6 neurons at random, and repeated. Once a threshold was found that resulted in a success rate of 20-30%, the search was stopped. On each iteration, all neurons were sampled from the entire available pool, including neurons that were selected from previous iterations.

9. Figure 5: the difference in Fig 5D (visual discrimination training) and Fig S15C is entirely unclear and seem contradictory. In Fig 5D – in addition to the decrease in 90 degrees - there is clearly also an increase in the fraction of responsive neurons to the orientations around 0 and 180 – presumably the vertically rewarded orientations. What is the data in Fig S15C representing? The results paragraph starting at line 400 is equally confusing in this regard.

Previous studies (e.g. Poort et al., 2015; Henschke et al., 2020, and others) have shown that there is an increased representation of selective neurons in V1 to task-relevant gratings during goal-directed behaviours, and increased discrimination between two task-related gratings (Poort et al., 2015). It is unclear how the authors interpret that their results relate to these previous findings. Figure 5D seems to indicate that their data show something similar (perhaps somewhat more complex with the orthogonal [90 degree] representation decreasing dramatically, although this was also evident in the Henschke et al., 2020 results), but this is not discussed (in results or discussion).

Yes, our results are consistent with Henschke 2020 and the recent Khan 2022 paper, however the experimental conditions are different in that in our study neural activity is acquired outside of

the training environment. We updated the premise (Results), relate our findings directly to Henschke in the Results, and expanded the Discussion as suggested.

Similarly, the decoding results are unclear. Is Fig 5E because the decoder is not sensitive enough to changes that are seen in 5D?

We use KNN classifiers to answer two different questions, and implemented the classifier in two different ways. In the first implementation we ask, given all the responsive neurons available, how accurately can stimulus orientation be decoded? This is a method to estimate the amount of information present in the neural activity recorded. It is interesting that despite the relative decrease in the number of neurons responding to 90 degree orientations, accuracy is maintained. This is now more clearly emphasized in the Discussion (last paragraph). In the second implementation, we used a 'fixed-classifier' in which the same tracked neurons, irrespective of whether they are responsive to the visual stimuli presented, are used in the classifier. This second implementation allows us to representational drift, and is indeed sensitive to the change that is seen in 5D.

What are the decoding properties specifically for task-related orientations?

Thank for this comment. We added this information. The new analysis is useful because it helps relate our results to Henschke 2020.

10. Line 110: a brief description of what the VS baseline sessions consisted of would be useful here. The word 'tuning' is mentioned at some point in the paragraph, but beyond that there is no indication of what conditions these sessions were recorded under.

A brief description was added: 'The acquisition of two VS baseline sessions allowed the stability of tuning to grating stimuli of varying orientation and spatial frequency to be assessed before BCI training was initiated.'

11. Line 127: it seems very important to mention here that it increases in their "summed" z-score activity that drove the cursor – in this way, one DP cell can dominate as is indicated in Fig 2. This is not clear until much later and if it was the case that all 3 would need to reach a certain z-score threshold individually, this would presumably lead to a very different outcome.

Agreed, 'summed' was added.

12. The description of how cells were selected is improved, however, I have some remaining questions. Why 12-18? This seems arbitrary (was this based on the step-size of the pitch increase per threshold crossing???) If so, this should be mentioned), but also seems like a wide range. If an arbitrary number is picked why not a single number that can apply for all mice. Also, the thresholds ranging from 6-15 is quite a large range, and equally arbitrary. It seems a bit strange to have both of these values as a range. What was chosen between a group of 6 cells with threshold 7 with 15 crossings and a group of 6 cells with threshold 10 and 14 crossings. Surely there was more than one group of 6 combinations of neurons in a FOV of that fell into these wide ranges? Then was a visual inspection applied to pick the final group?

We can see how the revised description still missed the mark. Several items are worth noting. We chose a particular set of direct neurons at random, and then we ran an algorithm to determine threshold. If the algorithm found a threshold that resulted in 12-18 threshold crossings within 10 minutes of recording- which corresponded to target baseline success rate of

20-30%, we stopped. There were likely many groups of neurons that would have been appropriate, but this search procedure stopped after identifying the first group. We have added a section to the methods *Brain Computer Interface Task*, Selection of direct neurons and target threshold, that describes in detail this selection procedure; see also our response to comment #8 above. Yes, most likely there were more than one group of six cells that met the criteria, however, given the search algorithm, we stopped once one set of 6 was identified.

13. It was previously requested to define the learning point more clearly. The description in the legend helps, but the text (lines 152-157) remains a bit unclear.

We updated the description, and added new analysis given that we are now including all BCI sessions:

All mice exhibited significant improvement in performance, defined as the frequency of trials in which the target was reached, within 4-8 days of training. Performance of a given session was considered significantly improved from the first session using a 2-sample Z-proportions Z-test, corrected for multiple comparisons (see Methods for analysis details). The rate of learning varied across mice, therefore in order to define a learning state that could be compared across mice, the earliest session at which linear regression of performance against session number was significant ($p < 0.05$) was identified and is referred to as the learning point (LP) session. The LP session was used to examine BCI-induced changes in direct neurons. All mice maintained the skill after the LP session; task performance was significantly higher in the last BCI session relative to the first BCI session in all mice (**Fig.1D**).

I also do not think the comment from reviewer 3 was adequately addressed in this regard: 'How did the performance evolve after the day indicated by the "learning point"?' I would think it would be possible to include all sessions from 1 up to the last visual stim imaging session in the plots for Fig 1C and to indicate with the arrow (or dot, etc., and to keep the associated red regression line) which was the learning point session. This way, the whole learning trajectory is shown (as is done with the new task in Fig 5B), and it is also clear that the designated LP (Fig 2 etc.) and the last session (Fig S10) are used as example days along the training continuum. In my experience mice often show variable performance across sessions even after they may have reached a learning criterion, this would be useful information for the reader and for reproducibility.

We agree, and updated the BCI performance plots to include all sessions. See also our response to Reviewer 3 comment #4.

14. I appreciate the data value added by the heatmaps (Fig 2C, S11, D, E, etc) – but they are somewhat difficult to parse visually. The scalebar may be too large and there seem to be abrupt transitions between zero and about 1-2. For example, the map goes from black (0) to dark pink (~1-2) within a single data point on the x-axis. This makes abrupt transitions between delta of 0-2 visually highlighted - which are irrelevant to the point. The important transitions are the higher ranges (light pink/white) which take some time to visually pick out. Either a change in the colour map (should not be strictly necessary) or an alteration of the scale (perhaps better) seems warranted to fix this. Visually, this works much better in Figure 3A for instance.

The abrupt transition appears in trials that the target threshold was crossed with a latency less than 3.8 seconds, we used gray to denote the trial was shorter. We agree, this was not effective and lightened the gray color to make more clear.

15. Why were both Emx1cre and SLC17a7cre mice used? I did not see a description of when these lines were used, i.e., for which experiments. Are there systematic differences between the populations of excitatory neurons labelled with these two lines?

Information regarding when these lines were used can be found in former Table 5 (re-labeled as Table 1); we now refer to this table in the main text in addition to the Methods section. It is unlikely that there are systematic differences between the two lines- both lines label roughly 80% of all excitatory neurons. In visual cortex, the Allen Brain Institute reports similar response properties across the two lines. Initially we maintained a breeding colony of mice to produce experimental mice that contain the following three alleles: Emx1cre x Ai93 (the sensor allele) x CamKII tet (the enhancer allele). However, some labs reported that Emx1cre triple allele mice show aberrant brain activity (Steinmetz et al. 2017, PMID 28932809). Therefore, the field has moved away from using Emx1cre mice crossed with Ai93. Although we did not find a high prevalence of aberrant brain activity in our low-density housed colony, and none of the mice used in this study exhibited aberrant activity, to help ensure studies can be compared across labs, we changed to using SLC17a7cre x Ai93 x CamKII tet.

16. Lines 174-176. This justification sentence reads to me as if there was a misinterpretation of the aim of the playback control. The point is not anything specific to sessions >10 (this is just a necessity to have the same amount of training as is present in the most trained experimental mice). Although this control would be relevant to type 1 errors, in my mind, a more straightforward description would be to control for any systematic increases in activity that may spontaneously occur with exposure to the task elements (reward, etc.). Please check this.

Thank you. We refocused the analysis on the maximum LP session number across the experimental mice, which was 10, and updated the sentence to: An additional control was performed to control for any systematic increases in neural activity that may spontaneously occur with exposure to elements of the BCI task, such as repeated head fixation and reward delivery.

17. Lines 186-190: The authors rely on Dippopa et al., 2018 to make the point that locomotion during darkness does not modulate neurons in V1 to a great extent. However, they also have this data in their own dataset presumably. Can they not provide evidence for/replicate this finding in their own mice/FOVs? For instance, with a simple index of activity modulation (active vs inactive periods as indicated in Fig S9) during the BCI task in the dark and visual stim baseline days in their populations (as in Fig 4). According to their methods this locomotion data was removed for tuning calculations, but there is no reason why they can not compare general stim (without tuning) vs dark using this same data. For Figure S9, locomotion may not be significantly correlated with overall performance, but could still be correlated to the activity of an individual DP (or DN) neuron (or the indirect populations) – since the performance is dictated by the sum of activity. This is still an important point on the population activity and single cell level.

Thank you for this suggestion. We calculated the locomotion modulation index as suggested. The analysis is presented in new Figure S12 A,B. We found that the impact of locomotion on activity levels was similar for DP and DN neurons. See also our response to Reviewer 1

comment #3.

Minor comments:

- Line 53: specify genetically encoded 'calcium' indicator

Updated as suggested.

- The sentence starting line 158 is confusing. The end of the sentence could use rewording.

Updated to: We characterized the activity of the 6 direct neurons at the time of success and noted that most of the drive to cross the target threshold in BCI session 1 came from the DP neurons, and this was also true in the LP session.

- Line 361: pdf missing degree symbol

Corrected

- Line 208: missing 'correlated'

This sentence was removed.

- Line 231: here, I would refer to the appropriate Figure panel for evidence of this improvement.

Updated as suggested.

- Lines 251 to 252: the word excitability is again used here instead of activity (see previous comment from reviewer 1). Please check throughout manuscript.

Corrected as suggested, here and two other places were identified.

- Fig 2D & E: where are the 'no change' neurons from D represented in E?

The 'no change' neurons are not represented in the histogram. We clarified the legend. The purpose of the plot is to show the neurons that scored as being significantly different between the two sessions.

- Line 257: include relevant stats at the end of the sentence (similar to line 264). What are the stats in the Fig 2F in relation to this?

Thank you, we updated the two descriptions to match, and now report Wilcoxon rank-sum test results for both the baseline control and BCI conditions. Note, there was a typo, the KS-test result was 0.180 in the control condition.

Line 259: check wording... indirect-direct indirect

The sentence was simplified to: This is an indication that BCI training drove an increase in functional connectivity among V1 neurons that persisted outside of the BCI task context.

- Lines 339 and 407: presumably S15 is meant here? And related: paragraph starting line 430: S16.

Corrected.

- Fig S15A: add mouse as an x-axis label in A

Updated as suggested.

- Line 461: 'assess' grammar

Updated to: The original functional properties of the network were assessed before and after the new skill was acquired.

Reviewer #3 (Remarks to the Author):

The manuscript has much improved and we appreciate the detailed answers and new figures which allowed us to better assess this study. The new data of the visual discrimination task in the second part is also an important improvement. The selection process of the conditioned neurons is clearer now and the shuffling analysis is a nice control.

Regarding the robustness of the BCI learning and how it supports the overall conclusions, the clarifications and details provided have unfortunately rather increased our concerns. We currently have the following interpretations:

We are uncertain why the new data and analysis have increased concerns regarding the robustness of BCI learning, given the enthusiasm that all three reviewers have for the shuffle analysis in which null distributions were generated using indirect neurons (current Figure 1E, former Fig S8,). The shuffle analysis allows us to conclude that statistically it is unlikely to see increased threshold crossings across sessions by chance. Furthermore, we now provide evidence that BCI-training induces changes in noise correlation that persist outside of the BCI-context. Regarding issues of clarity, we thank the reviewer for pointing specific cases in which it was unclear how the results support the overall conclusions- the manuscript is much improved by addressing the concerns of clarity.

(1) During the BCI learning process, only a small fraction of the 24 direct neurons actually change their firing pattern (S4-S8, old figure 5). The authors admit that it is mainly single (or maximum two) DP neurons that show actual changes.

Correct, our task design does not require all DP neurons to be simultaneously active. Analysis presented in the first submission revealed that most often a single neuron dominated in driving the control signal to cross threshold. This same information was presented in the resubmission (former S4-S8 panel A, former figure 5), and kept in the current version (Figs. 2A, S8A-S11A).

DN neurons never seem to change with learning (in addition, the selection process clearly seems to favor silent DN neurons). According to table 2 and 3, this means that this entire study relies on a total of ~6 DP neurons (across 4 mice) which modify their activity during learning. Since only a fraction of those (~3-4 neurons) actually show visual tuning, we end up with a highly underpowered data set. We don't think it is correct to consider all 24 direct neurons (half of which are DNs and thus show very little activity during conditioning, and the other half which

show small changes) for the analysis, but only the ones that actually change during conditioning.

We agree, that there is an insufficient number of direct neurons to make strong conclusions regarding whether direct neurons themselves exhibit BCI-induced changes in visual tuning outside of the BCI task. Importantly, this is not the topic of the manuscript. In the Discussion we now clarify that the goal of the manuscript is to determine whether BCI training and acquisition of the new skill alters visual processing outside of the BCI context. The premise for considering that our specific BCI design may alter indirect neurons not coupled to the device is based on updated Figure 3B (former Figure 2D,E). The analysis presented demonstrates that the majority of indirect neurons exhibited plasticity in response to BCI training.

The following paragraph was added to the Discussion:

In principle there are 3 ways in which vision would not be disrupted by BCI training. One, a small number of neurons in V1 change their activity during task performance. In this scenario, it would be highly unlikely that training would in any way influence the cortex's ability to process visual stimuli. We can rule out this possibility because we found that approximately 80% of the neurons in V1 exhibited plasticity during BCI. Two, a substantial number of V1 neurons change their activity during BCI, and these changes persist outside of the BCI context and impact visual stimulus tuning. However, at the population level the changes are in the null space of the downstream readout. For example, the neurons that changed do not project to V2 and there is sufficient redundancy such that it is the non-relevant neurons that changed. This second possibility can be ruled out because we did not observe an impact of training on visual stimulus tuning. Three, a large number of neurons show plasticity during BCI, but tuning to visual stimuli in V1 remains stable. In other words, the changes are specific to the BCI context. This is the outcome that we observed. An implication of our findings is that the ability to distinguish contexts may be fundamental to the maintenance of stable perception while acquiring new skills.

Importantly, regardless of whether all 24, or 0 out of 24 of the direct neurons changed, the main conclusion of this manuscript would remain the same. After consultation with the editor, for this reason, it was determined that additional BCI mice are not required.

(2a) Comparing neurons which have not undergone BMI related changes with the ones that have not been conditioned does not seem an adequate way to ask if conditioning impacts existing functions and interaction.

We are unsure what the Reviewer means by 'Comparing neurons which have not undergone BMI related changes with the ones that have not been conditioned'. Our best guess that 'ones that have not been conditioned' is referring to the indirect neurons. If that is the case, then perhaps 'neurons which have not undergone BMI related changes' is referring to any neuron (indirect or direct) that did not exhibit a change in activity during the BCI task (e.g., not present in former Fig. 2D,E; updated Figure 3B)? This is correct, we did not make such a comparison, indirect neurons that were not identified as having undergone a change in activity during BCI are included in the analyses in former Figure 4; updated Figure 5. This is because the distributions were not shifted. If a fraction of indirect neurons did change their tuning, then it would be of interest to examine in more detail. However, this was not the case, so further analysis would not change the main conclusions of the manuscript.

(2b) Ideally, the conditioned neurons should have been chosen from a pool of neurons with intermediate baseline activity. This way the DN could have been conditioned to actually decrease and the DP neurons to increase. Using single cell conditioning would have been another alternative. As pointed out in our first assessment this would mean adding many more experimental animals to the study.

Although there are many possible variations of BCI one could implement, incorporating additional forms of BCI in new animals will not change the main conclusions of the manuscript. We note also that distributing changes across many neurons directly coupled to the device could actually decrease the amount of change per neuron. After consultation with the editor, for these reasons, it was determined that additional BCI mice are not required.

(3) In order to convince the reader that the BCI learning point leads to a stable plateau performance (similar to the one in Fig. 5B for the visual behavior) we asked to show the performance of all sessions following the learning point (old figure 5) for all 4 mice. New figure 10 does not address this point as it only compares the first with the very last session. If the behaviors do not reach a plateau, then it is questionable if the assessment of visual function a few days after the learning point is meaningful. Our concerns thus still remain.

Thank you for clarifying, we understood the original concern was that the animals did not retain the new skill at the time that the final visual stimulation session was performed. The comment here is related to Reviewer #2's request for performance on the days in between the learning point and the final BCI session before the last visual stimulation imaging session. We now provide performance information for all days. While our conclusions do not require that animals reach a plateau performance, as indicated by Reviewer #2 comment #13, including this information does add depth to the interpretation of the results. We provide strong evidence that all 4 animals learned the BCI task, and that the skill is retained at the last BCI session.

(4) Finally, why is a binomial test used for this paired comparison?

We thank the Reviewer for this question, and now provide more details of how this analysis was performed (see Methods Section, 'Two-sample proportions Z-test (*Figure 1*)', and include this analysis in updated Figure 1. Readability is much improved.

(5) According to our reading, the new figure S9A also shows that mouse 1 has a positive correlation of performance and running. We can thus ask if the observed changes are not simply due to an increase in movements in this mouse? Removing this particular mouse would further remove another one of the four remaining neurons from the analysis...

We are uncertain what is meant by 'observed changes'. The observed changes in noise correlation were measured in the absence of locomotion, we apologize if this was not clear and added this information to the Fig. 3C legend. The results allow us to conclude that independent of locomotion, BCI training induced persistent changes to activity patterns of V1 neurons. Given we are not sure precisely what the reviewer means by 'observed changes', we also note the following: The correlation between performance and running in Mouse 1 is not statistically significant ($r= 0.58$, $p= 0.078$). Thus we cannot conclude that there is a positive correlation, although it is the case that the session with the second highest running level has the best performance. Even if there were a correlation in Mouse 1, based on the analysis presented in former Figure S8A (new Figure 1E), we can conclude that the linear regression R^2 value calculated from the set of 6 neurons directly coupled to the auditory cursor is statistically distinct

from the distribution of R^2 values calculated from the indirect neurons in the same mouse, experiencing the same running, not directly coupled to the device. See also our response to Reviewer 1 comment #3 and Reviewer 2 comment #17.

(6) Playback control: the “fictive” neurons should not be chosen not randomly, but according to the selection criteria described for the BCI mice.

We agree, and did indeed select the ‘fictive’ neurons according to the same criteria as for the BCI mice. The word random was removed from the legend; we apologize for the confusion. See line 175 (former line 177-179): ‘Three ‘fictive’ direct positive and three fictive direct negative neurons were selected following the procedure described above, and tracked across 14 sessions; the six fictive neurons were not directly coupled to reward delivery or auditory feedback.’ We added this information in the new Figure S7 legend as well, to make sure that this is clear.

In addition, we don’t think that a single mouse is sufficient for this type of control.

In terms of establishing that the BCI mice learned the task, the results of the shuffle-control analysis asked for by the Reviewer in the previous round of revisions (Fig. 1E, former Fig. S8) reveal that the probability of finding 4 mice that had R^2 values above the 95th percentile (dashed blue lines, Fig. 1E) was extremely low, $0.05^4 = 6.25E-6$. Adding more ‘playback control’ mice will not change the main conclusions of the manuscript. After consultation with the editor, for these reasons it was determined that adding more ‘playback control’ mice is not required.

(7) The analysis carried out in S8 will also be important for these control mice.

We now include the analysis in former Fig. S8 for the playback control. As expected, unlike the experimental mice in which the control signal was coupled to the auditory cursor, the null hypothesis cannot be rejected in the control mouse (Fig. S7A,B). That is, the R^2 value calculated from the 6 ‘fictive’ neurons appears to come from the same distribution of R^2 values calculated from any random 6 neurons. Furthermore, we now use the playback control mouse to create null distributions using parameters matched to each BCI mouse; the observed R^2 value in each BCI mouse is higher than the 95th percentile identified in the null distribution created from the playback control mouse.

Minor comments:

Fig. 2A: Strategy instead of “Stratagy”?

Updated.

Fig5E: training instead of “traning”?

Updated.

Line 286: there is no Fig 3F?

Correct, updated.

The first sentence of the abstract might need revision, since the computation of abstract properties would not be the first thing that comes to my mind when thinking about learning a new skill.

Thank you, updated to: 'The ability to learn and practice new skills applies to abstract skills such as neuroprosthetic control, as well as perceptual discrimination.'

There seems to be a mix-up in the figure numbering S14, S15, S16 on lines 399-446?

Updated.

Taken together, we are very sorry to not be more positive despite all the efforts the authors have put into the revision of the current manuscript.

Daniel Huber & Antoine Philippides

REVIEWERS' COMMENTS

Reviewer #1 (Remarks to the Author):

The authors have addressed all Reviewers' concerns, improving the manuscript in precision and clarity in many places. The added shuffle analysis, the presentation of all the training sessions, and the example neuron traces in Figure 1 strengthen the BCI performance results convincingly. The removal of the darkness experiments is welcome in order to focus on the main data which is already very rich. The text has been improved in many places.

I only have minor concerns on the new version, on a few points which remain unclear.

1. First, one of my comments was obviously not understood. I'm sorry for the confusion. "Some sentences should be cut in several" meant "should be cut in several distinct sentences", not cut from the text. Fortunately, the authors indeed kept those sentences or only modified them. For the sentence in Abstract lines 12-17 in the new version, it is still very long and difficult to understand. I recommend improving the readability of this sentence by separating it in two sentences. On lines 100-101, I would replace the first comma by a point, again creating two sentences.
2. Abstract line 12, "... the extent TO WHICH new skill learning..." would be nicer
3. Line 280, for clarity, please avoid the two ";".
4. Line 307, correct "baseline"
5. Line 335, I assume "amplitude of the stimulus" should be "response amplitude for the stimulus"
6. Line 347, is there a word missing between including and neurons? Direct, indirect?
7. Line 352, are the control mice the ones on Figure 5G, Left? Then why are there 6 points, when the text says 4 mice?
8. Line 362, correct "performace"
9. Line 362 and beyond, the added text might be better placed in the Introduction, around line 74.
10. Lines 424 and 448, refer to Figure 6 and not 5
11. Line 649, target threshold, add "(defined below)"
12. Line 664, range 6-15, please add this is the z-score value
13. Line 1227, p value missing
14. Line 1301, correct "than"
15. Line 1303, add "it"

Figures

1. Figure 1B, maybe it would be clearer to have the left scale in % of z-score threshold rather than the z-score raw values, which are not easy to understand and are specific to this case study. If you decide to keep the z-score values, I suggest writing somewhere on the graph it is a z-score.
2. Could you use a related color code for DP and DN in Figures 1 and 2? It's a bit puzzling right now.
3. Figure 4C, the labels have been improved. I still suggest:
 - replacing "Tuning similarity" by "delta Orientation preference", as in other figures
 - "delta $\leq 30^\circ$ " rather than " $\leq \text{delta } 30^\circ$ " (and same for \geq)
 - write DP2 next to the circle in the figure
4. Figure 4B, use a scale more similar to other figures: 0, 90, 135 degrees or better 0, 90, -45° ?
5. Figure 6A, writing 0° and 135° on the schematics would help
6. Figure 6B, you could indicate the imaging sessions, for example with arrows, at 0, 15 and 30
7. Figure 6, it took me a long time to understand when there were 6 mice and when 5, as it changes from one panel to the next. Maybe this could be improved in the main text and/or in the legend (even if it is in Table 1).
8. Figure 6D, I don't understand why there are 9 bars for 12 orientations tested. Could you keep it simple with one bar per orientation, centered on the orientation value as in panel F?
9. Figure 6D, you could indicate the 0° and 135° values, for example with arrows.
10. I don't think I understand Figure S7C, how it was obtained and what it means.
11. Suppl Figure 17B, is the control at 30 days or 15 days like in Figure 6D Left?
12. Suppl Figure 17C, what is the difference with Figure 6D? The legend says it is the same neurons

as Figure 6D Left but it cannot be, and the graphs look the same as Figure 6D Right, except the scale. Please clarify.

Reviewer #2 (Remarks to the Author):

The authors have largely addressed my comments. I appreciate the reorganization of the Figure panels and many details regarding the analysis methods are now clarified.

We thank the reviewers for their thoughtful comments and suggestions, the manuscript is improved. Our detailed responses can be found below.

Reviewer #1 (Remarks to the Author):

The authors have addressed all Reviewers' concerns, improving the manuscript in precision and clarity in many places. The added shuffle analysis, the presentation of all the training sessions, and the example neuron traces in Figure 1 strengthen the BCI performance results convincingly. The removal of the darkness experiments is welcome in order to focus on the main data which is already very rich. The text has been improved in many places.

Thank you for your positive feedback.

I only have minor concerns on the new version, on a few points which remain unclear. We are most grateful for bringing these issues to our attention and appreciate your time.

1. First, one of my comments was obviously not understood. I'm sorry for the confusion. "Some sentences should be cut in several" meant "should be cut in several distinct sentences", not cut from the text. Fortunately, the authors indeed kept those sentences or only modified them. For the the sentence in Abstract lines 12-17 in the new version, it is still very long and difficult to understand. I recommend improving the readability of this sentence by separating it in two sentences. On lines 100-101, I would replace the first comma by a point, again creating two sentences.

The abstract was updated and reduced to 150 words as required by the journal.

2. Abstract line 12, "... the extent TO WHICH new skill learning..." would be nicer
Updated as suggested.

3. Line 280, for clarity, please avoid the two ",".
Updated..

4. Line 307, correct "baseline"
Thank you. Corrected.

5. Line 335, I assume "amplitude of the stimulus" should be "response amplitude for the stimulus"
Updated to: amplitude of the stimulus response

6. Line 347, is there a word missing between including and neurons? Direct, indirect?
The phrase: ', including neurons response, but necessarily well-tuned for orientation or spatial frequency' was deleted.

7. Line 352, are the control mice the ones on Figure 5G, Left? Then why are there 6 points, when the text says 4 mice?
Thank you. Correct, there were 6 control mice (see also Table 1, column 'BCI decoding control Figure 5g'). The text was corrected.

8. Line 362, correct “performace”
Corrected.

9. Line 362 and beyond, the added text might be better placed in the Introduction, around line 74.

We agree that the rationale was a bit wordy. To address this, the rationale was edited to:

Converging evidence indicates visual discrimination training enhances the neural representation of rewarded stimuli by increasing selectivity for the stimuli experienced during training^{16,21} and in some cases improving response reliability¹⁶, and at the same time suppresses responses to non-relevant stimuli¹⁶. Enhanced responses to rewarded stimuli are known to generalize across task variations experienced in the training environment. However, selectivity for features such as orientation dissipates when reward contingencies are recognizably altered. As such, reward-induced changes in selectivity are considered to be context-specific. Furthermore, in many instances long-lasting changes observed in the training environment are restricted to stimulus-specific assemblies, and enhancement to more than one rewarded stimulus is possible due to assembly-specific plasticity^{20,21}.

10. Lines 424 and 448, refer to Figure 6 and not 5
Thank you, corrected.

11. Line 649, target threshold, add “(defined below)”
Updated as suggested.

12. Line 664, range 6-15, please add this is the z-score value
Updated as suggested.

13. Line 1227, p value missing
Thank you, corrected.

14. Line 1301, correct “than”
Thank you, corrected.

15. Line 1303, add “it”
Updated as suggested.

Figures

1. Figure 1B, maybe it would be clearer to have the left scale in % of z-score threshold rather than the z-score raw values, which are not easy to understand and are specific to this case study. If you decide to keep the z-score values, I suggest writing somewhere on the graph it is a z-score.

Thank you. Updated as suggested.

2. Could you use a related color code for DP and DN in Figures 1 and 2? It’s a bit puzzling right now.

We used the same scheme for 2b as in 1. We tried your suggestion in 2a, using various shadings of purple. However, the colors were not distinct. Given our goal is to emphasize the fraction of successful trials that each direct neuron strategy was used, we opted for the current color scheme. We updated the color representing the Σ DN strategy to brown, we hope that this alleviates the confusion.

3. Figure 4C, the labels have been improved. I still suggest:

- replacing "Tuning similarity" by "delta Orientation preference", as in other figures
- "delta $\leq 30^\circ$ " rather than " \leq delta 30° " (and same for \geq)
- write DP2 next to the circle in the figure

We reserved 'delta Orientation preference' as a measure of the change in orientation preference within a given neuron across sessions. In contrast, in Fig. 4C we are examining the difference in preferred orientation between indirect neurons and an example direct neuron. Therefore, to avoid the confusion between the two analyses, we elected to not use delta Orientation preference. We added 'Indirect-direct' in front of tuning similarity in the figure to make this clear and a 'DP2' label as suggested.

4. Figure 4B, use a scale more similar to other figures: 0, 90, 135 degrees or better 0, 90, -45° ?
Updated as suggested.

5. Figure 6A, writing 0° and 135° on the schematics would help
Updated as suggested.

6. Figure 6B, you could indicate the imaging sessions, for example with arrows, at 0, 15 and 30
Updated as suggested.

7. Figure 6, it took me a long time to understand when there were 6 mice and when 5, as it changes from one panel to the next. Maybe this could be improved in the main text and/or in the legend (even if it is in Table 1).

We now specify the color representing each mouse in Fig. 6g as in Fig. 6e to make it clear that mouse 2 from panel Fig. 6e was not included in the cross-session analysis.

8. Figure 6D, I don't understand why there are 9 bars for 12 orientations tested. Could you keep it simple with one bar per orientation, centered on the orientation value as in panel F?

Thank you, updated as suggested. (we originally thought that the Chi Square test assumptions might not be met if used >9 bins, however it turns out that the assumptions of the test are met when using 12 bins). The Chi Square p values are updated.

9. Figure 6D, you could indicate the 0° and 135° values, for example with arrows.
The axis labels are updated to indicate 0° and 135° .

10. I don't think I understand Figure S7C, how it was obtained and what it means.

Agreed. Briefly, we changed the labels in Fig. S7c to indicate that different target threshold were used, and updated the legend. Details: Figure S7C shows the null-distribution of R^2 values calculated using the threshold values from each of the 4 BCI mice. This was done to ensure that the range of threshold values that were used across the BCI mice did not result in

increasing performance with session number when neural activity was disassociated from BCI. We can see how the word 'parameter' made this ambiguous for the reader. The legend was updated to the following:

In addition to a target threshold of 7, the null distributions of R^2 values were calculated as in 'b' for the other three target thresholds used in this study. The dashed lines indicate the 95th percentile of the distribution. In all cases the actual performance of the BCI mouse was higher than the 95th percentile of its matching null distribution.

11. Suppl Figure 17B, is the control at 30 days or 15 days like in Figure 6D Left?
The control was 30 days in Suppl Figure 17b.

12. Suppl Figure 17C, what is the difference with Figure 6D? The legend says it is the same neurons as Figure 6D Left but it cannot be, and the graphs look the same as Figure 6D Right, except the scale. Please clarify.

Correct, the normalization is different. We clarified by updating the legend to:

Distribution of orientation preference for the neurons shown in Fig. 6d left, except that values were normalized by the total numbers of neurons imaged, rather than the number of visually responsive neurons.

Reviewer #2 (Remarks to the Author):

The authors have largely addressed my comments. I appreciate the reorganization of the Figure panels and many details regarding the analysis methods are now clarified.

Thank you for your positive feedback.